

# Snow albedo reduction in seasonal snow due to anthropogenic dust and carbonaceous aerosols across northern China

Xin Wang[1], Wei Pu[1], Yong Ren[1], Xuelei Zhang[2], Xueying Zhang[1], Jinsen Shi[1], Hongchun Jin[1],

Mingkai Dai[1], Quanliang Chen[3]

[1] Key Laboratory for Semi-Arid Climate Change of the Ministry of Education, College of Atmospheric Sciences, Lanzhou University, Lanzhou, 730000, China
[2] Key Laboratory of Wetland Ecology and Environment, Northeast Institute of Geography and Agroecology, Chinese Academy of Sciences, Changchun 130102, China
[3] College of Atmospheric Science, Chengdu University of Information Technology, and Plateau Atmospheric and Environment Laboratory of Sichuan Province, Chengdu 610225, China

Correspondence to: Xin Wang (wxin@lzu.edu.cn)





**Abstract**. A survey was performed to collect 92 seasonal snow samples across northern China in January 2014. The results show that mixing ratios of black carbon (BC), organic carbon (OC), and anthropogenic dust (AD) range from 60 to 4200 ng $g^{-1}$, 280 to 32000 ng $g^{-1}$, and 1300 to 5800 ng $g^{-1}$ in surface snow, respectively. A relatively high correlation ($R^2$=0.87, n=13) between OC and BC in surface snow is observed, indicating similar emission sources, and high OC/BC ratios suggest that biomass burning may be a major contributor at most snow sampling sites. In addition to the BC and AD parameters in the Snow, Ice, and Aerosol Radiation (SNICAR) model, the OC content in snow is considered an initial parameter for calculating snow albedo through a new radiative transfer model (Spectral Albedo Model for Dirty Snow, or SAMDS). The spectral albedo of snow reduction caused by OC (20 µg $g^{-1}$) is up to a factor of 3 for a snow grain size of 800 µm compared to 100 µm. We find a larger difference in snow albedo levels between the model simulations and surface measurements for higher insoluble light-absorbing impurities (ILAPs) using the measured snow grain radii. However, using the optical effective radii ($R_{eff}$), snow albedos modeled through both the SNICAR and SAMDS models are consistent with surface measurements, especially in the case of near-infrared wavelengths.



## 1 Introduction

Mineral dust (MD), black carbon (BC) and organic carbon (OC) are three insoluble light-absorbing particles (ILAPs) that play key roles in regional and global climate (IPCC, 2013; Jaffe et al., 1999). Anthropogenic dust (AD) is a major form of mineral dust. The AD content can influence air quality and human health through emission, transport, removal, and deposit processes (Aleksandropoulou et al., 2011; Chen et al., 2013; Huang et al., 2014, 2015a, 2015b; Kim et al., 2009; Li et al., 2009; Mahowald and Luo, 2003; Zhang et al., 2005, 2015). AD particles also act as cloud condensation nuclei that affect cloud formation (Coz et al., 2010; Givati and Rosenfeld, 2004; Rosenfeld et al., 2001) and deliver various trace nutrients to terrestrial and marine ecosystems (Acosta et al., 2011; Mahowald et al., 2005; Qiao et al., 2013). Ginoux et al. (2010) estimated that anthropogenic dust accounts for 25% of dust aerosols using observational data from MODIS Deep Blue satellite products combined with a land-use fraction dataset (Ginoux et al., 2010). Anthropogenic dust originates primarily from urban and regional sources, especially during the winter, and anthropogenic dust has already been enriched with heavy metals and other toxic elements (Kamani et al., 2015; Li et al., 2013; Wang et al., 2015; Zhang et al., 2013). Northeastern China and surrounding regions are generally regarded as industrial areas most affected by human activities. Because anthropogenic dust emissions from disturbed soils are not well constrained, we define mineral dust from areas disrupted by human activities, such as deforestation, overgrazing, agricultural and industrial activities, as anthropogenic dust (Aleksandropoulou et al., 2011; Tegen and Fung, 1995; Tegen et al., 2002, 2004; Thompson et al., 1988), which differs from natural mineral dust originating from desert regions (Che et al., 2011, 2013, 2015a; Goudie and Middleton, 2001; Li et al., 2012; Park and Park, 2014; Pu et al., 2015; Wallach and Fischer, 1970; Wang et al., 2008,




2010b). This assumption is consistent with the results of a recent study by Huang et al. (2015a), who found that anthropogenic dust contributions to regional emissions in eastern China are 91.8%, with India at 76.1% (e.g., Figure 10 in Huang et al., 2015a). This may be due to larger population densities, which are characterized by more intense

5 human activity in eastern China and India (Huang et al., 2015a; Wang et al., 2013a). Compared to AD, carbonaceous aerosols, such as BC and OC, generated from the incomplete combustion of fossil fuels and from biomass burning are also major anthropogenic pollutants. ILAPs deposited on snow have been found to shorten the snow cover season by decreasing the snow albedo and accelerating snow melt (Flanner

10 et al., 2007, 2009). Warren and Wiscombe (1980) found that a mixing ratio of 10 ng $g^{-1}$ of soot in snow can reduce snow albedo levels by 1%. Light et al. (1998) determined that 150 ng $g^{-1}$ of BC embedded in sea ice can reduce ice albedo levels by a maximum of 30% (Light et al., 1998). Among its main light-absorbing impurities, 1 ng $g^{-1}$ of BC has approximately the same effect on the albedo at 500 nm as 50 ng $g^{-1}$ of dust (Warren,

15 1982). Doherty et al. (2013) analyzed field measurements of vertical distributions of BC and other ILAPs in snow in the Arctic during the melt season and found significant melt amplification owing to an increased mixing ratio of BC by up to a factor of 5.

It is widely known that small snow albedo changes can have significant effects on global warming patterns, involving changes in snow morphology, sublimation, and melt

20 rates. Bond et al. (2013) estimated the industrial-era climate forcing of BC through all forcing mechanisms to be approximately +1.1 W $m^{-2}$, with 90% confidence limits of +0.17 to +2.1 W $m^{-2}$. This value includes the net effect of BC on radiation, clouds, and snow albedo, which have been found to heavily impact the earth's climate. Recent modeling studies have estimated regional and global radiative forcing caused by ILAPs

25 deposited on snow by reducing the surface albedo (Flanner, 2013; Flanner et al., 2007;



Hansen and Nazarenko, 2004; Jacobson, 2004; Koch et al., 2009; Qian et al., 2015; Zhao et al., 2014). Climate models have indicated that the reduction in surface albedo caused by BC leads to global warming and nearly global melting of snow and ice (Hansen and Nazarenko, 2004). Regional forcing by BC contamination in snow in

snow-covered regions, e.g., the Arctic and Himalayas (0.6 and 3.0 W m$^{-2}$, respectively), is comparable to carbon dioxide levels (1.5 W m$^{-2}$) in the atmosphere since the pre-industrial period (Flanner et al., 2007). The efficacy of radiative forcing due to BC deposited on snow has been estimated to be as high as 236% (Hansen et al., 2005).

Although AD is a less efficient absorber than BC and OC, in situ measurements of

seasonal snow across northern China and the Himalayas have shown high AD loadings (Guan et al., 2015; Kang et al., 2016; Wang et al., 2012, 2013a). In some regions, especially areas with thin and patchy snow cover and mountainous regions, soil dust significantly decreases the snow albedo, exceeding the influence of BC. However, models do not capture these potentially large sources of local dust in snowpack and

may overestimate BC forcing processes (Painter et al., 2007, 2010, 2012). Recently, several seasonal snow collection campaigns were performed across northern China, the Himalayas, North America, Greenland and the Arctic (Cong et al., 2015; Dang and Hegg, 2014; Doherty et al., 2010, 2014; Huang et al., 2011; Xu et al., 2009, 2012; Zhao et al., 2014). However, determining the effects of light-absorbing impurities on snow

albedo reduction continues to involve numerous challenges (Huang et al., 2011; Wang et al., 2013a; Ye et al., 2012; Zhang et al., 2013). We analyzed observed ILAPs in seasonal snow through a Chinese survey in 2014 following a snow campaign held in 2010 across northern China carried out by Huang et al. (2011). The area is seasonally covered with snow for approximately 3-6 months from late fall to early spring (Wang

et al., 2014). Our sampling sites were located across northeastern China and were





positioned far from the northern boundaries of desert regions, such as the Gobi Desert in southern Mongolia and the Badain Jaran and Tengger Deserts in northwestern China (Li et al., 2009). Zhang et al. (2013), using a positive matrix factorization (PMF) receptor model, showed that industrial pollution sources are a major factor affecting

seasonal snow across northeastern China. Huang et al. (2015a) developed a new technique for distinguishing anthropogenic dust from natural dust based on Cloud-Aerosol Lidar and Infrared Pathfinder Satellite Observation (CALIPSO) dust and planetary boundary layer (PBL) height retrievals along with a land-use dataset. The authors found that the annual mean contribution of anthropogenic dust in eastern China

is approximately 91.8% owing to recent urbanization and human activity (Huang et al., 2015a). In this paper, we explore the climatic effects of ILAPs (including BC, OC, and AD) on seasonal snow across northeastern China, which are highly correlated with industrial pollution resulting from human activity. Therefore, ILAPs in seasonal snow are examined during a snow campaign, and the snow albedo is measured using an HR-

1024 field spectroradiometer and simulated using two radiative transfer models (i.e., SNICAR and SAMDS).

## 2   Experimental procedures

### 2.1   Snow field campaign

In 2014, there was less snowfall in January than in previous years (e.g., 2010), and only

92 snow samples at 13 sites were collected. Samples from sites 90-93 were collected from grassland and cropland areas in Inner Mongolia. Sites 94-98 and sites 99-102 were located in the Heilongjiang and Jilin provinces, respectively, which are the most heavily polluted areas in northern China during winter. The snow sampling routes were similar to those used in the previous survey conducted in 2010 across northern China (Huang




et al., 2011). To prevent contamination, the sampling sites were positioned 50 km from cities and at least 1 km upwind of approach roads or railways; the only exception was site 101, which was positioned downwind and close to villages. We gathered snow samples every 5 cm from the surface to the bottom unless a particularly dusty or

polluted layer was present. Snow grain sizes were measured by visual inspection on millimeter-gridded sheets viewed through a magnifying glass. The snow samples were kept frozen until the filtration process was initiated. In a temporary lab based in a hotel, we quickly melted the snow samples in a microwave, let them settle for 3-5 minutes, and then filtered the resulting water samples through a 0.4-μm nuclepore filter to extract

particulates.

## 2.2    Chemical speciation

The chemical analysis of snow samples from the 2014 Chinese survey was described by Wang et al. (2015), and it was similar to the analysis applied for snow samples described by Hegg et al. (2009, 2010). Briefly, major ions ($SO_4^{2-}$, $NO_3^-$, $Cl^-$, $Na^+$, $K^+$,

and $NH_4^+$) were analyzed with an ion chromatograph (Dionex, Sunnyvale, CA), and trace elements were measured by inductively coupled plasma mass spectrometry (ICP-MS). These analytical procedures have been described elsewhere (Yesubabu et al., 2014). Previous studies have revealed considerable variations in iron (Fe) of 2-5% in dust (Lafon et al., 2006), although Al is more stable than Fe in the earth's crust. Hence,

we retrieved the mass concentration of minerals via the Al concentration assuming a fraction of 7% when estimating the dust levels (Arhami et al., 2006; Lorenz et al., 2006; Zhang et al., 2003). Sea salt was estimated following the method presented in Pio et al. (2007):

$$\text{Sea salt} = Na_{Ss}^+ + Cl^- + 0.12Na_{Ss}^+ + 0.038Na_{Ss}^+ + 0.038Na_{Ss}^+ + 0.25Na_{Ss}^+, \qquad (1)$$





where $Na_{Ss}$ was calculated using the following formula (Hsu et al., 2009):

$$Na_{Ss} = Na_{Total} - Al \times (Na/Al)_{Crust}. \qquad (2)$$

Following Hsu et al. (2009), the contribution of $K^+_{Biosmoke}$ was determined using the

following equations:

$$K^+_{Biosmoke} = K_{Total} - K_{Dust} - K_{Ss}, \qquad (3)$$

$$K_{Dust} = Al \times (K/Al)_{Crust}, \qquad (4)$$

$$Na_{Ss} = Na_{Total} - Al \times (Na/Al)_{Crust}, \qquad (5)$$

$$K_{Ss} = Na_{Ss} \times 0.038. \qquad (6)$$

Equations (4), (5), and (6) were derived from Hsu et al. (2009) and Pio et al. (2007).

**2.3    Spectrophotometric analysis**

Assuming that iron originated from mineral dust in seasonal snow, we measured the

mixing ratios of BC and OC using an integrating sphere/integrating sandwich

spectrophotometer (ISSW), which was first described by Grenfell et al. (2011) and used

by Doherty et al. (2010, 2014) and Wang et al. (2013a) to measure mixing ratios of BC

and OC. The equivalent BC ($C_{BC}^{equiv}$), maximum BC ($C_{BC}^{max}$), estimated BC ($C_{BC}^{est}$),

fraction of light absorption by non-BC ILAPs ($f_{non-BC}^{est}$), and absorption Ångstrom

exponent ($\text{Å}_{tot}$) were described by Doherty et al. (2010). Previous studies have

concluded that light-absorbing particles are primarily derived from BC, OC, and iron

(Fe). The mass loadings of BC and OC were calculated using the following equation:

$$\tau_{tot}(\lambda) = \beta_{BC}(\lambda) \times L_{BC} + \beta_{OC}(\lambda) \times L_{OC} + \beta_{Fe}(\lambda) \times L_{Fe}. \qquad (7)$$

Here, $L_{BC}$ and $L_{OC}$ can be determined from this equation assuming that the mass

absorption efficiencies ($\beta$) for BC, OC, and Fe are 6.3, 0.3, and 0.9 $m^2 g^{-1}$, respectively,



at 550 nm and that the absorption Ångstrom exponents for BC, OC and Fe are 1.1, 6, and 3, respectively (e.g., Equations (2) and (3) in Wang et al., 2013a).

## 2.4 Aerosol optical depth and snow albedo measurements

We used a portable and reliable Microtops Ⅱ Sun photometer at wavelengths of 340,

440, 675, 870, and 936 nm instead of the CE318 sun tracking photometer (Holben et al., 2006) to measure the in situ aerosol optical depth (AOD). Morys et al. (2001) provided a general description of the Microtops Ⅱ Sun photometer's design, calibration, and performance. The Microtops Ⅱ Sun photometer has been widely used in recent years (de Mourgues et al., 1970; Porter et al., 2001; Zawadzka et al., 2014) and is

recognized as a very useful tool for validating aerosol retrievals from satellite sensors. A Microtops Ⅱ Sun photometer was calibrated following the methods presented by Morys et al. (2001) and Ichoku et al. (2002b). To better understand the background weather conditions in the local atmosphere, a Microtops Ⅱ Sun photometer was used during the 2014 Chinese survey. AOD measurements were collected in cloud-free

conditions between 11:00 am and 1:00 pm (Beijing local time) to prevent the effects of optical distortions due to large solar zenith angles. We used the Moderate Resolution Imaging Spectroradiometer (MODIS) on the Aqua and Terra satellites to retrieve the AOD and fire spot datasets (Kaufman et al., 1997; Zhang et al., 2013; Zhao et al., 2014). The retrieved MODIS AOD is reliable and accurate when applied to three visible

channels over vegetated land and ocean surfaces (Chu et al., 2002; Ichoku et al., 2002a; Remer et al., 2002). Fire locations are based on data provided by the MODIS FIRMS system from October to January.

Spectroradiometers have been used to measure the surface spectral albedo (Kotthaus et al., 2014; Wright et al., 2014; Wuttke et al., 2006). In the 2014 Chinese survey, snow





albedo measurements were obtained using a HR-1024 field spectroradiometer (SVC, Spectra Vista Corporation, Poughkeepsie, NY, USA). This instrument has a spectral range of 350-2500 nm with resolutions of 3.5 nm (350–1000 nm), 9.5 nm (1000–1850 nm), and 6.5 nm (1850–2500 nm). According to previously described procedures, we

measured the snow albedo 1 m above the ground (Carmagnola et al., 2013). A standard "white" reflectance panel with a VIS–SWIR broadband albedo of 0.98 ($P_\lambda$) was used to measure the reflectance spectra along with the target. The reflectance spectra of surface snow ($R_s$) and the standard panel ($R_p$) were measured at least ten times. Then, the snow albedo ($\alpha$) was calculated as follows:

$$\alpha = (R_s/R_p) \times P_\lambda. \tag{8}$$

Further information on HR-1024 field spectroradiometer use and on the calibration procedure can be found in Wright et al. (2014).

## 2.5 Model simulations

BC and dust sensitivity effects on the snow albedo simulated by the Snow, Ice, and

15 Aerosol Radiation (SNICAR) model have been validated through recent simulations and field measurements (Flanner et al., 2007, 2009; Qian et al., 2014; Zhao et al., 2014). We used the offline SNICAR model to simulate the reduction in surface snow albedo resulting from ILAP contamination (Flanner et al., 2007), and we compared the results with our spectroradiometer surface measurements. The SNICAR model calculates the

20 snow albedo as the ratio of the upward and downward solar flux at the snow surface. The measured parameters, including the snow grain radius, snow density, snow thickness, solar zenith angle, and mixing ratios of BC and AD, were used to run the SNICAR model under clear sky conditions. The visible and near-infrared albedos of the underlying ground were 0.2 and 0.4, respectively, as derived from MODIS remote



sensing. The mass absorption cross section (MAC) was assumed to be 7.5 $m^2$ $g^{-1}$ at 550 nm.

The Spectral Albedo Model for Dirty Snow (SAMDS) for calculating spectral snow albedo as a function of the snow grain radius, the mixing ratios of ILAPs (BC, AD, and OC), and mass absorption efficiencies of impurities was used and is based on asymptotic radiative transfer theory. This model explicitly considers (i) mixing states between impurities and snow grains, (ii) the irregular morphology of snow grains and aerosol particles, (iii) specific mineral compositions and size distributions of dust in snow, (iv) aging processes of snow grains and soot aggregates, and (v) multilayers for studying vertical distributions of snow grains and impurities.

Briefly, the surface albedo can be calculated using the following asymptotic approximate analytical solution derived from radiative transfer theory (Kokhanovsk and Zege, 2004; Rozenberg, 1962; Zege et al., 1991):

$$R_d(\lambda) = \exp(-4S(\lambda)\mu(v_0)). \tag{9}$$

Here, $R_d(\lambda)$ is the plane albedo, $v_0$ is the solar zenith angle, and $u(v_0)$ can be parameterized following Kokhanovsky and Zege (2004):

$$\mu(v_0) = \frac{3}{7}(1+2\cos v_0 ), \tag{10}$$

where $\lambda$ is the wavelength, $S(\lambda)$ is the similarity parameter, and

$$S(\lambda) = \sqrt{\frac{\sigma_{abs}}{3\sigma_{ext}(1-g)}}. \tag{11}$$

Here, $\sigma_{abs}$ and $\sigma_{ext}$ are the absorption and extinction coefficients, respectively, and $g$ is the asymmetry parameter (the average cosine of the phase function of the medium).





According to Equations (18) and (25) in Kokhanovsky and Zege (2004), the extinction coefficients of particles can be expressed as follows:

$$\sigma_{ext} = \frac{l_{tr}}{1-g} = \frac{3C_v}{2r_{ef}},\tag{12}$$

where $l_{tr}$ is the photon transport path length, $C_v$ is the volumetric snow particle

concentration, and $r_{ef}$ is the effective grain size, which is equal to the radius of the

volume-to-surface equivalent sphere: $r_{ef} = \frac{3\overline{V}}{4\overline{A}}$, where $\overline{V}$ and $\overline{A}$ are the average

volume and average cross-sectional (geometric shadow) area of snow grains, respectively.

The absorption coefficient $\sigma_{abs}$ in Equation (11) for arbitrarily shaped and weakly

absorbing large grains is proportional to the volume concentration (Kokhanovsky and Zege, 2004):

$$\sigma_{abs} = B \cdot \frac{4\pi k(\lambda)}{\lambda} \cdot C_v,\tag{13}$$

where $k(\lambda)$ is the imaginary component of the complex refractive index for ice, and

$B$ is a factor that is only dependent on the particle shape.

The total absorption coefficient, $\sigma_{abs}$, can be derived from the absorption by snow,

$S_{abs}^{snow}$, and the absorption by light-absorbing impurities, $S_{abs}^{dust}$, $S_{abs}^{BC}$, and $S_{abs}^{OC}$:

$$\sigma_{abs} = \sigma_{abs}^{snow} + \sigma_{abs}^{dust} + \sigma_{abs}^{BC} + \sigma_{abs}^{OC}.\tag{14}$$


The hemispherical reflectance with a zenith angle $v_0$ can be expressed as follows:

$$R_d(\lambda) = \exp\left(-4 \cdot \sqrt{\frac{4B}{9(1-g)}} \cdot \frac{2\pi \cdot r_{ef}}{\lambda} \cdot k(\lambda) + \frac{\rho_{ice} \cdot 2 r_{ef}}{9(1-g)} \cdot MAC_{abs}^{dust} \cdot C_{dust}^* + \frac{\rho_{ice} \cdot 2 r_{ef}}{9(1-g)} \cdot MAC_{abs}^{BC} \cdot C_{BC}^* + \frac{\rho_{ice} \cdot 2 r_{ef}}{9(1-g)} \cdot MAC_{abs}^{OC} \cdot C_{OC}^* \cdot \frac{3}{7}(1+2\cos v_0)\right)$$

for spherical grains;

$$\exp\left(-\sqrt{94.746 \cdot \frac{r_{ef}}{\lambda} \cdot k(\lambda) + 5.163 \cdot r_{ef} \cdot (MAC_{abs}^{dust} \cdot C_{dust}^* + MAC_{abs}^{BC} \cdot C_{BC}^* + MAC_{abs}^{OC} \cdot C_{OC}^*)} \cdot (1+2\cos v_0)\right),$$

for hexagonal grains;

$$= \exp\left(-4.95 \cdot \sqrt{\frac{\pi \cdot r_{ef} \cdot (k(\lambda) + \alpha \cdot C_{dust}^* + \beta \cdot C_{BC}^* + \chi \cdot C_{OC}^*)}{\lambda}} \cdot (1+2\cos v_0)\right),$$

for hexagonal grains;

$$\exp\left(-4.38 \cdot \sqrt{\frac{\pi \cdot r_{ef} \cdot (k(\lambda) + \alpha \cdot C_{dust}^* + \beta \cdot C_{BC}^* + \chi \cdot C_{OC}^*)}{\lambda}} \cdot (1+2\cos v_0)\right),$$

for fractal grains.

(15)





A detailed description of asymptotic analytical radiative transfer theory for snowpack was presented by Zhang et al. (2016). Previous studies have also shown that the spectral snow albedo is more sensitive to snow grain size and light conditions than BC contamination and snow depths at near-infrared wavelengths (Warren, 1982). Therefore,

the snow grain optical effective radius was retrieved based on the spectral albedo measured at $\lambda$=1.3 $\mu$m (Table 4), where snow grain size dominates the snow albedo variations and the effects of light-absorbing particles at this wavelength are negligible (Warren and Wiscombe, 1980).

**3        Results**

**3.1      The spatial distribution of AOD**

The AOD is a major optical parameter for aerosol particles and a key factor affecting global climate (Holben et al., 1991, 2001; Smith et al., 2014; Srivastava and Bhardwaj, 2014). For most of the snow samples collected in the afternoon at the Aqua-MODIS (13:30 LT) overpass time, the averaged spatial AOD distribution was derived from the

Aqua-MODIS satellite "Aerosol" product during the sampling period across northern China. The AOD spatial distribution over northern China is shown in Figure 1 for the period from October to January. The average AOD in the studied area ranged from 0.1 to 1.0 and exhibited strong spatial inhomogeneity. The largest AODs (up to approximately 1.0) retrieved from the MODIS satellite were associated with

anthropogenic pollution over northeastern China during the 2014 sampling period. These large values, which exceeded 0.6, were related to local air pollution from industrial areas (Che et al., 2015b; Wang et al., 2010a). In contrast, the MODIS-Aqua results indicate that the smallest AODs (as low as 0.1) at 550 nm were found over the Gobi Desert in Inner Mongolia and were related to strong winter winds. Similar patterns





in the retrieved MODIS AOD were found by Zhao et al. (2014) and Zhang et al. (2013).
Although previous studies have indicated that AODs in northeastern China are among
the highest in East Asia (Ax et al., 1970; Bi et al., 2014; Che et al., 2009; Routray et al.,
2013; Wang et al., 2013b; Xia et al., 2005, 2007), field experiments of aerosol optical

properties across northeastern China have been limited. To evaluate the accuracy of the
AODs, in situ AOD measurements were collected during the snow sampling period in
January 2014. The in situ AOD measurements were generally consistent with the spatial
AOD distribution retrieved from MODIS across northern China, although the retrieved
AOD was slightly higher than that retrieved from the in situ measurements in the study

area. In Inner Mongolia, the average AOD was less than 0.25 for sites 1 to 6 under clear
conditions. We found large variations of 40-50% in the same area in the 1-hour
measurements collected from sites 6a and 6b; these variations were correlated with
regional biomass burning processes. However, AODs exceeded 0.3 at sites 9 and 12,
which were significantly influenced by anthropogenic air pollution from industrial

areas across northern China. MODIS active fires are often spatially distributed over
northeastern China and mainly result from human activities during colder seasons.

### 3.2    Sampling statistics

$C_{BC}^{est}$, $C_{BC}^{max}$, $C_{BC}^{equiv}$, $f_{nonBC}^{est}$, Å$_{tot}$ and snow parameters, such as snow depth, snow
density, snow grain radius, and snow temperature, are given in Table 1 and Figure 3 for

each snow layer following Wang et al. (2013a). Average $C_{BC}^{est}$, $C_{BC}^{max}$, and $C_{BC}^{equiv}$
values were calculated using "left" and "right" samples for each layer. The snow cover
was thin and patchy in Inner Mongolia, and the average snow depth was less than 10
cm from sites 90, 91, 93, and 94, which slightly higher values near the northern border
of China, ranging from 13 to 20 cm at sites 95-97. The maximum snow depth of 46 cm



at site 102 was found in a forest near the Changbai Mountains, averaging 27 cm and

varying from 13 to 46 cm at sites 98 to 102. Because less snow fell during the 2014

snow survey period, the surface snow grain radius varied considerably from 0.07 to 1.3

mm. The snow grain size increased with the snow depth from the surface to the bottom

and was larger than that recorded in previous studies because of snow melting by solar

radiation and the ILAPs (Hadley and Kirchstetter, 2012; Motoyoshi et al., 2005; Painter

et al., 2013; Pedersen et al., 2015). The snow density exhibited little geographical

variation across northern China at 0.13 to 0.38 g cm$^{-3}$. High snow densities result from

melting or snow aging. Similar snow densities have been found in the Xinjiang region

in northern China (Ye et al., 2012). $Å_{tot}$ ranged from 2.1 to 4.8. A higher $Å_{tot}$ is a good

indicator of soil dust, which is primarily driven by the composition of mineral or soil

dust. In contrast, a lower $Å_{tot}$ of 0.8-1.2 indicates that light-absorbing particulates in the

snow are dominated by BC (Bergstrom et al., 2002; Bond et al., 1999).

To better understand the BC distribution in seasonal snow across northern China, an

interannual comparison of the BC content in the surface and average snow measured

during the 2010 and 2014 Chinese surveys was performed, and the results are shown in

Figure 4 and Table 2. The spatial distributions of BC in the surface and average snow

measured using an ISSW spectrophotometer during the 2014 survey generally ranged

from 53 to 3651 ng g$^{-1}$ and 62 to 1635 ng g$^{-1}$, respectively. These variations were very

similar to those of the previous snow campaign, as shown in Figure 4 (Wang et al.,

2013a), although they were much higher than those in the Xinjiang region of

northwestern China (Ye et al., 2012), along the southern edge of the Tibetan Plateau

(Cong et al., 2015), and in North America (Doherty et al., 2014). In the 2014 Chinese

survey, we only collected one layer of snow samples from central Inner Mongolia, and

the BC mixing ratio was 334 ng g$^{-1}$ in aged snow. Along the northern Chinese border





at sites 91-95, BC contamination in the cleanest snow ranged from 27 to 260 ng g$^{-1}$; only a few values exceeded 200 ng g$^{-1}$. The $f_{nonBC}^{est}$ value varied remarkably from 29 to 78%, although BC was still a major absorber in this region. Heavily polluted sites were located in industrial regions across northeastern China (sites 99-102). The surface

snow BC in this region ranged from 508 to 3651 ng g$^{-1}$, and the highest BC in the sub-surface layer of the four sites was 2882 ng g$^{-1}$. In addition, the fraction of total particulate solar absorption due to $f_{nonBC}^{est}$ was typically 35-74%, indicating that light absorption in snow is mainly attributable to OC and AD from human activity.

Figure 5 compares the BC mixing ratio measured via the ISSW method with the

10 calculations. Ideally, the BC content calculated using Equation (7) should be equal to that measured via the ISSW method. The two results agreed very well (R$^2$=0.99), indicating that Equation (7) worked well for this measurement. Thus, the BC content measured via the ISSW method was found to be reasonable and reliable. To compare with the mixing ratio of OC calculated from Equation (7), we used the calculated BC

mixing ratios listed in Figures 6-8 and Table 3 in the following sections.

### 3.3 Emission factors

Typically, chemical components in seasonal snow originate from different emission sources. For example, OC and BC are emitted from the incomplete combustion of fossil fuels and biofuels. K$^+$ is a good tracer of biomass burning (Cachier et al., 1995), whereas

NO$_3^-$ and SO$_4^{2-}$ originate primarily from diesel oil and gasoline combustion and from coal burning with sulfur. NH$_4^+$ is an important indicator of fertilizer used in agricultural processes. OC/BC ratios are used to represent possible emission sources of biomass burning (Bond et al., 2013; Cao et al., 2007; Cong et al., 2015; Novakov et al., 2005). The OC/BC ratio in the sampled surface snow ranged from 1.4 to 17.6 (Figure 6); a





very high OC/BC ratio (17.6) was found at site 90, suggesting that biomass burning may have been a major contributor of OC through photochemical reactions during the 2014 Chinese survey. A relatively high correlation ($R^2$=0.87, n=13) was observed between OC and BC, indicating similar emission sources at all sampling sites except

for those in central Inner Mongolia, as shown in Figure 6a. These results are consistent with those of previous studies (Ming et al., 2010). The strong correlations between $NH_4^+/SO_4^{2-}$ and $NH_4^+/NO_3^-$ shown in Figures 6b and 6c ($R^2$=0.91 and $R^2$=0.94, respectively; n=12) suggest that fine particles characterized as $(NH_4)_2SO_4$ and $NH_4NO_3$ were derived from more intense agricultural and human activities occurring near

farmland areas (Ianniello et al., 2011). It is widely accepted that crustal Al originates from mineral or soil dust (Wedepohl, 1995). Therefore, the weak correlation between $K^+$ and Al could be explained by different emission sources of $K^+$ and Al (Figure 6d) often attributed to biomass burning and mineral or soil dust, respectively, in local or remote regions. Sampling site 101 (red dot in Figure 6d) is positioned very close to a

village. As a result, we found a much higher $K^+$ than Al value owing to high biomass burning via human activity in winter.

### 3.4    Mass contributions of chemical components

The land-cover types (Figure 7) were obtained from the Collection 5.1 MODIS global land-cover type product (MCD12C1) at a 0.05° spatial resolution and included 17

different surface vegetation types (Friedl et al., 2010; Loveland and Belward, 1997). The sampling areas were located in grasslands, croplands, and urban and built-up regions across northern China that are likely influenced by human activity (Huang et al., 2015a). According to Table 3 and Figure 7, the $NH_4^+$ concentrations emitted from agricultural sources at all sites accounted for less than 2.8% because the sites were



positioned 50 km from cities. However, large fractions of both $SO_4^{2-}$ and $NO_3^-$ were observed, varying from 14.8 to 42.8% at all sites, with the highest fraction of 24.2-42.8% found in industrial areas. These results show that $SO_4^{2-}$ and $NO_3^-$ made the greatest contributions to the total chemical concentration in the surface snow and are

significant anthropogenic sources of fossil-fuel combustion in heavy industrial areas. More specifically, the largest AD contribution ranged from 35.3 to 46% at sites 91 to 95 owing to strong winds during winter; the AD fractions were only 5.7 to 31% at the other sites. Fractions of BC and OC were similar to those above, showing that biomass burning was a major source during winter in the sampling region. Zhang et al. (2013)

showed that OC and BC fractions vary more widely in the winter than other seasons owing to industrial activity in China. Sulfate peaks were found in summer (15.4%), whereas nitrate peaks were observed in spring (11.1%). Potassium ($K^+_{biosmoke}$) was found to be a good tracer of biomass burning, ranging from 1.3 to 5.1% along the northern Chinese border compared to lower values found at lower latitudes (1.5-2.3%),

and it exhibited much higher contributions in Inner Mongolia and along the northern Chinese border owing to increased emissions from cooking, open fires, and agricultural activities. The fraction of sea salt aerosol was found to range from 6.3 to 20.9%. Wang et al. (2015) showed that higher $Cl^-/Na^+$ ratios in seasonal snow found in the 2014 Chinese survey were 1 - 2 times greater than those of seawater, implying that they

constituted a significant source of anthropogenic $Cl^-$ (Wang et al., 2015).

### 3.5    BC, OC and AD contributions to light absorption

As described by Wang et al. (2013a), light absorption by ILAPs can be determined from ISSW measurements combined with a chemical analysis of iron concentrations by assuming that the light absorption of dust is dominated by iron (Fe); however, iron can



also originate from industrial emissions. Doherty et al. (2014) used a similar method to distinguish between contributions of light-absorbing impurities in snow in central North America. Although heavy AD loading was observed in the study region, the fraction of light absorption due to AD (assumed to exist as goethite) was generally less than 10%

across northeastern China (Figure 8), which was much smaller than that observed in the Qilian Mountains (e.g., Figure 11 of Wang et al., 2013a). Light absorption was mainly dominated by BC and OC in snow in January 2014. By contrast, the fraction of light absorption due to BC varied from 48.3 to 88.3% at all sites, with only one site dominated by OC (site 90 in central Inner Mongolia). Compared to the light absorption

patterns in the Qilian Mountains, iron played a less significant role in particulate light absorption in snow across the northeastern China sampling area.

### 3.6    Comparisons between the observed and modeled snow albedo

Snow albedo reduction due to BC has been examined in previous studies (Brandt et al., 2011; Hadley and Kirchstetter, 2012; Yasunari et al., 2010). However, few observations

of the BC, OC, and AD mixing effects on snow albedo reduction in seasonal snow at middle latitudes exist. In this study, we measured the snow albedo at six sites under clear conditions. A comparison of the snow albedos derived from the SNICAR and SAMDS models is presented in Figure 9. Spectral snow albedos measured in our experiments and simulated through the SNICAR and SAMDS models are shown in

Figure 11. We ran the models at a solar zenith angle of 60 °C, which is consistent with our experimental method for measuring albedo and with light-absorbing impurities across northern China. Mixing ratios of BC, OC, and dust were chosen to exhibit the following ranges: 0-5 $\mu g\ g^{-1}$, 0-30 $\mu g\ g^{-1}$, and 0-6 $\mu g\ g^{-1}$, respectively, encompassing the values measured in snow surfaces across northern China in previous research



(Warren and Wiscombe, 1980) and in this study. The BC MAC used in this study was 7.5 m$^2$ g$^{-1}$ at 550 nm, which was assumed in the most recent climate assessment and is appropriate for freshly emitted BC (Warren, 1982; Bond and Bergstrom, 2006; Bond et al., 2013). The visible and near-infrared albedos of underlying ground surfaces were

0.2 and 0.4, respectively, according to the MODIS data. The spectral albedos of pure snow derived from the SNICAR (dash line) and SAMDS (solid line) models agreed well. However, there was a slight tendency for the SNICAR model values to become lower than SAMDS model values when BC and dust mixing ratios range from 1 to 5 μg g$^{-1}$ and 4 to 6 μg g$^{-1}$, respectively. The 1.5-2.5% deviation between the SNICAR and

SAMDS modeled snow albedos at 550 nm for higher BC mixing ratios indicates that albedo reduction by light-absorbing impurities in the SNICAR model was greater than that of the SAMDS model. More notably, snow albedos decreased significantly within the UV-visible wavelength, especially for the higher OC (dotted lines) contents in Figure 9. This may be attributed to the fact that OC strongly absorbs UV-visible

radiation and masks BC absorption for high AAE of OC, which decreases remarkably with increasing wavelengths (Warren and Wiscombe, 1980).

As is shown in Figure 10, we also estimated the reduction in the spectrally weighted snow albedo for different R$_{eff}$ values using the SNICAR and SAMDS models. Higher degrees of snow albedo reduction by both BC and dust-contaminated snow were

generally found for larger snow grains (Figure 10a-b). For example, snow albedo reduction attributable to 1000 ng g$^{-1}$, 1 μg g$^{-1}$, and 10 μg g$^{-1}$ for BC, dust, and OC, respectively, was 37%, 41%, and 38% greater in 200 μm snow (0.081, 0.0019, and 0.047) than that in 100 μm snow (0.059, 0.0013, and 0.034). Both the SNICAR and SAMDS models indicated that the snow albedo is more sensitive to BC, especially

during lower ILAP periods. For example, 200 ng g$^{-1}$ of BC decreased the snow albedo



by 3.4% for an optical effective radius of 200 µm, which is much larger than the snow

albedo reduction of 2 µg g$^{-1}$ for dust and OC at an optical effective radius 200 µm. As

Hadley and Kirchstetter (2012) noted, compared with pure 55 µm snow, 300 ng g$^{-1}$ of

BC contamination and growth to 110 µm causes a net albedo reduction of 0.11 (from

0.82 to 0.71), causing snow to absorb 61% more solar energy (Hadley and Kirchstetter,

2012). Previous studies have also indicated that the mixing ratio of BC (10-100 ng g$^{-1}$)

in snow may decrease its albedo by 1-5% depending on its aging process (Warren and

Wiscombe, 1980; Hansen and Nazarenko, 2004).

Figure 11 compares snow albedo values under clear sky conditions collected through

surface measurements and the SNICAR and SAMDS models based on Toon et al.'s

(1989) two-stream radiative transfer solution. The model input parameters are listed in

Tables 1 and 4. The MAC of BC used in the ISSW was 6.3 m$^2$ g$^{-1}$ at 550 nm, although

a value of 7.5 was used in the NICAR and SAMDS models. Thus, the mixing ratio of

BC was corrected by dividing it by 1.19 when BC was used as the input parameter to

the snow albedo models (Wang et al., 2013a). The snow albedos measured at 550 nm

varied considerably from 0.99 to 0.61 owing to different mixing ratios of ILAPs and

snow parameters, such as snow grain size. The snow albedos predicted by the SNICAR

and SAMDS models agreed well at each site based on the same input parameters. The

snow albedos of the SNICAR and SAMDS models retrieved from measured snow grain

sizes complemented the surface measurements for lower mixing ratios of BC, OC, and

AD. The highest BC mixing ratios were 1461 and 3651 ng g$^{-1}$ at sites 98 and 101,

respectively, across industrial regions, with median values found in integrated layers of

264 and 1635 ng g$^{-1}$, respectively. The OC and dust mixing ratios were as high as 13.3

µg g$^{-1}$ and 32 µg g$^{-1}$ in this region. The higher AD mixing ratios are consistent with

25   previous studies conducted by Zhang et al. (2013) and Huang et al. (2015a), who



indicated that AD is highly correlated with anthropogenic air pollution originating from human activity across northeastern China. Thus, we found a larger difference in snow albedo of up to 0.2 at higher BC, OC, and AD contents between the surface measurements and the modeled albedos for the measured snow grain sizes at sites 91,

98, and 101. When the reduction in albedo caused by light-absorbing impurities at the inferred wavelength by measured snow grain size was not accounted for, we also calculated the snow albedos from the SNICAR and SAMDS models using the optical effective size shown in Figure 11 (blue and red shaded bands), and these values were consistent with the surface measurements (gray shaded bands), especially at near-

infrared wavelengths. This may be attributed to the fact that the optical effective size at these three sampling sites was much larger than the measured grain size, indicating that the radiative perturbation of light-absorbing impurities was amplified with snow grain optical effective size. As indicated in Figure 11, BC, OC, and AD are three types of ILAPs found in snow that can reduce spectral snow albedo levels. Reduced snow albedo

due to ILAPs in our measurements were generally comparable to the modeled effects found in the commonly used SNICAR and SAMDS models (Flanner et al., 2007; Zhang et al., 2016). Therefore, 100-500 ng g$^{-1}$ of BC can lower the snow albedo by 2-6% relative to pure snow with a snow grain size of 200 μm according to our snow field campaign, and AD was found to be a weak absorber owing to its lower AD MAC,

supporting previous observations made by Warren and Wiscombe (1980). The OC MAC was also lower and comparable to that of the AD. A clear decreasing trend in the surface snow albedo owing to the high ambient mixing ratios of OC from Inner Mongolia to northeastern China was found. The radiative transfer modeling results presented by Zhang et al. (2016) and measurement results of this study show that the

spectral albedo of snow reduction caused OC levels (above 20 μg g$^{-1}$) to increase by a



factor of 3 for a snow grain size of 800 μm compared to 100 μm.

## 4    Discussion and conclusions

In this study, a Chinese survey was performed in January 2014. We collected 92 snow samples from 13 sites across northern China. Much less snow had fallen than in previous years; as a result, most of the surface snow samples were collected as aged snow, and snow grain sizes were much larger owing to solar radiation absorbed by ILAPs as a result of settlement processes. Although we selected study locations in remote regions located at least 50 km from cities, higher AODs measured using a sun photometer and remote sensing devices showed that heavily polluted areas remain in industrial regions across northern China. The estimated BC mixing ratios measured through the 2014 survey via the ISSW spectrophotometer in surface and average integrated snow of 53 to 3651 and 62 to 1635 ng g$^{-1}$, respectively, were much larger than those of previous snow field campaigns. The non-BC fraction showed that most of the ILAPs in seasonal snow were dominated by OC and AD. Owing to BC and OC mass absorption efficiencies of 6.3 and 0.3 m$^2$ g$^{-1}$, respectively, at 550 nm, light absorption was still dominated by BC and OC in seasonal snow during the entire campaign. AD contributions in snow to light absorption amounted to less than 10%. The large OC/BC ratios and correlation coefficients indicated that these contributions were mainly derived from common sources (e.g., biomass burning). Similarly, $NH_4^+$ was attributed to intense agricultural activity compared to industrial emissions of $SO_4^{2-}$ and $NO_3^-$.

The fraction of the AD mixing ratios was larger at high latitudes than at low latitudes owing to strong winds transporting snow. OC emitted from biomass burning and $SO_4^{2-}$ and $NO_3^-$ generated from fossil fuels and biofuels also played key roles in the mixing



ratios of chemical components in seasonal snow. Finally, a comparison between measured and simulated snow albedos was conducted. The snow albedos measured from a spectroradiometer and simulated using the SNICAR and SAMDS models agreed with the lower mixing ratios of BC, OC, and AD. Based on the snow grain optical

5   effective size, we found good agreement between surface measurements and radiative transfer models. Therefore, the snow grain optical effective size is preferred when measurements on snow grain size are available at the near-infrared wavelength. Although the MAC of OC was much lower than that of BC, we found that OC was a major absorber in snow owing to its high mixing ratio of OC from human activities

10   occurring across northeastern China. Moreover, 5000 ng g$^{-1}$ of OC was found to reduce the snow albedo by 1.5%-5% depending on the snow grain size and aging period. Therefore, we suggest that the mixing ratio of OC be added as an input parameter to the SNICAR model for determining snow albedos.



***Acknowledgements.*** This research was supported by the Foundation for Innovative Research Groups of the National Science Foundation of China (41521004), the National Science Foundation of China under Grants 41522505, and the Fundamental Research Funds for the Central Universities (lzujbky-2015-k01, lzujbky-2016-k06 and

5   lzujbky-2015-3). The MODIS data were obtained from the NASA Earth Observing System Data and Information System, Land Processes Distributed Active Archive Center (LP DAAC) at the USGS Earth Resources Observation and Science (EROS) Center.



**Table 1.** Statistics on seasonal snow variables measured using an ISSW in the study sites.

| Site | Layer | Latitude N | Longitude E | Site average snow depth (cm) | Sample depth (cm) Top | Bottom | Temperature (°C) | Snow density (g cm⁻³) | Snow grain radius (mm) | $C_{BC}^{equiv}$ (ng g⁻¹) | $C_{BC}^{max}$ (ng g⁻¹) | $C_{BC}^{max}$ (ng g⁻¹) | $f_{nonBC}^{est}$ (%) | $\mathring{A}_{aot}$ 450:600 nm |
|---|---|---|---|---|---|---|---|---|---|---|---|---|---|---|
| 90 | 1 | 45°02'44" | 116°22'45" | 3 | 0 | 5 | -14 | 0.38 | 0.15 | 862 | 530 | 334 (–, 473) | 61 (45, 110) | 3.8 |
| 91 | 1 | 50°02'48" | 124°22'41" | 8 | 0 | 5 | -25 | 0.23 | 0.08 | 250 | 200 | 161 (110, 200) | 35 (18, 55) | 2.1 |
|  | 2 |  |  |  | 5 | 10 | -25 | 0.18 | 0.15 | – | – | – | – | – |
| 92 | 1 | 50°39'07" | 122°23'53" | 14 | 0 | 5 | -16 | 0.17 | 0.08 | 80 | 72 | 53 (32, 68) | 34 (15, 60) | 2.3 |
|  | 2 |  |  |  | 5 | 10 | -16 | 0.18 | 1.25 | 108 | 97 | 77 (49, 97) | 29 (11, 55) | 2.2 |
| 93 | 1 | 50°24'50" | 124°54'20" | 8 | 0 | 1 | – | – | 0.07 | 172 | 97 | 70 (39, 90) | 53 (39, 73) | 2.5 |
|  | 2 |  |  |  | 2 | 6 | -21 | 0.21 | 0.175 | 313 | 136 | 100 (46, 128) | 68 (59, 85) | 2.7 |
| 94 | 1 | 50°09'05" | 125°46'06" | 8 | 0 | 1 | -24 | – | 0.1 | 1173 | 404 | 260 (108, 363) | 78 (69, 91) | 2.8 |
| 95 | 1 | 50°54'43" | 127°04'50" | 18 | 0 | 3 | -22 | 0.23 | 0.175 | 278 | 110 | 75 (37, 100) | 73 (64, 86) | 2.6 |
|  | 2 |  |  |  | 3 | 8 | -23 | 0.18 | 0.9 | 230 | 83 | 54 (25, 73) | 77 (68, 89) | 2.7 |
|  | 3 |  |  |  | 8 | 13 | -25 | 0.27 | 1 | 91 | 50 | 27 (7, 40) | 71 (55, 92) | 3.2 |
|  | 4 |  |  |  | 13 | 18 | -25 | 0.14 | 1.3 | 521 | 308 | 131 (–, 237) | 75 (54, 148) | 4.8 |
| 96 | 1 | 49°14'21" | 129°43'13" | 13 | 0 | 5 | -22 | 0.37 | 0.08 | 1340 | 385 | 241 (21, 332) | 83 (76, 99) | 3.4 |
|  | 2 |  |  |  | 5 | 10 | -23 | 0.25 | 0.6 | 234 | 85 | 45 (6, 71) | 81 (69, 98) | 3.3 |
|  | 3 |  |  |  | 10 | 15 | -22 | 0.22 | 1 | 371 | 126 | 78 (26, 111) | 80 (71, 93) | 3.0 |
| 97 | 1 | 47°39'18" | 131°13'10" | 20 | 0 | 5 | -22 | 0.36 | 0.2 | 1005 | 362 | 236 (93, 306) | 76 (69, 91) | 2.8 |
|  | 2 |  |  |  | 5 | 10 | -18 | 0.27 | 1.1 | 160 | 79 | 45 (15, 65) | 71 (57, 91) | 3.0 |
|  | 3 |  |  |  | 10 | 15 | -14 | 0.32 | 0.9 | 32 | 23 | 9 (3, 18) | 74 (44, 90) | 3.1 |
| 98 | 1 | 45°25'38" | 130°58'55" | 35 | 0 | 5 | -20 | 0.22 | 0.09 | 2154 | 2080 | 1461 (754, 1818) | 32 (16, 65) | 2.5 |
|  | 2 |  |  |  | 5 | 10 | -23 | 0.24 | 0.5 | 641 | 349 | 241 (123, 307) | 58 (46, 79) | 2.6 |


| Site | Layer | Latitude N | Longitude E | Site average snow depth (cm) | Sample depth (cm) Top | Bottom | Temperature (°C) | Snow density (g cm⁻³) | Snow grain radius (mm) | $C_{BC}^{equiv}$ (ng g⁻¹) | $C_{BC}^{max}$ (ng g⁻¹) | $C_{BC}^{max}$ (ng g⁻¹) | $f_{nonBC}^{est}$ (%) | $\mathring{A}_{abs}$ 450:600 nm |
|---|---|---|---|---|---|---|---|---|---|---|---|---|---|---|
|  | 3 |  |  |  | 10 | 15 | -15 | 0.24 | 0.75 | 97 | 55 | 25 (10, 45) | 74 (52, 90) | 3.1 |
|  | 4 |  |  |  | 15 | 20 | -14 | 0.26 | 0.75 | 61 | 39 | 16 (7, 31) | 75 (48, 89) | 3.1 |
|  | 5 |  |  |  | 20 | 25 | -13 | 0.28 | 1.1 | 432 | 261 | 166 (62, 221) | 62 (49, 86) | 2.8 |
|  | 6 |  |  |  | 25 | 30 | -11 | 0.29 | 1.3 | 348 | 165 | 112 (66, 148) | 68 (58, 81) | 2.5 |
|  | 7 |  |  |  | 30 | 35 | – | 0.33 | 1.3 | 288 | 118 | 72 (37, 101) | 75 (65, 87) | 2.6 |
| 99 | 1 | 43°36'10" | 125°42'04" | 13 | 0 | 5 | -2 | 0.18 | 0.085 | 1459 | 922 | 636 (235, 823) | 54 (40, 84) | 2.8 |
|  | 2 |  |  |  | 5 | 12 | -3 | 0.23 | 1.1 | 5847 | 3778 | 2882 (1549, 3559) | 51 (39, 74) | 2.5 |
|  | 3 |  |  |  | 12 | 18 | -4 | 0.28 | 1.1 | 2795 | 1360 | 1059 (630, 1306) | 62 (53, 77) | 2.3 |
|  | 4 |  |  |  | 18 | 24 | -4 | 0.28 | 0.7 | 2027 | 1518 | 1105 (537, 1381) | 46 (32, 73) | 2.6 |
| 100 | 1 | 43°30'45" | 127°15'34" | 18 | 0 | 2.5 | – | 0.14 | 0.075 | 1739 | 703 | 508 (250, 651) | 71 (63, 86) | 2.6 |
|  | 2 |  |  |  | 2.5 | 4 | -2 | 0.18 | 0.25 | 4126 | 2891 | 1729 (–, 2371) | 58 (43, 103) | 3.6 |
|  | 3 |  |  |  | 4 | 9 | -2 | 0.23 | 0.9 | 1062 | 564 | 414 (208, 529) | 60 (49, 80) | 2.6 |
|  | 4 |  |  |  | 9 | 15 | -3 | 0.24 | 0.8 | 1601 | 728 | 541 (266, 691) | 66 (57, 83) | 2.6 |
| 101 | 1 | 43°47'25" | 125°46'08" | 22 | 0 | 5 | -19 | 0.24 | 0.07 | 5634 | 5070 | 3651 (1333, 4644) | 35 (18, 76) | 2.9 |
|  | 2 |  |  |  | 5 | 10 | -17 | 0.24 | 0.9 | 4318 | 3826 | 2575 (882, 3284) | 40 (24, 79) | 2.9 |
|  | 3 |  |  |  | 10 | 15 | -15 | 0.23 | 1.1 | 657 | 389 | 313 (206, 386) | 46 (33, 63) | 2.1 |
|  | 4 |  |  |  | 15 | 20 | -14 | 0.28 | 1.3 | 297 | 254 | 188 (108, 239) | 39 (21, 67) | 2.5 |
| 102 | 1 | 42°12'34" | 126°37'50" | 46 | 0 | 3 | -8 | 0.13 | 0.07 | 2109 | 1734 | 1357 (808, 1755) | 37 (19, 61) | 2.3 |
|  | 2 |  |  |  | 3 | 8 | -8 | 0.26 | 0.1 | 594 | 377 | 279 (149, 371) | 53 (37, 75) | 2.6 |
|  | 3 |  |  |  | 8 | 13 | -8 | 0.24 | 0.3 | 2513 | 2088 | 1623 (425, 2120) | 35 (15, 81) | 3.1 |
|  | 4 |  |  |  | 13 | 18 | -8 | 0.28 | 0.6 | 1922 | 1073 | 773 (229, 1013) | 60 (48, 88) | 3.0 |
|  | 5 |  |  |  | 18 | 23 | -7 | 0.32 | 1 | 1112 | 629 | 450 (197, 590) | 60 (47, 82) | 2.7 |



| Site | Layer | Latitude N | Longitude E | Site average snow depth (cm) | Sample depth (cm) Top | Sample depth (cm) Bottom | Temperature (°C) | Snow density (g cm$^{-3}$) | Snow grain radius (mm) | $C_{BC}^{equiv}$ (ng g$^{-1}$) | $C_{BC}^{max}$ (ng g$^{-1}$) | $C_{BC}^{max}$ (ng g$^{-1}$) | $f_{nonBC}^{est}$ (%) | $\mathring{A}_{tot}$ 450:600 nm |
|---|---|---|---|---|---|---|---|---|---|---|---|---|---|---|
| | 6 | | | | 23 | 28 | -6 | 0.27 | 1.2 | 1466 | 1281 | 903 (300, 1167) | 38 (21, 80) | 2.9 |
| | 7 | | | | 28 | 33 | -5 | 0.26 | 1.2 | 858 | 493 | 344 (133, 457) | 66 (53, 87) | 2.9 |
| | 8 | | | | 33 | 38 | -5 | 0.3 | 1 | 426 | 197 | 109 (27, 163) | 74 (61, 95) | 3.2 |
| | 9 | | | | 38 | 43 | -4 | 0.29 | 0.8 | 524 | 245 | 157 (67, 220) | 71 (58, 87) | 2.8 |





**Table 2.** Estimates of integrated snowpack BC content in seasonal snow in the study sites for 2010 and 2014.

| Site | Date sampled (2014) | Snowpack average BC (ng g$^{-1}$) | Site | Date sampled (2010) | Snowpack average BC (ng g$^{-1}$) |
|---|---|---|---|---|---|
| 90 | 10-Jan | 334 | 1 | 11-Jan | – |
| 91 | 13-Jan | 161 | 2 | 12-Jan | – |
| 92 | 14-Jan | 65 | 3 | 12-Jan | 343 |
| 93 | 15-Jan | 94 | 4 | 13-Jan | 864 |
| 94 | 15-Jan | 260 | 5 | 13-Jan | 801 |
| 95 | 16-Jan | 62 | 6 | 13-Jan | 655 |
| 96 | 17-Jan | 140 | 7 | 15-Jan | 290 |
| 97 | 18-Jan | 105 | 8 | 15-Jan | 82 |
| 98 | 19-Jan | 264 | 9 | 16-Jan | 113 |
| 99 | 23-Jan | 1507 | 10 | 16-Jan | 98 |
| 100 | 24-Jan | 592 | 11 | 18-Jan | 159 |
| 101 | 26-Jan | 1635 | 12 | 19-Jan | 641 |
| 102 | 27-Jan | 583 | 13 | 19-Jan | 413 |
| | | | 14 | 20-Jan | 1781 |
| | | | 15 | 20-Jan | 213 |
| | | | 16 | 21-Jan | 939 |
| | | | 17 | 22-Jan | 379 |
| | | | 18 | 22-Jan | 424 |
| | | | 19 | 23-Jan | 368 |
| | | | 20 | 24-Jan | 168 |
| | | | 21 | 25-Jan | 89 |
| | | | 22 | 26-Jan | 55 |
| | | | 23 | 26-Jan | 64 |
| | | | 24 | 26-Jan | 148 |
| | | | 25 | 27-Jan | 255 |
| | | | 26 | 27-Jan | 193 |
| | | | 27 | 28-Jan | 179 |
| | | | 28 | 29-Jan | 441 |
| | | | 29 | 29-Jan | 1057 |
| | | | 30 | 31-Jan | 975 |
| | | | 31 | 31-Jan | 749 |
| | | | 32 | 1-Feb | 620 |
| | | | 33 | 2-Feb | 367 |
| | | | 34 | 3-Feb | 1519 |
| | | | 35 | 4-Feb | 1008 |
| | | | 36 | 5-Feb | 997 |
| | | | 37 | 6-Feb | 815 |
| | | | 38 | 6-Feb | 1596 |
| | | | 39 | 7-Feb | 632 |
| | | | 40 | 8-Feb | 1231 |



| Site | Date sampled (2014) | Snowpack average BC (ng g$^{-1}$) | Site | Date sampled (2010) | Snowpack average BC (ng g$^{-1}$) |
|------|---------------------|-----------------------------------|------|---------------------|-----------------------------------|
|      |                     |                                   | 41   | 18-Feb              | –                                 |
|      |                     |                                   | 42   | 18-Feb              | –                                 |
|      |                     |                                   | 43   | 19-Feb              | –                                 |
|      |                     |                                   | 44   | 19-Feb              | –                                 |
|      |                     |                                   | 45   | 19-Feb              | –                                 |
|      |                     |                                   | 46   | 20-Feb              | –                                 |





**Table 3.** Chemical species (ng g$^{-1}$) in surface snow for sites across northeastern China in January 2014.

| Site | AD | BC | OC | $K^+_{biosmoke}$ | $SO_4^{2-}$ | $NO_3^-$ | $NH_4^+$ | Sea salt |
|------|------|------|-------|------|-------|------|-----|------|
| 90 | 1900 | 380 | 6700 | 327 | 1685 | 213 | 22 | 868 |
| 91 | 1700 | 180 | 590 | 179 | 853 | 465 | 36 | 827 |
| 92 | 1300 | 60 | 280 | 150 | 511 | 105 | 19 | 456 |
| 93 | 1700 | 80 | 450 | 213 | 718 | 387 | 90 | 960 |
| 94 | 3300 | 300 | 2700 | 118 | 1335 | 550 | 28 | 554 |
| 95 | 2000 | 90 | 600 | 164 | 587 | 523 | 39 | 669 |
| 96 | 2300 | 280 | 3900 | 309 | 1285 | 493 | 91 | 1227 |
| 97 | 2400 | 280 | 2400 | 173 | 1163 | 407 | 38 | 753 |
| 98 | 3900 | 1600 | 13300 | 633 | 3096 | 747 | 195 | 2516 |
| 99 | 3000 | 770 | 4700 | 372 | 3379 | 1492 | 155 | 2310 |
| 100 | 3800 | 570 | 4000 | 260 | 4237 | 2258 | 487 | 2195 |
| 101 | 3500 | 4200 | 32000 | 1337 | 12382 | 2364 | – | 5131 |
| 102 | 5800 | 1700 | 2400 | 488 | 8034 | 3631 | 769 | 4420 |

**Table 4.** Measured and calculated snow grain sizes for sites in northeastern China in January 2014.

| Site | Measured Snow grain radius (µm) | Calculated R$_{eff}$ using SAMDS (µm) | Calculated R$_{eff}$ using SNICAR (µm) |
|------|------|------|------|
| 90 | 150 | 203-236 | 168-197 |
| 91 | 80 | 178-220 | 148-183 |
| 93 | 70 | 73-80 | 60-65 |
| 95 | 175 | 148-188 | 123-156 |
| 98 | 90 | 248-302 | 209-259 |
| 101 | 70 | 362-446 | 312-390 |





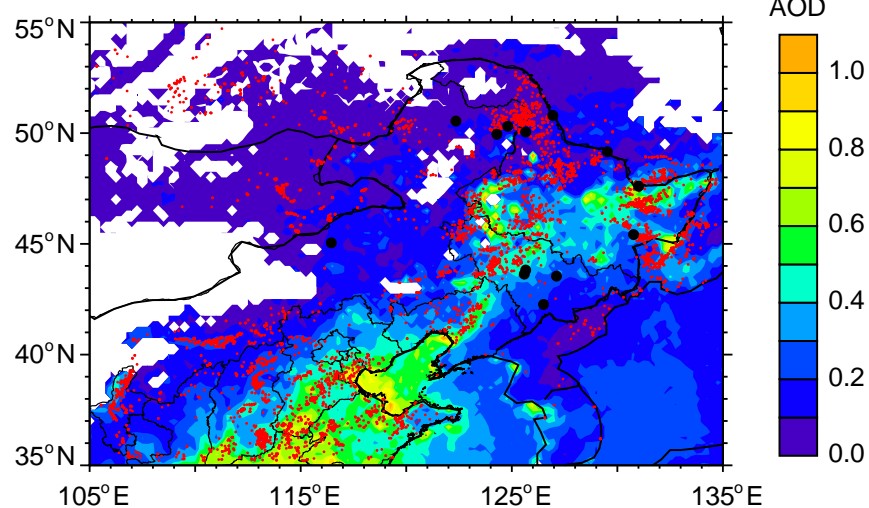

**Fig. 1.** Spatial distribution of the averaged AOD retrieved from Aqua-MODIS over northern China from October to January; the red regions are MODIS active fire locations; the black dots are the sampling locations.





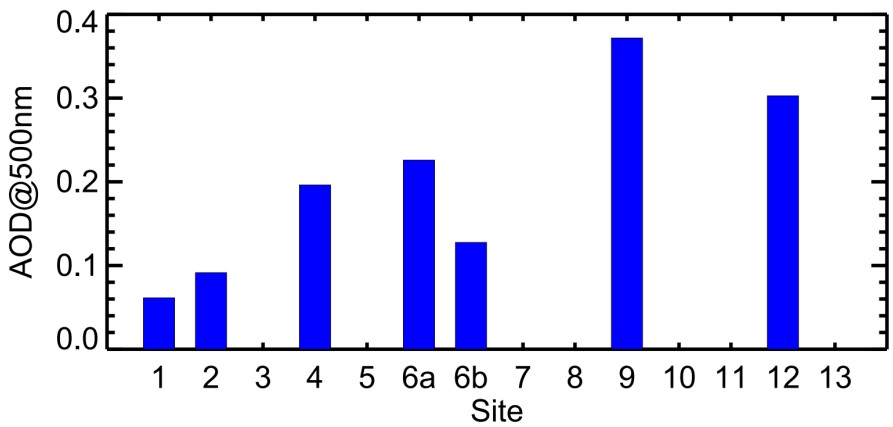

**Fig. 2.** The variation in AOD at different sites measured using a Microtops Ⅱ Sun photometer over northern China in January 2014.





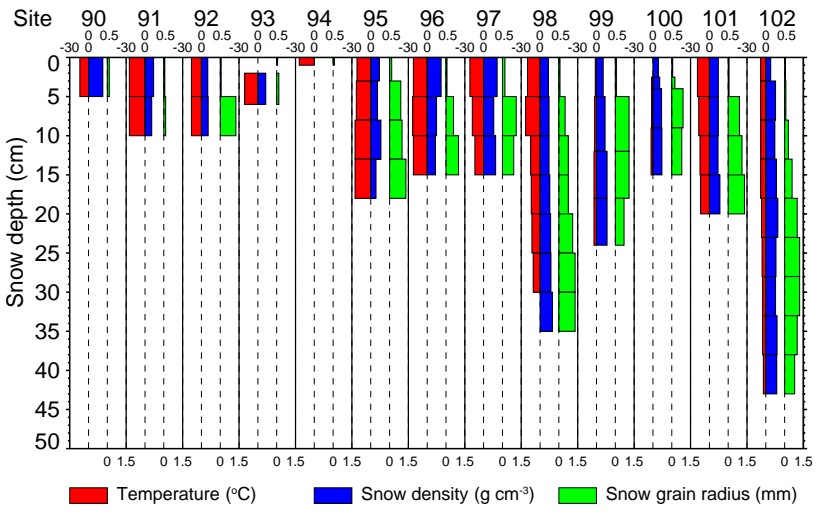

**Fig. 3.** Vertical temperature, snow density, and snow grain radius profiles at each site during the 2014 Chinese snow survey.



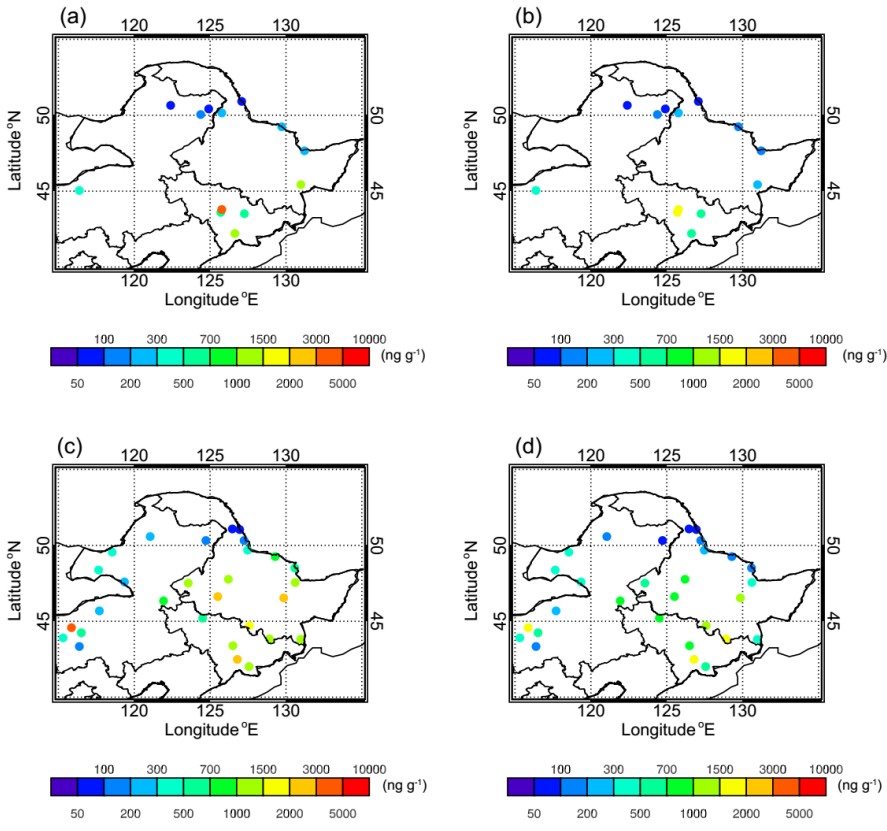

**Fig. 4.** Surface (a, c) and averaged and integrated (b, d) BC content in seasonal snow in 2014 and 2010 across northern China.





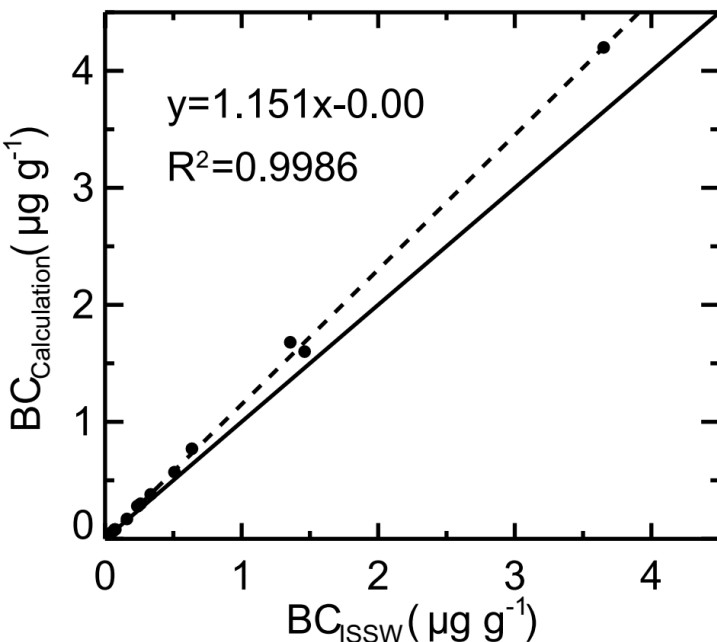

**Fig. 5.** Comparisons between the calculated and optically measured BC contents in surface snow in January 2014.





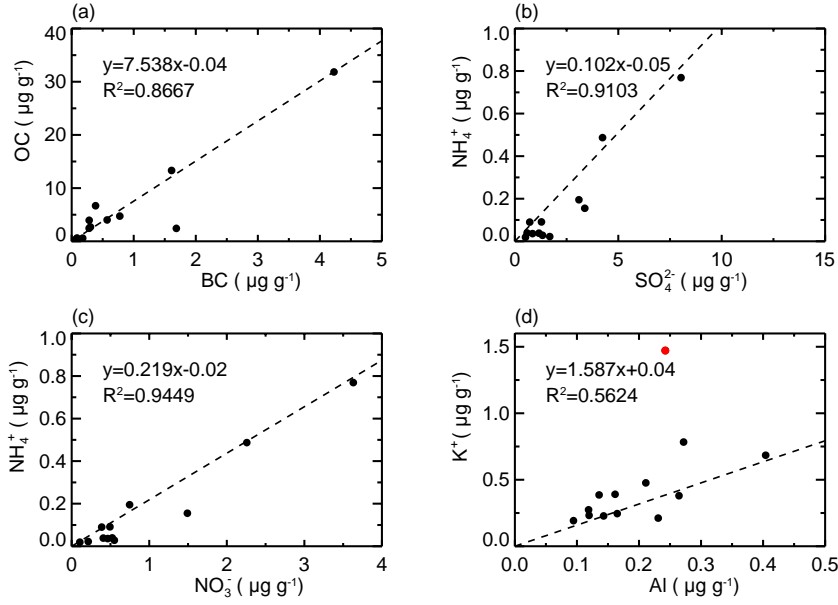

**Fig. 6.** Ratios of OC and BC, $NH_4^+$ and $SO_4^{2-}$, $NH_4^+$ and $NO_3^-$, and $K^+$ and Al in surface snow in January 2014.





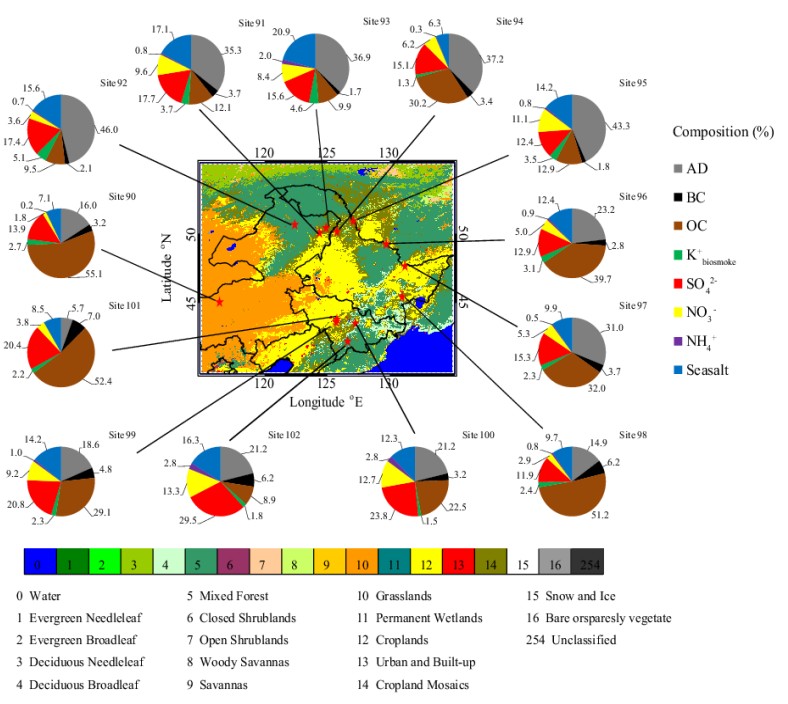

**Fig. 7.** The major components include AD, BC, OC, biomass smoke potassium, secondary aerosol ions (sulfate, nitrate, and ammonium), and sea salt in the surface snow samples collected in January 2014.





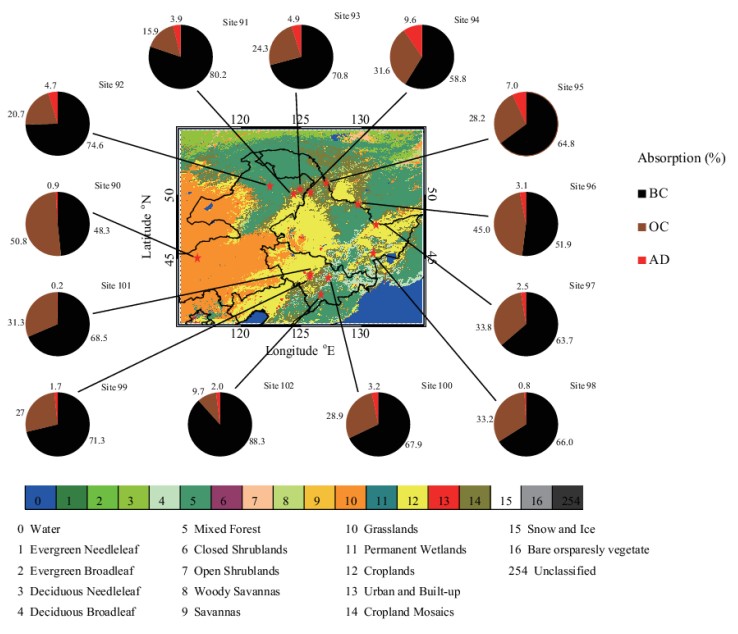

**Fig. 8.** The light absorption of ILAPs in surface snow in January 2014.





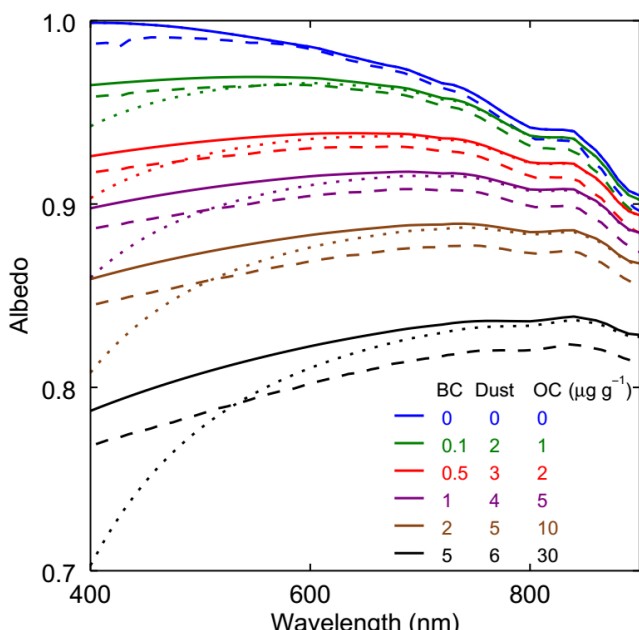

**Fig. 9.** Spectral albedo of snow with different contaminants for a 60° solar zenith angle and a 70 μm snow grain radius. (Solid and dashed lines show the SAMDS and SNICAR model predictions of BC and mineral dust. Dotted lines show the SAMDS model predictions for all ILAPs, including BC, mineral dust, and OC).



**Fig. 10.** Spectrally weighted snow albedo reduction over the 400–1400 nm solar spectrum attributed to (a) BC, (b) mineral dust, and (c) OC computed as the albedo of pure snow minus the albedo of contaminated snow for a 60° solar zenith angle. (Solid and dashed lines show the SAMDS and SNICAR models predictions. The MAC values of BC, Fe, and OC were assumed to be 7.5 m$^2$ g$^{-1}$, 0.9 m$^2$ g$^{-1}$, and 0.3 m$^2$ g$^{-1}$ at 550 nm, respectively).





**Fig. 11.** Measured and modeled spectral albedos of snow at sites (a) 90, (b) 91, (c) 93, (d) 95, (e) 98, and (f) 101. (Gray shaded bands correspond to measured spectral albedos, red and blue solid lines correspond to spectral albedos from the SAMDS and SNICAR models with measured snow grain radii, and red and blue shaded bands correspond to the albedos from the SAMDS and SNICAR models with calculated snow grain optical effective radii ($R_{eff}$). Contaminants only include BC and mineral dust in the SAMDS and SNICAR models. In the SNICAR model, the ratio of Fe in dust was found to be 2.8%. Dashed red lines are similar to solid red lines, although OC should be added to the list of contaminants in the model.





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
