# Peer review of "Observations and model simulations of snow albedo reduction in seasonal snow due to insoluble light-absorbing particles during 2014 Chinese survey"

_Atmospheric Chemistry and Physics, 2016_

## Referee Comment (RC1) · Anonymous Referee #1 · 26 Oct 2016

This paper uses snow sample observations across northern China in January 2014 and aerosol radiation models to examine the reduction of snow albedo due to black carbon, organic carbon and anthropogenic dust. The study suggests different contributions to the snow albedo reductions from these aerosols and suggests that bionmass burning may be a major contributor at most snow sampling sites. It also evaluates the model simulations based on the observations. In general, this paper provides useful information, particularly the different performance in snow ableldo reduction by three types of aerosols. However, there are still a few limitations. The small sample volume observed over short period at limited sites could make the results questionable. At least discussions about the uncertainties and potential issues in this study are neces-

sary. Corresponding to the limited data, reliable or strong quantitative conclusion is hard to obtain. The authors could provide more quantitative results and discuss the reliability of the findings in the study.

Comments:

(1)Are the snow samples fresh or aged snow?

(2)Snow albedo reduction due to these kinds of aerosols is well known, more quantitative results are needed and more valuable, which could be summarized in the abstract. At current version, it is a little hard for me to summarize the findings I can learn from this paper.

(3)Page 3, line 11-14, why is Ginoux et al. 2010 cited two times in one sentence (begin and end). This is repeated description and one should be deleted.

(4)Page 4, line 4-5, you use "larger" and "more intense" in the sentence, but I did not see any comparison descriptions around this sentence. What you are comparing?

(5)Page 4, Line 12-13, why is Light et al. 1998 cited two times in one sentence (begin and end). This is repeated description and one should be deleted.

(6)Page 5, line 15-24, you mentioned several campaigns for snow collection. What are the differences or similarities for the findings among them? Also, it shows that there is a snow campaign over the examined region in 2010 carried out by Huang et al. 2011. Why do not you also include the observation from this campaign so that you have enough data samples and you can also compare the differences/similarities in two winters?

(7)Page 7, line 1-3, is this data criteria enough to prevent contamination? And why do you include site 101 if it does not fit your data criteria?

(8)Page 7, line 5-10, what is the uncertainty introduced due to your visual inspection and data processing method?

(9)Page 8, line 1-9, what do some variables (not all) stand for?, such as Ss?

(10)Page 8, line 17, why do not you introduce Fe when iron is first used in paper?

(11)Page 9, line 4-6, How do you know Microtops II is reliable, or more reliable than CE318?

(12)Page 9, Line 22, do you mean "in 2014"?

(13)Page 10, line 4-5 and line 19-20, why do the observations and models calculate the albedo at different height (1 m above snow vs at surface)? Also, the downwelling solar radiation in the real sky includes diffuse radiation. How does the model consider the diffuse radiation (such as the contribution from aerosols and clouds in the sky)?

(14)Page 11, line 15, the symbol should be $\mu$, please explain its meaning.

(15)Page 15, line 7-10, both AOD from MODIS and ground are retrieved, please describe clearly.

(16)Page 15, line 16, colder -> cold

(17)Page 15, line 23, This sentence is not complete. I believe what you want to say is "..., which with ..."

(18)Page 16, line 2-4, "Because less snow fell during the 2014 snow survey period, the surface snow grain radius varied considerably from 0.07 to 1.3 mm.". First, I do not understand this causal relationship, please clarify. Second, what does it compare when using 'less', the same period of other years or other locations?

(19)Page 16, line 19-24, why the variations (BC and snow spatial distribution) you found were much higher than the findings from other studies?

(20)Page 17, line 1-8, I cannot catch the main points you would like to deliver here.

(21)Page 17, Figure 5, the sample volume is too small. How reliable are the relationships found here?

(22)Page 17, line 22-24, I do not understand from Figure 6 how you got this range in OC/BC ratio.

(23)Page 18, line 1-16, it is hard to conclude due to limited data and sites.

(24)Page 18, line 18-22. How did you get the observation regarding land-cover type when the land is covered by snow?

(25)Page 19, Line 17-20, why is Wang et al. 2015 cited two times in one sentence (begin and end). This is repeated description and one should be deleted.

---

## Short Comment (SC1) · 31 Oct 2016

The authors conducted extensive field measurements of BC, OC, and dust concentrations in snow over the northern China and employed the SNICAR and SAMDS model simulations to investigate snow albedo reduction caused by the light-absorbing aerosols. This study provides a valuable observational dataset to improve our understanding in the effects of light-absorbing aerosol deposition on snow albedo reduction. I have a short comment on snow albedo modeling.

The authors mentioned that the SAMDS model considers aerosol-snow mixing state and the irregular morphology of snow grain by using asymptotic radiative transfer theory. However, the authors did not provide enough discussions on the effects of Printer-friendly version

BC/dust/OC-snow mixing state and snow grain shape on snow albedo reduction as well as compare with some recent studies. These factors are not trivial in evaluating aerosol-snow albedo effects, in addition to snow grain size and aerosol concentration. For example, recent studies by Liou et al. (2014) and He et al. (2014) developed and applied a stochastic snow model to study BC/dust-induced snow albedo reduction, which explicitly simulates different aerosol-snow mixing states and snow grain shapes. They found that using a realistic snowflake shape reduces BC-induced snow albedo reduction by 20–40% compared to a spherical snow grain, while multiple internal mixing of BC and snow increases the albedo reduction by 40–60% relative to the external mixing. I would suggest discussing these recent findings, which could be very helpful for people to understand potential uncertainty and improvement in snow albedo modeling.

References:

He, C., Li, Q. B., Liou, K. N., Takano, Y., Gu, Y., Qi, L., Mao, Y. H., and Leung, L. R.: Black carbon radiative forcing over the Tibetan Plateau, Geophys. Res. Lett., 41, 7806–7813, doi:10.1002/2014gl062191, 2014.

Liou, K. N., Takano, Y., He, C., Yang, P., Leung, L. R., Gu, Y., and Lee, W. L.: Stochastic parameterization for light absorption by internally mixed BC/dust in snow grains for application to climate models, J. Geophys. Res.-Atmos., 119, 7616–7632, doi:10.1002/2014jd021665, 2014.

---

## Referee Comment (RC2) · Anonymous Referee #2 · 10 Nov 2016

Reviewer Comments: Wang et al

Overall Comments:

Due to the high amounts of industrialization in China, there is a need for studies of impurities in seasonal snow. This study is on a Chinese snow survey conducted in 2014. However, it is not clear how much of the data presented in this study has already been published. The authors are recommended to clarify, in much more detail, what data has been previously published, and what are novel results from this study and included in this manuscript. For example, it is not clear whether this study presents new snow chemistry data, or if this data has been published in another manuscript, such as Wang 2015. It is also unclear why the 2010 Chinese Snow survey data is included

in the study. Averaging results from the 2010 and 2014 surveys seems inappropriate, especially because the snow conditions/depths appear to vary widely since so many sites did not even have enough snow to resample in 2014. The paper does not appear to be ready for publication in this submission, even with the following revisions. Major restructuring is needed.

The novelty of the paper appears to begin in the model comparisons with the albedo surface observations made in 2014. The reader suggests the authors significantly revise the paper to focus solely on the 2014 observations (unless the reader is confused by the text and the 2010 data has not yet been published?).

Additionally, according to the methods used, the reader asks the authors to report the BC as equivalent black carbon (eBC) rather than black carbon. If there is a reason otherwise to report as BC, please explain.

It appears (though clarification is needed from the authors) that a new model, SAMDS, similar to the SNICAR model, is applied here. However, only a small portion of this very long paper is devoted to this section. A short discussion/conclusion at the end needs further expansion. Then for the authors to focus on the observed vs. modeled albedo, as well as provide more information and justification for applying the SAMDS model more widely. The SAMDS model appears to be the main highlight of the paper. The title and paper could be restructured around the SAMDS model.

The scientific methods used could be explained in a more logical way throughout the methods section of the paper. Also, in the site description, an explanation why the site numbers begin at 90 in 2014 should be given. Also, a map including GPS coordinates and the sampling sites would aid the reader. Additionally, it is stated that the same volume of snow was not collected at all sites. How much volume was filtered at each site? This leads the reader to believe that if the same top volume of snow was not collected at each site, the integrated mass concentration of BC in each snow sample would be different?

In conclusion, much of the intro and re-presenting of previously published data from the 2010 and 2014 Chinese snow surveys could be removed. And/or better support/clarification for the reason for including it could be included.

Suggestions for Revisions:

The abstract is not currently sufficient. The first sentence states that 92 samples were collected in 2014, however, the 92 points appear to be presented as fewer averages later. The abstract should state how many average values are presented in the study and if the 92 samples were collected across X sites, X snowpits, and at X resolution within the snowpit. Additionally, only surface sample average concentrations are presented in the abstract, how many samples were collected at the surface? And how many in snow pits, and integrated pit samples?

First of all, at the end of the introduction, previous data published from the 2010 and 2014 Chinese snow survey must be presented clearly. Then, a paragraph outlining what is specifically novel about this study must then be stated. Throughout the manuscript, when previously published data is presented, a reference must be cited for where that data has been published.

Additionally, is this paper the first presentation of the SAMDS model? This should be more clearly stated in the abstract, as this appears to be the main point/novelty of the paper. In general, the paper could be edited to more clearly explain what new data is presented in the study. There are pervasive run-on sentences; the manuscript should be edited to make the language more concise.

The discussion should be expanded to support a more widespread use of the SAMDS model. Also, stating how the model will be disseminated, would aid readers in applying it to their own research. Is there a plan to do so?

Overall, the paper needs major restructuring and revision to portray a logical flow of information. The main weakness is that it is very unclear what is new data and what

has previously been published from the 2010 and 2014 surveys and this paper should focus only on what is new.

Specific Line Comments:

Comment 1: Jaffee 1999 cited on Page 3 Line 4 is not included in the bibliography.

Comment 2: Page 4 Line 3: Writing edit: ', with India at' => ' and ' 76.1% in India'

Comment 3: Page 4 Line 13: 'Among its main light-absorbing impurities' This sentence introduction does not make sense. Could just remove the intro and start with '1ng g-1 of BC'....

Comment 4: Page 4 Line 14: 'on the albedo' => 'on the albedo of snow and ice'.

Comment 5: Page 4 Line 17: 'owing to' doesn't read well... please alternate words here, as well as multiple other locations in the manuscript. Other suggestions could be, 'due to', etc.

Comment 6: Page 4 Line 20, first word. Include citation(s) for the previous sentence.

Comment 7: Page 4 Line 21: In the Bond 2013 paper, sub-title 0.2.3 Synthesis of black carbon climate forcing terms, number 2: "The best estimate of industrial-era climate forcing of black carbon through all forcing mechanisms is +1.1 W m-2 with 90% uncertainty bounds of +0.17 to +2.1 W m-2." Rather than 90 % confidence interval as listed in this manuscript.

Comment 8: Page 5 Line 9: 'in situ' => 'in situ' and elsewhere in the manuscript.

Comment 9: Page 5 Line 20: 'involve' use different word such as 'present' or 'include'

Comment 10: Page 5 Line 21: Begin new sentence with 'Previously, . . .we analyzed' if some data from the Chinese snow survey in 2014 was already published and include citation. The reader is confused if some of this data has already been published, or if this data is all unpublished and included in this study. If all the data is unpublished, could

start the sentence with, 'Here' instead of 'previously', such as ' Here we analyze...' instead of 'analyzed' to convey to the reader this is new data being presented.

Comment 11: Page 5 Line 21: Here you could explain how this study is different than what has already been published from the Chinese snow survey in 2014.

Comment 12: Page 6 Line 11: I think it is a bit misleading to say 'we explore climatic effects of ILAPs' as far as the reader can see, albedo reduction is explored, but climate models are not employed. Please rephrase or clarify.

Comment 13: Page 6 Line 13, 'Therefore' => 'Here'. Or clarify if something else is meant.

Comment 14: Page 6 Line 14: Create two sentences. 'campaign. Snow albedo is also...' Also, when was the snow campaign? 2014? Please include year for clarity.

Comment 15: Page 6 Line 15: Is this the first presentation of SAMDS? If so rephrase sentence to explain 'SNICAR and 'the new' SAMDS...'

Comment 16: Page 6 Line 19: Include sub-heading for 'Site Introduction'. Also, include the time range the samples were collected in this paragraph, was each site visited once over the month?

Comment 17: Page 7: In the site introduction, please explain why only sites 90 – 102 were visited. Why do the numbers begin at 90? Where there 90 sites in 2010 where there was not snow in 2014?

Comment 18: Page 7: This is troubling to the reader. If different volumes of sample were collected based on how dirty a site was, does this make the measurements comparable? When the sample is filtered, does the volume not matter? How much volume was filtered? The total sample? Please explain.

Comment 19: Page 7 Line 13: Wang 2015 => It appears some of the 2014 Chinese snow survey data has been previously published. A more in depth explanation in the

introduction should explain how this study builds on previously published results of the same snow study, and what data is new. Also, please state what chemistry was analyzed in Wang 2015, just stating it was 'similar' to other studies by Hegg et al., is not sufficient. Also, it is not clear with the next sentence, were the major ions described in Wang 2015? This paragraph needs to be re-written to clearly state what has been published and what is new data being presented here. Or if results from Wang 2015 are being applied here in a new analysis.

Comment 20: Page 8 Line 11: The first sentence should be rewritten as its not clear to the reader how iron would originate from mineral dust in seasonal snow. Please explain in the text why this assumption is made.

Comment 21 Page 9 Line 5: Why is it stated that the Microtops II Sun Photometer was used instead of the CE318? Please explain the significance for stating this and the difference between the two. What does the Microtops II do that the CE318 does not?

Comment 22 Page 9 Line 14: include ' in this study'... during the 2014 Chinese survey. (If that is what the authors mean, otherwise please clarify).

Comment 23 Page 9 Line 24: Spectroradiometers have been used in a number of other studies than listed here. Please include 'e.g.' and then the citations.

Comment 24 Page 10 Line 5: How was the spectroradiometer held level? What was the field of view? How did you minimize shadows from the leveling device? What were the solar zenith angles? This data should all be included in the text and solar zenith angles should be presented in a table.

Comment 25 Page 11 Line 3: Is this the first presentation of the SAMDS model in publication? If so this needs to be highlighted more in the abstract, title, and especially the text here. Otherwise please include a citation for the SAMDS model.

Comment 26: In general it seems this paper could be more focused around the SAMDS model as this appears to be the novelty of the paper.

Comment 27: Page 14 Line 13: This sentence is confusing and needs to be revised. For example: Delete 'For' and begin sentence with 'Most of the snow samples were collected in the afternoon corresponding to the Aqua-MODIS (13:30LT) overpass time. Then the next sentence starting with 'The averaged...'"

Comment 28: Page 15 Lines 5 – 10. These sentences need to be restructured and concise.

Comment 29: Page 15 Line 13: Remove 'processes'.

Comment 30: Page 15 Line 21: Please explain 'left and right' samples. Do the authors mean sample duplicates? This sentence needs restructuring.

Comment 31 Page 15 Lines 21 - 25: Major run-on sentence, which needs revision.

Comment 32: Page 15 Lines 21 – 25: If the snowpack was so thin and patchy – won't blowing soil/dirt be an issue for the measurements? Please explain how this is accounted for.

Comment 33 Page 16: What type of snow was present? Fresh snow or old snow? Please explain using the international snow classification. Also, how soon were samples collected after snow falls? Also, it would be assumed that since the study was conducted in January, during Chinese winter, some sites could have fresh snow, where others could be 'older' snow?

Comment 34 Page 16 Lines 14 - 19: Run-on sentence again. Please separate into two sentences. 'The results from the 2010 – 2014 Chinese snow surveys are shown in.... )

Table 1: Needs to be reorganized by dates collected (i.e. 2010 and 2014). Also include the number of samples incorporated in the average values. Sample depth is confusing... so the samples were collected over 5 cm and integrated into one measurement? What about the surface samples were less volume was collected at sites that were particularly dusty/polluted, i.e. Page 7 Line 4. Separate surface samples and

snow pit data. Also, it appears

Table 2: Include snowpack depth for each site, to see how snow pack depth varied from 2010-2014 as that may have influenced measured BC content. Also, given the method used, should BC = eBC? And was this 2010 data published previously? If so, include a citation. If the 2010 data has been published before, then it appears that there are only 12 new data points in this study from 2014. Why are sites 41 – 46 listed with sample dates but no avg BC value?

Table 3: Wasn't this data publishes in Wang 2015, from sub section 2.2 Chemical Speciation Page 7 Line 13? If not, or if so clarify either way, provide references, and explain why the data is being presented again and what is new.

Figure 4: Site numbers should be included with the map and the 2014 data should only be presented since the 2010 data is already published? Atleast a distinction should be made between the 2010 and 2014 data. They should not be averaged together as there are many factors that would influence the differing values. Figure 5: These data do not appear to be normally distributed as they are heavily weighted on the low end, and just a couple larger concentrations. Also provide the p-value to show the significance of the fit.

Figure 6: Again, like Figure 5, these data do not appear to be normally distributed as they are heavily weighted on the low end, and just a couple larger concentrations. Also provide the p-value to show the significance of the fit. Why is there one red dot at 0.25 AL and 1.5 K+?

Figure 7: This is a nice figure. The land cover legend should be mentioned in the figure legend.

Figure 8: As in Figure 7, the land cover legend should be mentioned in the figure legend.

Figure 9: Why is only the visible range presented? Please clarify in Figure legend.

Comment 34 Page 16 Lines 14 - 19: The reader doesn't understand this sentence. Are these averaged values from the surface samples?

Comment 34 Page 16 Line 19: Include standard deviations with the ranges. How many samples were in this range?

Comment 35: Page 17 Line 5: Include a reference to the table where this data is displayed.

Comment 36: Page 17 Line 11: Where is this regression value from? Reference figure.

Comment 37: Page 18 Line 2: Pleas explain how photochemical reactions are related to biomass burning contributions of OC?

Comment 37: Page 18 Line 3: p-value? And please include for the other figures.

Comment 38 Page 19: The reader finds this paragraph confusing. Are the percentges from Figure 7? If the samples were collected in January 2014, why are there seasonal/time references? i.e. Line 11" Sulfate peaks were found in summer... this should be clearly tied to the Zhang 2013 reference. If the next sentence, line 12 – 17 (Which is a very long run-on sentence) also refers to Zhang 2013 then this needs to be more clearly stated. The combination of presenting results from this study and Zhang 2013 needs to be reorganized more clearly. For example a paragraph explaining Zhang 2013 could come first, and then compare this study to those. Also Zhang 2013 should probably be more clearly explained in the introduction. Again, it also appears that Wang 2015 already published snow chemistry data from these results? This needs to be more clearly referenced in the paper.

Comment 39 Page 20 Line 1: Include citation for iron originating from industrial emissions.

Comment 39 Page 20 Line 3: Is this sentence about 'this' study, or Doherty 2014?

Comment 39 Page 20 Line 6: 'Here, light absorption... '

Comment 40: Page 21 Line 4- 6: This has already been stated and should be in the methods.

Comment 41 Page 21: The reader is not sure what the authors mean by 'Higher degrees of snow albedo...' do they mean more reduction or a high albedo? 'There was a larger reduction in snow albedo for....'

Figure 10: Include labels for A, B, C. Same goes for all other figures.

Comment 42 Page 22: Why is BC now refereed to as the ' BC mixing ratios'. Please clarify where this change in terminology was introduced.

Comment 43: Overall, ILAPs is not used consistently in the paper as light absorbing aerosols is written out many times. The reader suggests to standardize this.

Comment 44 Page 23 Lines 23 – 35: Please explain why, " ...this study shows that the spectral albedo fo snow reduction caused OC levels to increase?" Shouldn't the reduction in spectral albedo be a result of the OC levels?

Comment 45: The paper needs a lot of reorganization. For a 25 page paper, one page of Discussion does not seem adequate. Rename Subtitle 4, "Conclusions".

Comment 46 Page 24 Line 4: Incude the 13 sites also in the abstract. This description from lines 3 – 7 should be included in the site description and does not need to be repeated in the conclusion. Comment 47: Page 24 Line 22: Please state the range in latitudes covered in this study in the site description. Did the 13 sites cover a large latitudinal gradient?

Comment 48 Page 25 Line 13: The last sentence of the entire manuscript seems like an odd conclusion, 'to include OC as an input parameter in the SNICAR model'. Why not argue for the use of the SAMDS model? Does it not include OC as an input parameter? A better conclusion would be to compare the SAMDS and SNICAR models. Is the reason the SNICAR model is suggested to incorporated OC because it is widely used? Did the authors try incorporating OC as an input parameter in the off-line SNICAR

code? The discussion and conclusion of this manuscript needs revision.

---

## Author Response (AR1)

**Response to Referee #1**

We are very grateful for the referee's critical comments and suggestions, which have helped us improve the paper quality substantially. We have addressed all of the comments carefully as detailed below in our point-by-point responses. Our responses start with "R:".

This paper uses snow sample observations across northern China in January 2014 and aerosol radiation models to examine the reduction of snow albedo due to black carbon, organic carbon and anthropogenic dust. The study suggests different contributions to the snow albedo reductions from these aerosols and suggests that biomass burning may be a major contributor at most snow sampling sites. It also evaluates the model simulations based on the observations. In general, this paper provides useful information, particularly the different performance in snow albedo reduction by three types of aerosols. However, there are still a few limitations. The small sample volume observed over short period at limited sites could make the results questionable. At least discussions about the uncertainties and potential issues in this study are necessary. Corresponding to the limited data, reliable or strong quantitative conclusion is hard to obtain. The authors could provide more quantitative results and discuss the reliability of the findings in the study.

R: We agree with the reviewer that the number of snow sample is small from the only 13 monitoring sites. One reason for the small number of samples in this season (2014) was due to the less snowfall across northern China. Despite with the small snow data samples, we have designed the study carefully by running two models and combing with various other data sets to generate new scientific knowledge. The importance of our study can be reflected in the following aspects:

(1) Seasonal snow amount at mid-high latitude regions has large spatial variations in any given year. One novelty of the present study is to investigate the spatial and vertical variations of BC in seasonal snow and attribute the light absorption to BC, OC and mineral dust.

(2) Several recent studies have indicated that the mixing states of BC and the irregular morphology of snow grain have large effects in snow albedo reduction (He et al., 2014; Liou et al., 2011, 2014). Based on the comments from this reviewer and from the interactive comments provided by Cenlin He, we have provided a new figure in the revised paper discussing the uncertainties in the mixing states and the irregular morphology of snow grains on snow albedo reduction estimation using SAMDS model simulations (Figure 10).

(3) The title has been changed as "Observations and model simulations of snow albedo reduction in seasonal snow due to insoluble light-absorbing particles during 2014 Chinese survey".

The manuscript is totally rewritten, and more results in discussing the SAMDS for snow albedo reduction due to ILAPs in snow, internal/external mixed BC in snow, and the snow grain shapes were given in the abstract, introduction, methods, results, and the conclusions. Details have already been illustrated in comment 1.

Are the snow samples fresh or aged snow?

R: We have already updated this information in Table 1 and Figure 1.

(2)Snow albedo reduction due to these kinds of aerosols is well known, more quantitative results are needed and more valuable, which could be summarized in the abstract. At current version, it is a little hard for me to summarize the findings I can learn from this paper.

R: We agree with the reviewer that optical properties of black carbon in snow have been investigated in earlier studies (e.g. Liou et al., 2014; Qian et al., 2014; Warren et al., 1980). However, very few studies have focused on light absorption by OC and dust in snow, which is one of the focuses of our study here. We have investigated snow albedo reduction by OC and mineral dust using the SNICAR model and a new SAMDS model. The impact of the uncertainties in the mixing states and the irregular morphology of snow grain on albedo reduction is also illustrated using SAMDS model simulations (Figure 10)..

Page 3, line 11-14, why is Ginoux et al. 2010 cited two times in one sentence (begin and end). This is repeated description and one should be deleted.

R: We have deleted the second citations of Ginoux et al., (2010), and we have also corrected the similar mistakes throughout the manuscript.

Page 4, line 4-5, you use "larger" and "more intense" in the sentence, but I did not see any comparison descriptions around this sentence. What you are comparing?

R: We have added a reference of Guan et al. (2016), which illustrated the relationship between anthropogenic dust and population over global semi-arid regions, and the source attribution of insoluble light-absorbing particles in seasonal snow across northern China. Furthermore, Wang et al. (2013a) also indicates that BC emission sources in China are strongest in far eastern China of our northeast China snow sampling than the other regions across northern China (Figure 11 in Wang et al., 2013a).

Page 4, Line 12-13, why is Light et al. 1998 cited two times in one sentence (begin and end). This is repeated description and one should be deleted.

R: Corrected.

Page 5, line 15-24, you mentioned several campaigns for snow collection. What are the differences or similarities for the findings among them? Also, it shows that there is a snow campaign over the examined region in 2010 carried out by Huang et al. 2011. Why do not you also include the observation from this campaign so that you have enough data samples and you can also compare the differences/similarities in two winters?

R: In this manuscript, we try to focus on the new findings due to ILAPs in snow during 2014 field campaigns more clearly. As a result, we deleted figure 6 and section

3.3. Figure 4 was also modified as figure 4a and 4b based on 2014 snow field campaigns, which weren't published in previous studies.

As Doherty et al., (2015) indicated that with no measure of the interannual variability of the mixing ratio of BC in snow, it is difficult to determine the representativeness of the samples collected in the Arctic survey.

R: We agree with the reviewer. For this reason, we note that further field campaigns on measuring mixing ratio of BC in snow should be performed worldwide in northern Hemisphere. However, these datasets were much useful for climate models to reduce the uncertainty of the climate effects due to BC in snow.

Page 7, line 1-3, is this data criteria enough to prevent contamination? And why do you include site 101 if it does not fit your data criteria?

R: We indicate that the datasets used in this study are criteria enough to prevent contamination. The reason is that the snow sample collection and the analysis procedure were strictly performed followed by Doherty et al. (2010) and Grenfell et al. (2011). Although site 101 is close to the village, we point out that the ILAPs in snow are more representative for the country village regions across northern China.

Page 7, line 5-10, what is the uncertainty introduced due to your visual inspection and data processing method?

R: As shown in Grenfell et al. (2011), visual comparison is best carried out under diffuse reflected illumination with the filters sitting on a white diffusing background. Uncertainties including personal bias involved on measuring BC in snow is approximately a factor of ~1.5-2. The causes of the bias in the visual estimates of the China 2010 field filters vs. those from the expedition reported by Grenfell et al. (2011) is unknown (Grenfell et al., 2011; Wang et al., 2013a).

Page 8, line 1-9, what do some variables (not all) stand for?, such as Ss?

R: The subscript of Ss means sea salt sources. We have added explanations for all the variables that were not labeled in this manuscript.

Page 8, line 17, why do not you introduce Fe when iron is first used in paper?

R: We indicated that Fe in this manuscript is the same as iron (Page 9, line 17) . Another consideration is that when discussing chemical elements, the symbol Fe instead of iron is commonly used.

Page 9, line 4-6, How do you know Microtops II is reliable, or more reliable than CE318?

R: We indicated that Microtops II and CE318 are both effective instruments on measuring aerosol optical depth (AOD) (More et al., 2013; Porter et al., 2001; Zawadzka et al., 2014). However, the major difference between Microtops II and CE318 is that the Microtops II is portable for the field experiments, but CE318 is immovable. For the snow survey, it is better to use the Microtops II instrument

instead of CE318 to measure aerosol optical depth.

Page 9, Line 22, do you mean "in 2014"?

R: We have already corrected this sentence as "Fire locations were based on data provided by the MODIS FIRMS system from October 2013 to January in 2014."

Page 10, line 4-5 and line 19-20, why do the observations and models calculate the albedo at different height (1 m above snow vs at surface)? Also, the downwelling solar radiation in the real sky includes diffuse radiation. How does the model consider the diffuse radiation (such as the contribution from aerosols and clouds in the sky)?

R: Normally the relative position of the sighting laser spot is at a distance of 1m from the optical element for the active field of view for the instrument in strict accordance with the user manual of the SVC HR-1024 spectroradiometer (Figure 6 Setup for FOV map). The direction of the instrument was oriented to the Sun Horizon angles in order to receive more direct solar radiation. The small size of the fore optics greatly reduces errors associated with instrument self-shadowing. Even when the area viewed by the fore optic is outside the direct shadow of the instrument, the instrument still blocks some of the illumination (either diffuse skylight or light scattered off surrounding objects) that would normally be striking the surface under observation for measuring full-sky-irradiance throughout the entire 350 - 2500 nm wavelengths. This spectroradiometer is used for measuring the direct component of solar irradiance because of the minimized relative radiometric errors between total and direct irradiance measurements. For instance, Bi et al. (2013) used a set of broadband radiometers and sun/sky photometers during 2013 field campaign in the middle latitude across northern China to measure the direct and diffuse solar irradiance, and the result indicated that the diffuse solar radiation is 10% lower than the total solar irradiance. Therefore, we indicated that the spectroradiometer in the clean sky condition mainly measured the direct solar irradiance during 2014 snow campaign. The above materials have been added in section 2.4 of the revised paper to present the relative parameter of the spectroradiometer.

Page 11, line 15, the symbol should be μ, please explain its meaning.

R: We have changed the symbol as μ in Page 14, line 16, and μ refers to the escape function in radiative transfer theory (Kokhanovsk and Zege, 2004).

Page 15, line 7-10, both AOD from MODIS and ground are retrieved, please describe clearly.

R: The ground AOD is retrieved by Microtops II sun photometer. We have already indicated that Microtops II sun photometer is an effective instrument on measuring AOD. However, the weakness of this instrument is that it can only precisely measure AOD during the clean sky. So we only measured the ground AOD dataset in six sites. Then we used the MODIS AOD dataset to compare with our ground measurements to indicate the spatial variations of AOD across northern China. The active open fire retrieved from MODIS was also used to show the possible sources of the BC and OC.

Page 15, line 16, colder -> cold

R: corrected.

Page 15, line 23, This sentence is not complete. I believe what you want to say is "..., which with ..."

R: This sentence has been modified as: "In Inner Mongolia, the snow cover was thin and patchy. The average snow depth at sites 90, 91, 93, and 94 was less than 10 cm, which was significantly smaller than those (13 to 20 cm) at sites 95-97 near the northern border of China." in page 18 lines 20-24.

Page 16, line 2-4, "Because less snow fell during the 2014 snow survey period, the surface snow grain radius varied considerably from 0.07 to 1.3 mm.". First, I do not understand this causal relationship, please clarify. Second, what does it compare when using 'less', the same period of other years or other locations?

R: We have revised the sentence as: "The maximum snow depth was found to be 46 cm at site 102 inside a forest near the Changbai Mountains. Snow depth varied from 13 to 46 cm at sites 98 to 102 with an average of 27 cm. $R_m$ of the snow samples varied considerably from 0.07 to 1.3 mm. $R_m$ increased with the snow depth from the surface to the bottom, larger than previously recorded because of snow melting by solar radiation and the ILAPs." in page 19 lines 5-10.

Page 16, line 19-24, why the variations (BC and snow spatial distribution) you found were much higher than the findings from other studies?

R: BC in snow in this manuscript was mostly collected in heavy industrial regions in northern China, where the mixing ratios of BC and OC were much higher than in the other regions of northern China (Flanner et al., 2007; Wang et al., 2013a; Zhao et al., 2014).

Page 17, line 1-8, I cannot catch the main points you would like to deliver here.

R: This section has been completely rewritten for clarification.

Page 17, Figure 5, the sample volume is too small. How reliable are the relationships found here?

R: We have added the datasets of BC measurements in seasonal snow during 2010 field campaign in Figure 5. The caption of Figure 5 was rewritten as "Comparisons between the calculated and optically measured $C_{BC}^{est}$ in surface snow during 2010 and 2014 snow surveys. The datasets of measured $C_{BC}^{est}$ in 2010 from sites 3-40 were reprinted from Wang et al. (2013a).". We have also provided the confidence test of the fitting in Figure 5.

Page 17, line 22-24, I do not understand from Figure 6 how you got this range in OC/BC ratio.

R: See our answer to the next question.

Page 18, line 1-16, it is hard to conclude due to limited data and sites.

R: Following the reviewer's suggestions, we chose to concentrate on the ILAPs in snow and the snow albedo reduction due to internal/external ILAPs in snow, and the snow grain shapes. Therefore, we have deleted figure 6, and added a new figure in discussing the snow albedo reduction due to internal/external mixed BC in snow and difference snow grain shapes as Figure 10 in this revised manuscript.

Page 18, line 18-22. How did you get the observation regarding land-cover type when the land is covered by snow?

R: The land-cover types (Figure 7) were obtained from the Collection 5.1 MODIS global land-cover type product (MCD12C1) at a 0.05°spatial resolution. The dataset included 17 different surface vegetation types (Friedl et al., 2010; Loveland and Belward, 1997). What we wanted to demonstrate is that most of the sampling regions were correlated with human activities, while this manuscript mainly focused on the anthropogenic dust and the other ILAPs in seasonal snow.

(25)Page 19, Line 17-20, why is Wang et al. 2015 cited two times in one sentence (begin and end). This is repeated description and one should be deleted.

R: Corrected.

References:

Bi, J. R., Huang, J. P., Fu, Q., Ge, J. M., Shi, J. S., Zhou, T., and Zhang, W.: Field measurement of clear-sky solar irradiance in Badain Jaran Desert of Northwestern China, J. Quant. Spectrosc. Ra., 122, 194-207, 10.1016/j.jqsrt.2012.07.025, 2013.

Doherty, S. J., Steele, M., Rigor, I., and Warren, S. G.: Interannual variations of light-absorbing particles in snow on Arctic sea ice, J. Geophys. Res.-Atmos., 120, 11391-11400, 2015.

Doherty, S. J., Warren, S. G., Grenfell, T. C., Clarke, A. D., and Brandt, R. E.: Light-absorbing impurities in Arctic snow, Atmos. Chem. Phys., 10, 11647-11680, 2010.

Flanner, M. G., Zender, C. S., Randerson, J. T., and Rasch, P. J.: Present-day climate forcing and response from black carbon in snow, J. Geophys. Res.-Atmos., 112, D11202, 2007.

Friedl, M. A., Sulla-Menashe, D., Tan, B., Schneider, A., Ramankutty, N., Sibley, A., and Huang, X. M.: MODIS Collection 5 global land cover: Algorithm refinements and characterization of new datasets, Remote Sens. Environ., 114, 168-182, 2010.

Ginoux, P., Garbuzov, D., and Hsu, N. C.: Identification of anthropogenic and natural dust sources using Moderate Resolution Imaging Spectroradiometer (MODIS) Deep Blue level 2 data, J. Geophys. Res.-Atmos., 115, D05204, 2010.

Grenfell, T. C., Doherty, S. J., Clarke, A. D., and Warren, S. G.: Light absorption from particulate impurities in snow and ice determined by spectrophotometric analysis of filters, Appl. Opt., 50, 2037-2048, 2011.

Guan, X. D., Huang, J. P., Zhang, Y. T., Xie, Y. K., and Liu, J. J.: The relationship between anthropogenic dust and population over global semi-arid regions, Atmos. Chem. Phys., 16, 5159-5169, 2016.

He, C. L., Li, Q. B., Liou, K. N., Takano, Y., Gu, Y., Qi, L., Mao, Y. H., and Leung, L. R.: Black carbon radiative forcing over the Tibetan Plateau, Geophys. Res. Lett., 41, 7806-7813, 2014.

Huang, J. P., Fu, Q. A., Zhang, W., Wang, X., Zhang, R. D., Ye, H., and Warren, S. G.: Dust and Black Carbon in Seasonal Snow across Northern China, Bull. Amer. Meteor. Soc., 92, 175-181, 2011.

Kokhanovsky, A. A., and Zege, E. P.: Scattering optics of snow, Appl. Opt., 43, 1589-1602, 2004.

Liou, K. N., Takano, Y., and Yang, P.: Light absorption and scattering by aggregates: Application to black carbon and snow grains, J. Quant. Spectrosc. Ra., 112, 1581-1594, 2011.

Liou, K. N., Takano, Y., He, C., Yang, P., Leung, L. R., Gu, Y., and Lee, W. L.: Stochastic parameterization for light absorption by internally mixed BC/dust in snow grains for application to climate models, J. Geophys. Res.-Atmos., 119, 7616-7632, 2014.

Loveland, T. R., and Belward, A. S.: The IGBP-DIS global 1 km land cover data set, DISCover: first results, Int. J. Remote Sens., 18, 3291-3295, 1997.

More, S., Kumar, P. P., Gupta, P., Devara, P. C. S., and Aher, G. R.: Comparison of Aerosol Products Retrieved from AERONET, MICROTOPS and MODIS over a Tropical Urban City, Pune, India, Aerosol. Air. Qual. Res., 13, 107-121, 2013.

Porter, J. N., Miller, M., Pietras, C., and Motell, C.: Ship-based sun photometer measurements using Microtops sun photometers, J. Atmos. Oceanic Technol., 18, 765-774, 2001.

Qian, Y., Wang, H. L., Zhang, R. D., Flanner, M. G., and Rasch, P. J.: A sensitivity study on modeling black carbon in snow and its radiative forcing over the Arctic and Northern China, Environ. Res. Lett., 9, 064001, 2014.

Wang, X., Doherty, S. J., and Huang, J. P.: Black carbon and other light-absorbing impurities in snow across Northern China, J. Geophys. Res.-Atmos., 118, 1471-1492, 2013a.

Wang, X., Pu, W., Zhang, X. Y., Ren, Y., and Huang, J. P.: Water-soluble ions and trace elements in surface snow and their potential source regions across northeastern China, Atmos. Environ., 114, 57-65, 2015.

Warren, S. G., and Wiscombe, W. J.: A Model for the Spectral Albedo of Snow .2. Snow Containing Atmospheric Aerosols, J. Atmos. Sci., 37, 2734-2745, 1980.

Yasunari, T. J., Koster, R. D., Lau, W. K. M., and Kim, K. M.: Impact of snow darkening via dust, black carbon, and organic carbon on boreal spring climate in the Earth system, J. Geophys. Res.-Atmos., 120, 5485-5503, 2015.

Zawadzka, O., Makuch, P., Markowicz, K. M., Zielinski, T., Petelski, T., Ulevicius, V., Strzalkowska, A., Rozwadowska, A., and Gutowska, D.: Studies of aerosol optical depth with the use of Microtops II sun photometers and MODIS detectors in coastal areas of the Baltic Sea, Acta Geophysica, 62, 400-422, 2014.

Zhao, C., Hu, Z., Qian, Y., Leung, L. R., Huang, J., Huang, M., Jin, J., Flanner, M. G., Zhang, R., Wang, H., Yan, H., Lu, Z., and Streets, D. G.: Simulating black carbon and dust and their radiative forcing in seasonal snow: a case study over North China with field campaign measurements, Atmos. Chem. Phys., 14, 11475-11491, 2014.

**Response to Cenlin He's comments**

We thank Cenlin He for his comments, which have heled us improve the paper quality. We have incorporated these comment into the revised paper as detailed below.

The authors conducted extensive field measurements of BC, OC, and dust concentrations in snow over the northern China and employed the SNICAR and SAMDS model simulations to investigate snow albedo reduction caused by the light-absorbing aerosols. This study provides a valuable observational dataset to improve our understanding in the effects of light-absorbing aerosol deposition on snow albedo reduction. I have a short comment on snow albedo modeling. The authors mentioned that the SAMDS model considers aerosol-snow mixing state and the irregular morphology of snow grain by using asymptotic radiative transfer theory. However, the authors did not provide enough discussions on the effects of BC/dust/OC-snow mixing state and snow grain shape on snow albedo reduction as well as compare with some recent studies. These factors are not trivial in evaluating aerosol-snow albedo effects, in addition to snow grain size and aerosol concentration. For example, recent studies by Liou et al. (2014) and He et al. (2014) developed and applied a stochastic snow model to study BC/dust-induced snow albedo reduction, which explicitly simulates different aerosol-snow mixing states and snow grain shapes. They found that using a realistic snowflake shape reduces BC-induced snow albedo reduction by 20–40% compared to a spherical snow grain, while multiple internal mixing of BC and snow increases the albedo reduction by 40–60% relative to the external mixing. I would suggest discussing these recent findings, which could be very helpful for people to understand potential uncertainty and improvement in snow albedo modeling.

R: We noted that the two references by Liou et al. (2014) and He et al. (2014) are both very important achievements for understanding light absorption by internally mixed BC/dust in snow grains for application to climate models. Therefore, we also used the SAMDS model to simulate the snow albedo change by internal/external mixing of BC and snow associated with the irregular morphology of snow grains using asymptotic radiative transfer theory (Figure 10). The following discussions on snow albedo change due to internal/external mixing states of ILAPs associated with irregular snow grains in snow have been added in the revised manuscript.

1.  Introduction
"Warren and Wiscombe (1980) found that a mixing ratio of 10 ng g$^{-1}$ of soot in snow can reduce snow albedo levels by 1%. Light et al. (1998) determined that 150 ng g$^{-1}$ of BC embedded in sea ice can reduce ice albedo levels by a maximum of 30%. 1 ng g$^{-1}$ of BC has approximately the same effect on the albedo of snow and ice at 500 nm as 50 ng g$^{-1}$ of dust (Warren, 1982). Doherty et al. (2013) analyzed field measurements of vertical distributions of BC and other ILAPs in snow in the Arctic during the melt season and found significant melt amplification due to an increased

mixing ratio of BC by up to a factor of 5. Yasunari et al. (2015) suggested that the existence of snow darkening effect in the Earth system associated with ILAPs contributes significantly to enhanced surface warming over continents in northern hemisphere midlatitudes during boreal spring, raising the surface skin temperature by approximately 3–6 K near the snowline. Warren and Wiscombe (1985) pointed out that modeling soot in snow as an "external mixture" (impurities particles separated from ice particles) may underestimate the true effect of the impurities as a given reduction of albedo by about half as much soot, if the soot is instead located inside the ice grains as an "internal mixture". Hansen et al. (2004) and Cappa et al. (2012) noted that for a given BC mass on snow albedo, the internal mixing of BC in snow is a better approximation than external mixing, whereas internal mixing increases the BC absorption coefficient by a factor of two, for better agreement with empirical data. Hadley and Kirchstetter (2012) also indicated that increasing the size of snow grains could decrease snow albedo and amplify radiative perturbation of BC. For a snow grain optical effective radius ($R_{eff}$) of 100 μm, the albedo reduction caused by 100 ng $g^{-1}$ of BC is 0.019 for spherical snow grains but only 0.012 for equidimensional nonspherical snow grains (Dang et al., 2016). Fierce et al. (2016) pointed out that BC coated with non-absorbing particles absorbs more strongly than the same amount of BC in an uncoated particle, but the magnitude of this absorption enhancement is still a challenge. He et al. (2014) indicated that BC-snow internal mixing increases the albedo forcing by 40–60% compared with external mixing, and coated BC increases the forcing by 30–50% compared with uncoated BC aggregates, whereas Koch snowflakes reduce the forcing by 20–40% relative to spherical snow grains using a global chemical transport model in conjunction with a stochastic snow model and a radiative transfer model."

2.5 Model simulations

The contents the asymmetry factor for calculating the snow albedo change by irregular snow grains have been added in Section 2.5 as: "
[revised manuscript text omitted]

**Response to Referee #2**

We greatly appreciate this referee's critical comments and suggestions, which have helped us improve the paper quality substantially. We have addressed all of the comments carefully as detailed below in our point-by-point responses below. Our responses start with "R:".

Due to the high amounts of industrialization in China, there is a need for studies of impurities in seasonal snow. This study is on a Chinese snow survey conducted in 2014. However, it is not clear how much of the data presented in this study has already been published. The authors are recommended to clarify, in much more detail, what data has been previously published, and what are novel results from this study and included in this manuscript. For example, it is not clear whether this study presents new snow chemistry data, or if this data has been published in another manuscript, such as Wang 2015. It is also unclear why the 2010 Chinese Snow survey data is included in the study. Averaging results from the 2010 and 2014 surveys seems inappropriate, especially because the snow conditions/depths appear to vary widely since so many sites did not even have enough snow to resample in 2014. The paper does not appear to be ready for publication in this submission, even with the following revisions. Major restructuring is needed.

R: We admit that the ILAPs and the chemical species in seasonal snow across northern China have already been investigated in previous studies (Huang et al., 2011; Wang et al., 2013a, 2015; Ye et al., 2012, Zhang et al., 2013a). Similar experiments on measuring ILAPs in seasonal snow have also been conducted on the Arctic, Greenland, North America, even in Loess Plateau in recent years (Doherty et al., 2010; 2014; Ming et al., 2009). For instance, a similar paper on the mixing ratios of ILAPs in Arctic snow (Doherty et al., 2010) has been widely used for validating modeled snow BC mixing ratios. There are also heavy loadings of mineral dust in present snow, which could also lead a rapid snow albedo reduction (Yasunari at al., 2015). However, the ability to test model representation of ILAPs in snow via climate modeling is still critical. According to the model simulations by Qian et al. (2014) and Zhao et al. (2014), although the model simulates reasonably well the magnitude of BC mixing ratios in the middle latitudes, the models generally moderately underestimates BC in snow in the clean regions but significantly overestimates BC in some polluted regions. We thus realized that the snow albedo could be reduced remarkably due to the large variations of ILAPs in seasonal snow in those regions across northern China, which could lead large biases of the radiative forcing of ILAPs in seasonal snow due to the model simulations. Based on the above mentioned studies, we think there is a need to investigate the spatial and vertical variations of BC in seasonal snow and attribute the light absorption to BC, OC and mineral dust. Another purpose of the study is to reveal the snow albedo reduced by ILAPs in snow between surface measurements and model simulations using a standard spectroradiometer, two snow/ice radiative models of the Snow, Ice, and Aerosol Radiation (SNICAR) model, and a new radiative model (Spectral Albedo Model for Dirty Snow, or SAMDS).

Based on the comments from all the reviewers, we have made the following major revisions to improve the quality and clarity of the paper:

(1) The title has been changed to "Observations and model simulations of snow albedo reduction in seasonal snow due to insoluble light-absorbing particles during 2014 Chinese survey".

(2) The abstract has been rewritten to accurately reflect the new results.

(3) We have deleted the part of discussion in the Introduction related to radiative forcing of ILAPs. We, however, have added discussion on the internal/external mixing state of BC in snow and different snow grain shapes in affecting snow albedo, which is more correlated with the scope of this manuscript.

(4) The result section was changed from "section 3.1-3.6" to "section 3.1-3.4" as follows:
3.1 The spatial distribution of AOD
3.2 Contributions to light absorption by ILAPs
3.3 Simulations of snow albedo
3.4 Comparison between the observed and modeled snow albedo

(5) We have deleted the Figure 6 and section 3.3, whih discussed the emission sources by using the chemical species in previous version.

(6) We have added analysis on the uncertainty of the snow albedo reduction by the mixed BC in snow by different snow grain shapes. Therefore, a new figure is added as Figure 10 in the revised manuscript to present the snow albedo change due to the internal/external mixing of BC in snow and the snow grain shapes by using the SAMDS model compared with previous studies (He et al., 2014; Liou et al., 2011, 2014).

(7) We have revised the "Conclusions" to better reflect the revised contents.

The novelty of the paper appears to begin in the model comparisons with the albedo surface observations made in 2014. The reader suggests the authors significantly revise the paper to focus solely on the 2014 observations (unless the reader is confused by the text and the 2010 data has not yet been published?).

R: The 2010 snow survey datasets used in this study was deleted. We noted that most of the chemical datasets used in this study were to retrieve the new datasets of sea salt and $K^+_{Biosmoke}$ depending on the previous studies, which were not shown by Wang et al. (2015). However, only $SO_4^{2-}$, $NO_3^-$ and $NH_4^+$ data from Wang et al. (2015) were reused in the present study to indicate the mass contribution of the ILAPs with the other chemical species in snow across northern China. To make this clear, we have added a section to explain the datasets published by Wang et al., (2015) used in this study.

Additionally, according to the methods used, the reader asks the authors to report the BC as equivalent black carbon (eBC) rather than black carbon. If there is a reason otherwise to report as BC, please explain.

R: We have modified "BC" as "$C^{est}_{BC}$" or "$C^{equiv}_{BC}$" in the revised manuscript.

It appears (though clarification is needed from the authors) that a new model, SAMDS, similar to the SNICAR model, is applied here. However, only a small portion of this very long paper is devoted to this section. A short discussion/conclusion at the end needs further expansion. Then for the authors to focus on the observed vs. modeled albedo, as well as provide more information and justification for applying the SAMDS model more widely. The SAMDS model appears to be the main highlight of the paper. The title and paper could be restructured around the SAMDS model.

R: The title of this manuscript has been modified as "Observations and model simulations of snow albedo reduction in seasonal snow due to insoluble light-absorbing particles during 2014 Chinese survey". The manuscript has also been restructured around the SAMDS model throughout the revised manuscript (See abstract, introduction, results, and the conclusions).

The scientific methods used could be explained in a more logical way throughout the methods section of the paper. Also, in the site description, an explanation why the site numbers begin at 90 in 2014 should be given. Also, a map including GPS coordinates and the sampling sites would aid the reader. Additionally, it is stated that the same volume of snow was not collected at all sites. How much volume was filtered at each site? This leads the reader to believe that if the same top volume of snow was not collected at each site, the integrated mass concentration of BC in each snow sample would be different?

R: The site numbers during the snow survey in 2014 have been labeled in Figure 1. We noted that the site numbers beginning at 90 in this study are numbered in chronological order following Wang et al. (2013a) and Ye et al., (2012). Therefore, we have modified the captions in Table 1 and Figure 1 to make this clear. We need to clarify this question about the volume of snow samples. Firstly, enough snow samples have been collected during field campaigns. Secondly, we quickly melt the snow samples in the lab, and filtrated the snow samples by using different liquid snow volume, which depended on the degree of contamination for snow samples. The calculation of $C_{BC}^{max}$, $C_{BC}^{est}$, and $f_{BC}^{est}$ are listed as follows:

$$C_{BC}^{max} = \frac{L_{BC}^{max} \times A}{V}$$

$$C_{BC}^{est} = C_{BC}^{max} \times f_{BC}^{est}$$

$C_{BC}^{max}$ (ng g$^{-1}$): maximum BC is the mass of BC per mass of snow, if all aerosol light absorption at 650–700 nm is due to BC.

$C_{BC}^{est}$ (ng g$^{-1}$): estimated BC is the estimated true mass of BC per mass of snow, derived by separating the spectrally resolved total light absorption.

$f_{BC}^{est}$: Fraction of light absorption by true mass of BC per mass of snow.

Yes, the volume is a very important parameter in collecting the ILAPs on the nuclepore filter. However, due to the mass loading of BC ($L_{BC}^{max}$) can be measured by using the ISSW instrument, we note that the mixing ratios of $C_{BC}^{max}$ and $C_{BC}^{est}$ for each snow samples are definitely comparable based on above calculation. Then, the averaged $C_{BC}^{est}$ for each vertical profile of snowpack is calculated by using following equation:

$$\text{average BC} = \frac{\sum_i (C_{BC_i}^{est} * \rho_i * h_i)}{\sum_i (\rho_i * h_i)}$$

where i is the number of snow layer, $C_{BC_i}^{est}$, $\rho_i$ and $h_i$ are the estimated BC mixing ratio, snow density and snow depth, respectively, at the layer i.

In conclusion, much of the intro and re-presenting of previously published data from the 2010 and 2014 Chinese snow surveys could be removed. And/or better support/clarification for the reason for including it could be included.

R: We have deleted the previously published datasets from 2010 and 2014 in conclusion, and focused more attention on the model simulations, especially for the description of SAMDS model.

Suggestions for Revisions:
The abstract is not currently sufficient. The first sentence states that 92 samples were collected in 2014, however, the 92 points appear to be presented as fewer averages later. The abstract should state how many average values are presented in the study and if the 92 samples were collected across X sites, X snowpits, and at X resolution within the snowpit. Additionally, only surface sample average concentrations are presented in the abstract, how many samples were collected at the surface? And how many in snow pits, and integrated pit samples?

R: We have modified the abstract. The first sentence has been revised as "A snow survey was carried out to collect 13 surface snow samples (10 for fresh snow, and 3 for aged snow), and 79 sub-surface snow samples in seasonal snow at 13 sites in January 2014 across northeastern China. A spectrophotometer and the chemical analysis were used to separate snow particulate absorption by insoluble light-absorbing particles (ILAPs, e.g. black carbon, BC; mineral dust, MD; and organic carbon, OC) in snow, and the snow albedo was measured using a field spectroradiometer during this period."

First of all, at the end of the introduction, previous data published from the 2010 and 2014 Chinese snow survey must be presented clearly. Then, a paragraph outlining what is specifically novel about this study must then be stated. Throughout the manuscript, when previously published data is presented, a reference must be cited for where that data has been published.

R: We have removed all of the datasets from 2010 snow surveys throughout the revised manuscript except for Figure 5, and in the last paragraph we have added discussion on the novelty of this study. We have added the citation and the explanation in the figures very carefully, when used the published datasets by Wang et al., (2015).

Additionally, is this paper the first presentation of the SAMDS model? This should be more clearly stated in the abstract, as this appears to be the main point/novelty of the paper. In general, the paper could be edited to more clearly explain what new data is presented in the study. There are pervasive run-on sentences; the manuscript should be edited to make the language more concise.

R: Yes, the SAMDS model is the first time to be used to reveal the effects of ILAPs, the internal/external mixing of BC and snow, and the snow grain shapes on snow albedo reduction. We have also rewritten the abstract to better illustrate the new

SAMDS model. The revised manuscript is also reconstructed to make the language more concise following the reviewer's comments and suggestions.

The discussion should be expanded to support a more widespread use of the SAMDS model. Also, stating how the model will be disseminated, would aid readers in applying it to their own research. Is there a plan to do so?

R: Yes, the SAMDS model will be definitely disseminated, and we are glad to release the source code after the publication by Zhang et al. (2016). Except the discussion of SAMDS used in this study, the SAMDS model also consider the following processes: (i) mixing states between impurities and snow grains, (ii) the irregular morphology of snow grains and aerosol particles, (iii) specific mineral compositions and size distributions of MD in snow, (iv) aging processes of snow grains and soot aggregates, and (v) multilayers for studying vertical distributions of snow grains and impurities. A detailed description of the SAMDS model will be presented by Zhang et al. (2016). We also note that the SAMDS model can be used to couple with the global climate model (e.g. NCAR Community Atmosphere Model, CAM) to simulate the radiative forcing due to ILAPs in snow and the multiple internal/external mixing of BC in various types of snow grains, which is similar with the previous studies (Flanner et al., 2009; He et al., 2014; Zhao et al., 2014).

Overall, the paper needs major restructuring and revision to portray a logical flow of information. The main weakness is that it is very unclear what is new data and what has previously been published from the 2010 and 2014 surveys and this paper should focus only on what is new.

R: As shown in our responses to the general comments above, we have reconstructed the paper substantially. The main purpose of this manuscript focuses on the spatial variations of ILAPs in seasonal snow during less snow year in 2014, and the discrepancy of snow albedo change by ILAPs in snow and the shapes of snow grains between model simulations and observations.

Specific Line Comments:

Comment 1: Jaffe 1999 cited on Page 3 Line 4 is not included in the bibliography.

R: We have added this reference in the bibliography as follows:

Reference:
Jaffe, D., Anderson, T., Covert, D., Kotchenruther, R., Trost, B., Danielson, J., Simpson, W., Berntsen, T., Karlsdottir, S., Blake, D., Harris, J., Carmichael, G., and Uno, I.: Transport of Asian air pollution to North America, Geophys. Res. Lett., 26, 1999.

Comment 2: Page 4 Line 3: Writing edit: ', with India at' => ' and ' 76.1% in India'

R: revised as suggested.

Comment 3: Page 4 Line 13: 'Among its main light-absorbing impurities' This sentence introduction does not make sense. Could just remove the intro and start with '1ng g-1 of BC'. . ..

R:Revised as suggested.

Comment 4: Page 4 Line 14: 'on the albedo' => 'on the albedo of snow and ice'.

R: Revised as suggested.

Comment 5: Page 4 Line 17: 'owing to' doesn't read well... please alternate words here, as well as multiple other locations in the manuscript. Other suggestions could be, 'due to', etc.

R: We have replaced "owing to" as "due to", "because of", and "Assumption" throughout the revised manuscript.

Comment 6: Page 4 Line 20, first word. Include citation(s) for the previous sentence.

R: Three citations have been added after the first sentence as follow:

Brandt, R. E., Warren, S. G., and Clarke, A. D.: A controlled snowmaking experiment testing the relation between black carbon content and reduction of snow albedo, J. Geophys. Res.-Atmos., 116, 2011.
Hadley, O. L., and Kirchstetter, T. W.: Black-carbon reduction of snow albedo, Nat. Clim. Change, 2, 437-440, 2012.
Warren, S. G. and Wiscombe, W. J.: Dirty Snow after Nuclear-War, Nature, 313, 467-470, 1985.

Comment 7: Page 4 Line 21: In the Bond 2013 paper, sub-title 0.2.3 Synthesis of black carbon climate forcing terms, number 2: "The best estimate of industrial-era climate forcing of black carbon through all forcing mechanisms is +1.1 W m-2 with 90% uncertainty bounds of +0.17 to +2.1 W m-2." Rather than 90 % confidence interval as listed in this manuscript.

R: The sentence has been revised as "Bond et al. (2013) estimated the industrial-era climate forcing of BC through all forcing mechanisms to be approximately +1.1 W m$^{-2}$, with 90% uncertainty bounds of +0.17 to +2.1 W m$^{-2}$."

Comment 8: Page 5 Line 9: 'in situ' => 'in situ' and elsewhere in the manuscript.

R: We have removed "in situ" throughout the manuscript, and instead using "field campaigns". Similar corrections have also been made throughout the revised manuscript.

Comment 9: Page 5 Line 20: 'involve' use different word such as 'present' or 'include'

R: We have changed "involve" as "present" in Page 6, line 24.

Comment 10: Page 5 Line 21: Begin new sentence with 'Previously, . . .we analyzed' if some data from the Chinese snow survey in 2014 was already published and include citation. The reader is confused if some of this data has already been published, or if this data is all unpublished and included in this study. If all the data is unpublished, could start the sentence with, 'Here' instead of 'previously', such as

'Here we analyze...' instead of 'analyzed' to convey to the reader this is new data being presented.

R: The introduction has been mostly reconstructed, and the datasets of the chemical species used in this study have been carefully rewritten in section 2.2.

Comment 11: Page 5 Line 21: Here you could explain how this study is different than what has already been published from the Chinese snow survey in 2014.

R: We have rewritten this paragraph in revised manuscript as "To our knowledge, there are only a few studies that compare modeled and observed snow albedo reduction due to ILAPs in snow (Dang et al., 2015; Flanner et al., 2007, 2012; Grenfell et al., 1994; Liou et al., 2014; Warren et al., 1980). In this study, a 2014 snow survey was performed across northeastern China to analyze light absorption of ILAPs in seasonal snow, and the comparison of snow albedo reduction due to internal/external mixed BC in snow and different snow grain shapes. In section 2, we present the experimental procedures, including a new radiative transfer model (Spectral Albedo Model for Dirty Snow, or SAMDS). After describing our methods (Sect. 2), we demonstrate the light absorption by snowpack containing ILAPs across northeastern China for less snow fallen year through a Chinese survey in 2014 following the snow surveys held in 2010 and 2012 across northern China carried out by Huang et al. (2011) and Ye et al. (2012). Then, a comparison of the snow albedo reduction under clear sky conditions measured by using a field spectroradiometer and simulated by the Snow, Ice, and Aerosol Radiation (SNICAR) model and SAMDS model based on two-stream radiative transfer solution is present. The SAMDS model is also used for the computation of light absorption by complex ILAPs in snow for application to analyze the effects of snow grain shapes (fractal grains, hexagonal plates/columns, and spheres) and internal/external mixing of BC and snow on snow albedo. Finally, conclusions are given in section 4.".

Comment 12: Page 6 Line 11: I think it is a bit misleading to say 'we explore climatic effects of ILAPs' as far as the reader can see, albedo reduction is explored, but climate models are not employed. Please rephrase or clarify.

R: See our reply to comment 11 above.

Comment 13: Page 6 Line 13, 'Therefore' => 'Here'. Or clarify if something else is meant.

R: See our reply to comment 11 above.

Comment 14: Page 6 Line 14: Create two sentences. 'campaign. Snow albedo is also. . .' Also, when was the snow campaign? 2014? Please include year for clarity.

R: See our reply to comment 11 above.

Comment 15: Page 6 Line 15: Is this the first presentation of SAMDS? If so rephrase sentence to explain 'SNICAR and 'the new' SAMDS. . .'

R: Yes, this is the first presentation of the SAMDS model, therefore, we have revised the abstract, results, and the conclusions for the SAMDS model. For the revised paper, the sentence has been revised as "In section 2, we present the experimental procedures, including a new radiative transfer model (Spectral Albedo Model for Dirty Snow, or SAMDS). After describing our methods (Sect. 2), we demonstrate the light absorption by snowpack containing ILAPs across northeastern China for less snow fallen year through a Chinese survey in 2014 following the snow surveys held in 2010 and 2012 across northern China carried out by Huang et al. (2011) and Ye et al. (2012). Then, a comparison of the snow albedo reduction under clear sky conditions measured by using a field spectroradiometer and simulated by the Snow, Ice, and Aerosol Radiation (SNICAR) model and SAMDS model based on two-stream radiative transfer solution is present. The SAMDS model is also used for the computation of light absorption by complex ILAPs in snow for application to analyze the effects of snow grain shapes (fractal grains, hexagonal plates/columns, and spheres) and internal/external mixing of BC and snow on snow albedo. Finally, conclusions are given in section 4."

Comment 16: Page 6 Line 19: Include sub-heading for 'Site Introduction'. Also, include the time range the samples were collected in this paragraph, was each site visited once over the month?

R: This sentence has been revised as "In 2014, there was less snowfall in January than in previous years (e.g., 2010), and only 92 snow samples (13 surface snow, and 79 sub-surface snow samples) at 13 sites were collected during this snow survey." The sub-heading has also been revised as "Snow field campaign in January 2014".

Comment 17: Page 7: In the site introduction, please explain why only sites 90 –102 were visited. Why do the numbers begin at 90? Where there 90 sites in 2010 where there was not snow in 2014?

R: We have collected snow samples in 86 sites in the 2010 and 2012 campaigns across northern China (Figure 1 in Ye et al., 2012), and three snow samples were collected in Lanzhou in 2013 winter, which weren't shown in this study. As a result, the sampling sites in this manuscript are numbered in chronological order beginning at 90 followed Wang et al. (2013a), and Ye et al., (2012). We have added one sentence as "The snow sampling sites in this study began at 90, which are numbered in chronological order followed by Wang et al. (2013a) and Ye et al. (2012)."

Comment 18: Page 7: This is troubling to the reader. If different volumes of sample were collected based on how dirty a site was, does this make the measurements comparable? When the sample is filtered, does the volume not matter? How much volume was filtered? The total sample? Please explain.

R: We need to clarify this question. Yes, the volume is a very important parameter in collecting the ILAPs on the nuclepore filter. We recorded the filtration volume (V) and the surface area of the filter loaded by ILAPs (A), and measured the mass loading of BC ($L_{BC}^{max}$) by using the ISSW instrument, Then, the calculation of $C_{BC}^{max}$, $C_{BC}^{est}$, and $f_{BC}^{est}$ are listed as follows:

$$C_{BC}^{max} = \frac{L_{BC}^{max} \times A}{V}$$

$$C_{BC}^{est} = C_{BC}^{max} \times f_{BC}^{est}$$

$C_{BC}^{max}$ (ng g$^{-1}$)*:* maximum BC is the mass of BC per mass of snow, if all aerosol light absorption at 650–700 nm is due to BC.

$C_{BC}^{est}$ (ng g$^{-1}$): estimated BC is the estimated true mass of BC per mass of snow, derived by separating the spectrally resolved total light absorption.

$f_{BC}^{est}$: Fraction of light absorption by true mass of BC per mass of snow.

Therefore, the calculation of the mixing ratio of BC in snow needs to use the volume to convert the mass loading of BC as mixing ratio of BC in snow; details could be found by Grenfell et al. (2011) and Doherty et al. (2010). Another issue is that there were most of drifting sampling sites in Inner Mongolia and Qilian Mountains as indicated by Wang et al. (2010). Drifted snow is wind-blown, and is more likely to have mixed with locally wind-blowing soil during the drifting process. We thus collected "left" and "right" samples at all sites, and all of the datasets from table 1 are the average values from the two adjacent samples through the whole depth of the snowpack. For the above reason, the dirty layers and new fallen snow were collected in all sites separately, and we think the dirty snow for the single layer has not affected the results, even for the filtration processes. Doherty et al. (2010) also indicated that if there was obvious layering, for example a thin top layer of newly fallen snow or drift snow, that layer was collected separately, however thin.

Comment 19: Page 7 Line 13: Wang 2015 => It appears some of the 2014 Chinese snow survey data has been previously published. A more in depth explanation in the introduction section should explain how this study builds on previously published results of the same snow study, and what data is new. Also, please state what chemistry was analyzed in Wang 2015, just stating it was 'similar' to other studies by Hegg et al., is not sufficient. Also, it is not clear with the next sentence, were the major ions described in Wang 2015? This paragraph needs to be re-written to clearly state what has been published and what is new data being presented here. Or if results from Wang 2015 are being applied here in a new analysis.

R: We thank the reviewer for this helpful suggestion. We have added more explanations about the chemical analysis to clarify the difference between this study and the previous study by Wang et al. (2015). The paragraph has been rewritten as follows:

"The major water-soluble ions and trace elements in surface snow samples during this snow survey have already been investigated by Wang et al. (2015). However, the ILAPs in seasonal snow during this survey have not shown yet. For the importance of the ILAPs in snow, we will present the contribution and the emission sources of the ILAPs together with suites of other corresponding chemical constituents in seasonal snow. For instance, Hegg et al., (2009, 2010) analyzed the source attribution of the ILAPs in arctic snow by using a positive matrix factorization (PMF) model consisted with trajectory analysis and satellite fire maps. Briefly, major ions ($SO_4^{2-}$, $NO_3^-$, $Cl^-$, $Na^+$, $K^+$, and $NH_4^+$) were analyzed with an ion chromatograph (Dionex, Sunnyvale, CA), and trace elements of Fe and Al were measured by inductively coupled plasma mass spectrometry (ICP-MS). These analytical procedures have been described elsewhere (Yesubabu et al., 2014). In this paper, the major ions are used to retrieve the sea salt and biosmoke potassium ($K_{Biosmoke}^+$), which datasets were not shown by Wang et al. (2015). However, only $SO_4^{2-}$, $NO_3^-$, and $NH_4^+$ were reprinted from Wang et

al. (2015) to reveal the mass contribution of the ILAPs and the chemical constituents in seasonal snow during this snow survey."

Comment 20: Page 8 Line 11: The first sentence should be rewritten as its not clear to the reader how iron would originate from mineral dust in seasonal snow. Please explain in the text why this assumption is made.

R: As Alfaro et al, (2004) indicated that the light absorption by mineral dust should be highly sensitive to their content in iron oxides (hematite, goethite, etc.) based on Mie theory. Sokolik and Toon, (1999) also pointed out that computations performed with optical models show that the absorbing potential of mineral dust is more sensitive to the presence of strongly absorbing iron oxides such as hematite and goethite than to other minerals. Thus it is now possible to assess the absorption properties of mineral dust by using iron oxide content (Bond et al., 1999). In this study, the sampling sites were positioned 50 km from cities and at least 1 km upwind of approach roads or railways to prevent contamination.
For the above reason, the first sentence has been rewritten as "Recent studies indicated that the light absorption by MD should be highly sensitive to the presence of strongly absorbing iron oxides such as hematite and goethite than to other minerals (Alfaro et al., 2004; Sokolik and Toon, 1999).. Thus it is now possible to assess the absorption properties of MD by using iron oxide content (Bond et al., 1999). In this study, the iron (Hereinafter simply "Fe") in seasonal snow is assumed to be originating from MD during this survey."

Comment 21 Page 9 Line 5: Why is it stated that the Microtops II Sun Photometer was used instead of the CE318? Please explain the significance for stating this and the difference between the two. What does the Microtops II do that the CE318 does not?

R: We indicate that Microtops II and CE318 are both effective instruments on measuring aerosol optical depth (AOD) ( More et al., 2013; Porter et al., 2001; Zawadzka et al., 2014). However, the major difference between Microtops II and CE318 is that the Microtops II is portable for the field experiments, but CE318 is immovable. For it's a snow survey, it is better to use the Microtops II instrument instead of CE318 to measure aerosol optical depth.

Comment 22 Page 9 Line 14: include ' in this study'. . . during the 2014 Chinese survey. (If that is what the authors mean, otherwise please clarify).

R: Right, we have replaced this sentence as "To better understand the background weather conditions in the local atmosphere during this snow survey, we used a portable and reliable Microtops II Sun photometer at wavelengths of 340, 440, 675, 870, and 936 nm instead of the CE318 sun tracking photometer to measure the surface AOD in this study."

Comment 23 Page 9 Line 24: Spectroradiometers have been used in a number of other studies than listed here. Please include 'e.g.' and then the citations.

R: We have modified this sentence as "Snow albedo plays a key role in affecting the energy balance and climate in the cryosphere (e.g. Hadley and Kirchstetter, 2012; Liou et al., 2014; Warren and Wiscombe, 1985). Wright et al. (2014) indicated that

the spectral albedo measured by using an Analytical Spectral Devices (ASD) spectroradiometer at 350-2200 nm is in agreement with albedo measurements at the baseline Surface Radiation Network (BSRN). Wuttke et al. (2006a) pointed out that the spectroradiometer instrument is considered as the more capable, rapid, and mobile to conduct spectral albedo measurements during short time periods, especially in the very cold regions (e.g. in the Arctic). The major advantage is the more extensive wavelength range, and the cosine error is less than 5% for solar zenith angles below 85o at the wavelength of 320 nm (Wuttke et al., 2006a, b)."

Comment 24 Page 10 Line 5: How was the spectroradiometer held level? What was the field of view? How did you minimize shadows from the leveling device? What were the solar zenith angles? This data should all be included in the text and solar zenith angles should be presented in a table.

R: Normally the relative position of the sighting laser spot is at a distance of 1m from the optical element for the active field of view for the instrument in strict accordance with the user manual of the SVC HR-1024 spectroradiometer (Figure 6 Setup for FOV map). The nominal filed of view (FOV) lens is $8^o$ to enable the instrument to look at different size targets. The measured solar zenith angles and the other datasets used to simulate snow albedo have been labeled in Figure 11. The direction of the instrument was oriented to the Sun Horizon angles in order to receive more direct solar radiation. The small size of the fore optics greatly reduces errors associated with instrument self-shadowing. Even when the area viewed by the fore optic is outside the direct shadow of the instrument, the instrument still blocks some of the illumination (either diffuse skylight or light scattered off surrounding objects) that would normally be striking the surface under observation for measuring full-sky-irradiance throughout the entire 350-2500 nm wavelengths. This spectroradiometer is used for measuring the direct component of solar irradiance because of the minimized relative radiometric errors between total and direct irradiance measurements. For instance, Bi et al., (2013) used a set of broadband radiometers and sun/sky photometers during 2013 field campaign in the middle latitude across northern China to measured the direct and diffuse solar irradiance, and the result indicated that the diffuse solar radiation is lower than 10% compared with the total solar irradiance. Therefore, we indicated that the spectroradiometer in the clean sky condition mainly measured the direct solar irradiance during 2014 snow campaign. Therefore, the above paragraph was added to present the relative parameter of the spectroradiometer in section 2.4.

Comment 25 Page 11 Line 3: Is this the first presentation of the SAMDS model in publication? If so this needs to be highlighted more in the abstract, title, and especially the text here. Otherwise please include a citation for the SAMDS model.

R: Yes, the SAMDS model is the first time to be presented in open publication. So, we have addressed more information about the snow albedo reduction by the SAMDS model throughout the revised manuscript.

Comment 26: In general it seems this paper could be more focused around the SAMDS model as this appears to be the novelty of the paper.

R: We agree with the reviewer, therefore, we have changed the focus to investigating ILAPs, the internal/external of BC with snow, snow grain shapes in affecting the snow albedo reduction between surface measurements and model simulations.

Comment 27: Page 14 Line 13: This sentence is confusing and needs to be revised. For example: Delete 'For' and begin sentence with 'Most of the snow samples were collected in the afternoon corresponding to the Aqua-MODIS (13:30LT) overpass time. Then the next sentence starting with 'The averaged. . .'

R: The sentence has been revised as: "Most of the snow samples were collected in the afternoon at the Aqua-MODIS (13:30 LT) overpass time in order to compare the local AODs in sampling sites by using an spectroradiometer with the satellite remote sensing. The AOD spatial distribution derived from the Aqua-MODIS satellite over northern China associated with sampling site numbers is shown in Figure 1 during this snow survey."

Comment 28: Page 15 Lines 5 – 10. These sentences need to be restructured and concise.

R: These sentences have been restructured as: "Compared with the retrieved AOD by the remote sensing, the surface measurements of AOD were also conducted during this snow survey. Generally, the measured AOD were gradually higher from Inner Mongolia regions moved to the industrial area across northeastern China."

Comment 29: Page 15 Line 13: Remove 'processes'.

R: We have deleted "processes" as the reviewer's suggestion.

Comment 30: Page 15 Line 21: Please explain 'left and right' samples. Do the authors mean sample duplicates? This sentence needs restructuring.

R: Right. The sentence has been modified as "Two vertical profiles of snow samples ("left" and "right") were collected through the whole depth of the snowpack at all sites to reduce the possible contamination by artificial effects during the sampling process, and the dusty or polluted layers were separately collected during the sampling process.", and the caption of table 1 was added with "… All of the datasets are the average values from the left and right snow samples."

Comment 31 Page 15 Lines 21 - 25: Major run-on sentence, which needs revision.

R: The sentence has been modified as "In Inner Mongolia, the snow cover was thin and patchy. The average snow depth was less than 10 cm from sites 90, 91, 93, and 94, which was significantly lower than that near the northern border of China, ranging from 13 to 20 cm at sites 95-97. The snow samples were collected from drifted snow in Inner Mongolia, and the mass loadings of ILAPs in seasonal snow are mainly due to blowing soil dust. Therefore, the vertical profiles of snow samples mixed with blowing soil from these sites are insufficient to represent the seasonal evolution of wet and dry deposition to snow (Wang et al., 2013a). However, the light absorption of ILAPs is still dominated by OC in these regions, which has been illustrated in the following section."

Comment 32: Page 15 Lines 21 – 25: If the snowpack was so thin and patchy – won't blowing soil/dirt be an issue for the measurements? Please explain how this is accounted for.

R: We indicated that the mass loading of blowing soil could be the dominant factor in snow sampling in Inner Mongolia, when the snowpack was thin and patchy. However, the light absorption of ILAPs in snow is still dominated by OC in these regions.

Comment 33 Page 16: What type of snow was present? Fresh snow or old snow? Please explain using the international snow classification. Also, how soon were samples collected after snow falls? Also, it would be assumed that since the study was conducted in January, during Chinese winter, some sites could have fresh snow, where others could be 'older' snow?

R: We have already updated this information in Figure 1 and Table 1. The fresh snow was defined as the snow fell less than two days.

Comment 34 Page 16 Lines 14 - 19: Run-on sentence again. Please separate into two sentences. 'The results from the 2010 – 2014 Chinese snow surveys are shown in. . .. )

R: The sentence has been modified as "To better understand the distribution of $C_{BC}^{est}$ in seasonal snow across northern China, the spatial distribution of $C_{BC}^{est}$ in the surface and average snow measured during this snow survey are shown in Figure 4. The spatial distributions of $C_{BC}^{est}$ in the surface and average snow measured using the ISSW spectrophotometer during the 2014 survey generally ranged from 50 to 3700 ng g$^{-1}$ and 60 to 1600 ng g$^{-1}$, with the medium values of 260 ng g$^{-1}$, and 260 ng g$^{-1}$, respectively. These variations of $C_{BC}^{est}$ were very similar to those of the previous snow campaign by Wang et al. (2013a), however, much higher than those in the Xinjiang region of northwestern China (Ye et al., 2012), along the southern edge of the Tibetan Plateau (Cong et al., 2015), and across North America (Doherty et al., 2014)."

Table 1: Needs to be reorganized by dates collected (i.e. 2010 and 2014). Also include the number of samples incorporated in the average values. Sample depth is confusing. . . so the samples were collected over 5 cm and integrated into one measurement? What about the surface samples were less volume was collected at sites that were particularly dusty/polluted, i.e. Page 7 Line 4. Separate surface samples and snow pit data. Also, it appears

R: We need to clarify that the average snow depth in each site is measured 4 times in nearby locations. If the snow depth is thin and patchy, the snow samples could be collected from the drifted snow (e.g. Inner Mongolia regions), which is higher than the average snow depth in each site. So, we noted that the site average snow depth and the snow sampling depth for each site are both shown in Table 1. Therefore, the sentence has been modified as "Two vertical profiles of snow samples ("left" and "right") were collected through the whole depth of the snowpack at all sites to reduce the possible contamination by artificial effects during the sampling process, and we note that the dusty or polluted layers were separately collected during the sampling process. All of the datasets in seasonal snow from Table 1 are the average values from the two adjacent snow samples through the whole depth of the snowpack.".

Table 2: Include snowpack depth for each site, to see how snow pack depth varied from 2010-2014 as that may have influenced measured BC content. Also, given the method used, should BC = eBC? And was this 2010 data published previously? If so, include a citation. If the 2010 data has been published before, then it appears that there are only 12 new data points in this study from 2014. Why are sites 41 – 46 listed with sample dates but no avg BC value?

R: We agree with the reviewer that the datasets of surface and everaged $C_{BC}^{est}$ in snow in Table 2 has already been published by Wang et al. (2013a). Therefore, we deleted Table 2, and Table 4, and the manuscript mainly focused on the new results in 2014 snow survey.

Table 3: Wasn't this data publishes in Wang 2015, from sub section 2.2 Chemical Speciation Page 7 Line 13? If not, or if so clarify either way, provide references, and explain why the data is being presented again and what is new.

R: We note that only the datasets of $SO_4^{2-}$, $NO_3^-$, and $NH_4^+$ were published in Wang et al. (2015), but the other chemical speciation calculated by the water-soluble ion in section 2.2 was new. However, we indicated that the datasets of $SO_4^{2-}$, $NO_3^-$, and $NH_4^+$ were useful to show the attribution from different sources. We have added the following sentence in Table 2, and Figure 6 as "We noted that the datasets of $SO_4^{2-}$, $NO_3^-$, and $NH_4^+$ were reprinted from Wang et al. (2015).".

Figure 4: Site numbers should be included with the map and the 2014 data should only be presented since the 2010 data is already published? At least a distinction should be made between the 2010 and 2014 data. They should not be averaged together as there are many factors that would influence the differing values. Figure 5: These data do not appear to be normally distributed as they are heavily weighted on the low end, and just a couple larger concentrations. Also provide the p-value to show the significance of the fit.

R: We deleted Figure 4c, 4d, and Figure 6 to concentrate the novelty of this manuscript. We also provided the confidence test of the fit in Figure 5.

Figure 6: Again, like Figure 5, these data do not appear to be normally distributed as they are heavily weighted on the low end, and just a couple larger concentrations. Also provide the p-value to show the significance of the fit. Why is there one red dot at 0.25 AL and 1.5 K+?

R: We have deleted figure 6, and added a new figure in discussing the snow albedo reduction due to internal/external mixed BC in snow and difference snow grain shapes in Figure 10.

Figure 7: This is a nice figure. The land cover legend should be mentioned in the figure legend.

R: We have added the land cover legend in Figure 6 & Figure 7 as "The distribution of 17 different surface vegetation types retrieved from MODIS global land cover type product (MCD12C1) with 0.05 spatial resolution were used in this study."

Figure 8: As in Figure 7, the land cover legend should be mentioned in the figure legend.

R: See our answer to the above question.

Figure 9: Why is only the visible range presented? Please clarify in Figure legend.

R: We have already extended the spectral wavelengths from 400-1400 nm in Figure 8.

Comment 34 Page 16 Lines 14 - 19: The reader doesn't understand this sentence. Are these averaged values from the surface samples?

R: The sentence has been revised as "The spatial distributions of $C_{BC}^{est}$ in the surface and average snow measured using the ISSW spectrophotometer during the 2014 survey generally ranged from 50 to 3700 ng g$^{-1}$ and 60 to 1600 ng $^{g-1}$, with the medium values of 260 ng g$^{-1}$, and 260 ng g$^{-1}$, respectively.". Similar mistakes were corrected throughout the revised manuscript.

Comment 34 Page 16 Line 19: Include standard deviations with the ranges. How many samples were in this range?

R: We have changed the sentences as "The spatial distributions of $C_{BC}^{est}$ in the surface and average snow measured using the ISSW spectrophotometer during the 2014 survey generally ranged from 50 to 3700 ng g$^{-1}$ and 60 to 1600 ng $^{g-1}$, with the medium values of 260 ng g$^{-1}$, and 260 ng g$^{-1}$, respectively.".

Comment 35: Page 17 Line 5: Include a reference to the table where this data is displayed.

R: We have indicated that the datasets are from table 1.

Comment 36: Page 17 Line 11: Where is this regression value from? Reference figure.

R: The sentence has been revised as "The two results agreed very well ($R^2$=0.99), indicating that Equation (7) agreed well for this measurement in Figure 5.".

Comment 37: Page 18 Line 2: Pleas explain how photochemical reactions are related to biomass burning contributions of OC?

R: Actually, we didn't separate the contributions to light-absorbing OC from primary and secondary sources because our measurements do no provide such information. Therefore, the emission sources of the carbonaceous aerosols and the chemical species of Figure 6 have been deleted in the revised manuscript.

Comment 37: Page 18 Line 3: p-value? And please include for the other figures.

R: We have deleted Figure 6 and section 3.3 of emission factors. The confidence tests have been added in Figure 5 and their corresponsive results.

Comment 38 Page 19: The reader finds this paragraph confusing. Are the percentages from Figure 7? If the samples were collected in January 2014, why are there seasonal/time references? i.e. Line 11" Sulfate peaks were found in summer. . . this should be clearly tied to the Zhang 2013 reference. If the next sentence, line 12 – 17 (Which is a very long run-on sentence) also refers to Zhang 2013 then this needs to be more clearly stated. The combination of presenting results from this study and Zhang 2013 needs to be reorganized more clearly. For example a paragraph explaining Zhang 2013 could come first, and then compare this study to those. Also Zhang 2013 should probably be more clearly explained in the introduction. Again, it also appears that Wang 2015 already published snow chemistry data from these results? This needs to be more clearly referenced in the paper.

R: The results from this paragraph have been deleted, because this study will mainly focused on the ILAPs in seasonal snow, and its optical effects on snow albedo.

Comment 39 Page 20 Line 1: Include citation for iron originating from industrial emissions.

R: Two citations were added in this sentence as "however, Fe can also originate from industrial emissions, such as the metal and steel industries (Hegg et al., 2010; Ofosu et al., 2012)."

Comment 39 Page 20 Line 3: Is this sentence about 'this' study, or Doherty 2014?

R: The sentence means this study of 2014 snow survey.

Comment 39 Page 20 Line 6: 'Here, light absorption. . . '

R: We have modified this sentence.

Comment 40: Page 21 Line 4- 6: This has already been stated and should be in the methods.

R: The sentences have been moved to the method section as the reviewer's suggestion.

Comment 41 Page 21: The reader is not sure what the authors mean by 'Higher degrees of snow albedo. . .' do they mean more reduction or a high albedo? 'There was a larger reduction in snow albedo for. . ..''

R: The sentence has been revised as "A larger reduction in snow albedo by both BC and MD-contaminated snow was found for larger snow grains. . ."

Figure 10: Include labels for A, B, C. Same goes for all other figures.

R: Yes, we have replotted all of the figures in the revised manuscript, and labeled for (a), (b)… very clearly.

Comment 42 Page 22: Why is BC now refereed to as the ' BC mixing ratios'. Please clarify where this change in terminology was introduced.

R: We have unified all of mixing ratios and BC as $C_{BC}^{est}$, which are consistent with the previous studies by Doherty et al. (2010, 2015) and Wang et al. (2013a).

Comment 43: Overall, ILAPs is not used consistently in the paper as light absorbing aerosols is written out many times. The reader suggests to standardize this.

R: We have standardized the light absorbing aerosols as ILAPs throughout the revised manuscript.

Comment 44 Page 23 Lines 23 – 35: Please explain why, " . . .this study shows that the spectral albedo of snow reduction caused OC levels to increase?" Shouldn't the reduction in spectral albedo be a result of the OC levels?

R: This study indicated that the snow albedo reduction is not only correlated with the increased mixing ratios of OC in Figure 9c, but also highly correlated with the different snow grain sizes (e.g. snow grain size of 800 μm compared to 100 μm). Therefore, the sentence has been edited as "The radiative transfer modeling results presented by Zhang et al. (2016) and measurement results of this study show that the spectral albedo of snow reduction due to the increased OC mixing ratios (above 20 μg g$^{-1}$) is larger for a factor of 3 by assuming the snow grain size of 800 μm compared to 100 μm.".

Comment 45: The paper needs a lot of reorganization. For a 25 page paper, one page of Discussion does not seem adequate. Rename Subtitle 4, "Conclusions".

R: The conclusions have already renamed, and more discussion has been updated.-

Comment 46 Page 24 Line 4: Include the 13 sites also in the abstract. This description from lines 3 – 7 should be included in the site description and does not need to be repeated in the conclusion. Comment 47: Page 24 Line 22: Please state the range in latitudes covered in this study in the site description. Did the 13 sites cover a large latitudinal gradient?

R: The conclusion has been rewritten the same as comments 48.

Comment 48 Page 25 Line 13: The last sentence of the entire manuscript seems like an odd conclusion, 'to include OC as an input parameter in the SNICAR model'. Why not argue for the use of the SAMDS model? Does it not include OC as an input parameter? A better conclusion would be to compare the SAMDS and SNICAR models. Is the reason the SNICAR model is suggested to incorporated OC because it is widely used? Did the authors try incorporating OC as an input parameter in the off-line SNICAR code? The discussion and conclusion of this manuscript needs revision.

R: The conclusions were totally rewritten as follows:
"In this study, a snow survey was performed in January 2014, and 92 snow samples were collected at 13 sites across northern China. We found that higher AODs measured using a sun photometer and remote sensing devices showed that heavily

[revised manuscript text omitted]

~~This study offers not only an explanation for the discrepancy of the snow albedo reduction between modeled and observed snow albedo reduction due to ILAPs in snow, but also demonstrates the enhancement of the model simulations of the snow albedo reduction by using the optical effective radii ($R_{eff}$) of snow grains than that the measured snow grian radii due to SMDAS and SNICAR models, especially in the case of near-infrared wavelengths.A survey was performed to collect 92 seasonal snow samples at 13 sties across northern China in January 2014, and the mixing ratios of Insoluble Light Absorbing Particles (ILAPs, e.g. black carbon, organic carbon, and mineral dust (MD) in seasonal snow were measured by using an integrating sphere/integrating sandwich spectrophotometer (ISSW), and the chemical analysisBased on the surface measurements of ILAPs in snow, a new radiative transfer model (Spectral Albedo Model for Dirty Snow, or SAMDS) is developed to simulate the spectral albedo reduction due to ILAPs in snow based on the asymptotic radiative transfer theory. We calculate that XX XX %, and XX XX % of snow albedo reduction in surface snow resides within the concentrations of BC and~~

MD in the ranges of XX ng g, and XX ng g, respectively. A comparison between SAMDS and the SNICAR models indicated that the snow albedo can reduce 1-3%, 2-5%, and 2-4% due to ng g, XXX ng g, and xxx ng g of black carbon (BC), and mineral dust (MD) in snow. We note that tThe organic carbon (OC) is also another key parameter in affecting snow albedo of 1-5% due to XXX ppb in snow due to SDAMS model simulation. For a given shape (Koch snowflake, hexagonal plates/columns, and spheres), the snow albedo reduction due to spherical snow grains is graduately larger than Koch snowflake, and hexagonal plates/colums with the increased concentration of BC in snow. The internal mixing of BC in snow absorbs substantially higher than the external mixing at the wavelengths of 400 nm -1400 nm. In addition to the BC and AD parameters in the Snow, Ice, and Aerosol Radiation (SNICAR) model, the OC content in snow is considered an initial parameter for calculating snow albedo through a new radiative transfer model (Spectral Albedo Model for Dirty Snow, or SAMDS). The spectral albedo of snow reduction caused by OC (20 $\mu g\ g^{-1}$) is up to a factor of 3 for a snow grain size of 800 $\mu m$ compared to 100 $\mu m$. We find a larger difference in snow albedo levels between the model simulations and surface measurements for higher insoluble light absorbing impurities (ILAPs) using the measured snow grain radii. Compared with the observed snow albedo, we also note that the optical effective radii ($R_{eff}$) of snow grains can significantly enhance the model simulations of snow albedo reduction than that the measured snow grian radii due to SMDAS and SNICAR models, 
[revised manuscript text omitted]
 aggregates and snow grain shapes (fractal particles grains, hexagonal plates/columns, and spheres) with the snow albedo reduction due to and internal and /external mixing structures of BC and snow on snow albedo. Finally, conclusions are Concluding and discussing remarks are given in section 4. across northeastern China, which are highly correlated with industrial pollution resulting from human activity. Therefore, ILAPs in seasonal snow are examined during a snow campaign, and the snow albedo is measured using an

HR-1024 field spectroradiometer and simulated using two radiative transfer models (i.e., SNICAR and SAMDS).

**2    Experimental procedures**

**2.1    Snow field campaign in January 2014 January**

In 2014, there was less snowfall in January than in previous years (e.g., 2010), and only 92 snow samples (13 surface snow, and 79 sub-surface snow samples) at 13 sites were collected during this snow survey. There were 10 fresh snow, and 3 aged snow for surface snow. In 2014, there was less snowfall in January than in previous years (e.g., 2010), and only 92 snow samples at 13 sites were collected. The snow sampling sites in this study were began at 90 (see Figure 1 and Table 1), which are numbered in chronological order followed by Wang et al. (2013a), and Ye et al. (2012). Samples from sites 90-93 were collected from grassland and cropland areas in Inner Mongolia. Sites 94-98 and sites 99-102 were located in the Heilongjiang and Jilin provinces, respectively, which are the most heavily polluted areas in northern China during winter. The snow sampling routes were similar to those used in the previous survey conducted in 2010 across northern China (Huang et al., 2011). To prevent contamination, the sampling sites were positioned 50 km from cities and at least 1 km upwind of approach roads or railways; the only exception was site 101, which was positioned downwind and close to villages. Two vertical profiles of snow samples ("left" and "right") were collected through the whole depth of the snowpack at all sites to reduce the possible contamination by artificial effects during the sampling process, and we note that the dusty or polluted layers were separately collected during the sampling process. We gathered the "left" and "right" snow samples at vertical intervals of snow samples every 5 cm for each layer from the surface to the bottom

unless a particularly dusty or polluted layer was present. All of the datasets of $C_{BC}^{equiv}$, $C_{BC}^{equiv}$, and $C_{BC}^{est}$ in seasonal snow from Ttable 1 
[revised manuscript text omitted]
 \text{\color{red}{\sout{C}}}_{\sout{s}} C_{OC}^* \cdot \frac{3}{7}(1+2\cos v_0))}\quad\text{, for}$$

$$\exp\left(- \sqrt{94.746 \cdot \frac{r_{eff}}{\lambda} \cdot k(\lambda) + 5.163 \cdot r_{eff} \cdot \left(MAC_{abs}^{\text{\color{red}{BC}}} \cdot C_{BC}^* + MAC_{abs}^{\text{\color{red}{dustMD}}} \cdot C_{\text{\color{red}{dustMD}}}^* + MAC_{abs}^{OC} \cdot C_{OC}^*\right) \cdot (1+2\cos v_0)}\right) ,\quad\text{for spherical grains;}$$

$$= \quad \exp\left(-4.95 \cdot \sqrt{\frac{\pi \cdot r_{eff} \cdot \sout{\not p}(k(\lambda) + \alpha \cdot C_{\sout{dust}BC}^* + \beta \cdot C_{\sout{BC}MD}^* + \chi \cdot C_{OC}^*)}{\lambda}} \cdot (1+2\cos v_0)\right) ,\quad\text{for hexagonal grains;}$$

(15)

$$\exp\left(-4.38 \cdot \sqrt{\frac{\pi \cdot r_{eff} \cdot \sout{\not p}(k(\lambda) + \alpha \cdot C_{\sout{dust}BC}^* + \beta \cdot C_{\sout{BC}
[revised manuscript text omitted]
$ the measured grain size. We innovatively suppose , indicating that, for the same snow grains, the radiative perturbation of ILAPslight-absorbing impurities was are amplified able to enhance with the $R_{eff}$ in spite of the same $R_m$snow grain optical effective size. Nevertheless, due to the limited measurements of snow albedo, this supposition is quite uncertain and needed to be verified by numerous field measured snow albedos. As indicated in Figure 1Combinations of the results of Figure 9 and Figure 112+, that BC, OC, and ADMD are three types of ILAPs found in snow that can reduce spectral snow albedo levels and. rReduced snow albedo due to ILAPs in our measurements were generally comparable to the modeled effectsthat found in the commonly used SAMDS SNICAR and commonly used SNICAR SAMDS models (Flanner et al., 2007; Zhang et al., 2016) when the mixing ratios of ILAPs are not quite high. , Thereforewe indicate that, 100-500 ng g$^{-1}$ of BC can lower the snow albedo by 0.014Z-0.0396% relative to pure snow with a snow grain size of 1Z00 μm according to our snow field campaign, and ADMD was found to be a weak absorber owing due to its lower ADMD MAC, supporting previous observations made by Warren and Wiscombe (1980). The OC MAC iswas also lower and comparable to that of the ADMD. A clear decreasing trend in the surface snow albedo owing due to the high ambient mixing ratios of OC from Inner Mongolia to northeastern China was found.

The radiative transfer modeling results presented by Zhang et al. (2016) and measurement results of this study show that the spectral albedo of snow reduction due to the increased OC mixing ratios concentration (above 20 μg g$^{-1}$) is larger for a factor of 3 by assuming the snow grain size of 800 μm compared to 100 μm.

**4     The radiative transfer modeling results presented by Zhang et al. (2016) and**

**Conclusions**

In this study, a snow survey was performed in January 2014, and 92 snow samples were collected at 13 sites across northern China. We found that higher AODs measured using a sun photometer and remote sensing devices showed that heavily polluted areas remain in industrial regions across northern China. The measured $C_{BC}^{est}$  through the 2014 survey via the ISSW spectrophotometer in surface and average snow of 50 to 36700 and 60 to 1600 ng g⁻¹, with the medium values of 260 ng g⁻¹, and 260 ng g⁻¹, respectively, were much larger than those of previous snow field campaigns. The chemical composition analysis showed that the mass contributions in seasonal snow was dominated by OC and MD. However, assuming the MACs for BC, OC, and Fe are 6.3, 0.3, and 0.9 m² g⁻¹, respectively, at 550 nm, light absorption was still dominated by BC and OC in seasonal snow during the entire campaign. The light-absorbing contribution  of the MD  was larger at high latitudes than at low latitudes due to strong winds transporting snow. Then,

~~In this study, a Chinese survey was performed in January 2014. We collected 92 snow samples from 13 sites across northern China. Much less snow had fallen than in previous years; as a result, most of the surface snow samples were collected as aged snow, and snow grain sizes were much larger owing to solar radiation absorbed by ILAPs as a result of settlement processes. Although we selected study locations in~~

remote regions located at least 50 km from cities, higher AODs measured using a sun photometer and remote sensing devices showed that heavily polluted areas remain in industrial regions across northern China. The estimated BC mixing ratios measured through the 2014 survey via the ISSW spectrophotometer in surface and average integrated snow of 53 to 3651 and 62 to 1635 ng g$^{-1}$, respectively, were much larger than those of previous snow field campaigns. The non-BC fraction showed that most of the ILAPs in seasonal snow were dominated by OC and AD. Owing to BC and OC mass absorption efficiencies of 6.3 and 0.3 m$^2$ g$^{-1}$, respectively, at 550 nm, light absorption was still dominated by BC and OC in seasonal snow during the entire campaign. AD contributions in snow to light absorption amounted to less than 10%. The large OC/BC ratios and correlation coefficients indicated that these contributions were mainly derived from common sources (e.g., biomass burning). Similarly, NH$_4^+$ was attributed to intense agricultural activity compared to industrial emissions of SO$_4^{2-}$ and NO$_3^-$.

The fraction of the AD mixing ratios was larger at high latitudes than at low latitudes owing to strong winds transporting snow. In this study, we indicated that the present a new spectral snow albedo model (SAMDS) for simulating the surface albedo of snow with deposited ILAPs aerosol impurities (e.g. Black carbon, Organic carbon, Mineral dust, volcano ash, and snow algae) by using the asymptotic analytical radiative transfer theory. Given the measured BC, MD and OC mixing ratios of 100-5000 ng g$^{-1}$, 2000-6000 ng g$^{-1}$, and 1000-30000 ng g$^{-1}$ in surface snow across northeastern China, we ran the models at a solar zenith angle $\theta$ of 60°, and the results indicated that the albedo of fresh snow at 550 nm is generally in a range of 0.95-0.75 with R$_{eff}$ of 100 μm. This model can also be used to investigate the snow albedo influenced by the internal/external mixing of BC and snow with impurities, irregular morphology of

snow grains  ,  and the vertical distribution of snow grains and impurities for multilayer snow. ~~Additionally, the properties of different snow grain shapes (Fractal particles, Hexagonal plate/column, and spheres) and the internal/external mixing with BC in snow by using SAMDS model might be useful to researchers who are conducting studies involving ILAPs and snow interaction and feed back in snow albedo reduciton. Compare to the SNICAR model, the snow albedo reduction is in agreement with the SAMDS model, different types of impurity could be included in the parameterization in SAMDS model, such as organic carbon and biogenic particles.thea given shape (spheres, hexagonal plates/columns, and fractal particles), it shows thatby 0.017-0.073, and 0.008-0.036as a function of BC mixing ratios (0-5000 ng g$^{-1}$)it shows that snow albedo by spherical snow grains typically decrease by 0.017-0.073, and 0.008-0.036 as a function of BC mixing ratios (0-5000 ng g$^{-1}$), which is compared with the fractal snow grains and hexagonal plates/columns snow grains.snow absorbs substantially more light than external mixing subsequently~~. For fresh snow grains of hexagonal plates/columns with $R_{eff}$ of 100 μm, the difference of snow albedo between internal and external mixing of BC and snow is up to 0.036 for 3000 ng g$^{-1}$ BC in snow in the heavy industrial regions across northeastern China, whereas by low to 0.005 for 100 ng g$^{-1}$ BC in snow in the further north China near the border of Siberia. The spectral albedo of snow reduction caused by OC (20 μg g$^{-1}$) is larger by up to a factor of 3 for a snow grain size of 800 μm compared to 100 μm by using SAMDS model.

A comparison between measured and simulated snow albedos was conducted. The snow albedos measured from a spectroradiometer and simulated using the SNICAR and SAMDS models agreed well  at the lower mixing ratios of BC, MD, and MDOC. However, a large discrepancy in snow albedo  between the model simulations and surface measurements for heavy loading of ILAPs in snow was found by using R$_m$. We demonstrate that the simulated snow albedo reduction by SMDAS and SNICAR models is significantly enhanced by using R$_{eff}$ of snow grains compared with R$_m$, especially in the case of near-infrared wavelengths.Based on the snow grain optical effective size,  a remarkable improvement of the snow albedo reduction  to present the snow albedo reduction  Although the MAC of OC is much lower than that of BC, we found that OC  was a major absorber in snow  due to its high mixing ratio  from human activities occurring across northeastern China. Moreover, 5000 ng g$^{-1}$ of OC was found to reduce the snow albedo by 0.016-0.059 depending on the snow grain size and aging period. Therefore, we suggest that the mixing ratio of OC should be added as an input parameter to the SNICAR model for determining snow albedos. Although the SAMDS model might be useful to researchers who are conducting studies involving ILAPs and snow interaction and feedback in snow albedo change,

for dirty snow due to ILAPs, and multiple internal/external mixing stats of BC associated with irregular snow grains, we indicate that further snow surveys across northern China should be performed for the following reasons:. First(1), large variations of ILAPs in seasonal snow across northern China can lead higher uncertainties of snow albedo reduction, especially in the industrial regions,. and (2)Second, we only measured the snow albedo at 6 sampling sites by using the spectroradiometer in the clear sky condition due to much less snow fallen in January 2014 than that in previous years. Comparinge model simulations with the observations, we found that $R_{eff}$the optical effective snow grains could seemingly be enhanced by the high concentrationsmixing ratios of ILAPs in snow, however, we note that further snow surveys on measuring snow albedo should be conducted to reveal this phenomenon. Finally, there are large uncertainties in affectingsimulating snow albedo reduction and radiative forcing due to the ILAPs mixed with snow/ice and the irregular morphology of snow grains, the potential snow albedo reductionchange for aged snow should be investigated in the following snow surveys accordingly to test the capability of SAMDS model, which will provide more valuable and useful information for the climate models.

[revised manuscript text omitted]

| Site | Layer | Latitude N | Longitude E | Snow type | Site average snow depth (cm) | Sample depth (cm) Top | Sample depth (cm) Bottom | Temperature (°C) Snow | Temperature (°C) Bottom | Snow density (g/cm³) | grain $R_m$ (mm) | $C_{BC}^{equiv}$ (ng g$^{-1}$) | $C_{BC}^{max}$ (ng g$^{-1}$) | $C_{BC}^{est}$ (ng g$^{-1}$) | $f_{non\text{-}BC}^{est}$ (%) | $\text{Å}_{tot}$ 450:600 nm |
|---|---|---|---|---|---|---|---|---|---|---|---|---|---|---|---|---|
| 1 | 4 | | | Aged | | 13 | 18 | -8 | | 0.28 | 0.6 | 1900 | 1100 | 770 (230, 1000) | 60 (48, 88) | 3.0 |
| | 5 | | | Aged | | 18 | 23 | -7 | | 0.32 | 1 | 1100 | 930 | 450 (200, 5??) | 60 (47, 82) | 2.7 |
| 06 | 6 | 45°02'44" | 116°22'45" | Aged | | 23 | 28 | -6 | | 0.27 | 1.2 | 1500 | 1300 | 900 (300, 1200) | 38 (21, 80) | 2.9 |
| 16 | 7 | 50°02'48" | 124°22'41" | Aged | | 28 | 33 | -5 | | 0.26 | 1.2 | 860 | 490 | 340 (130, 400) | 66 (53, 87) | 2.9 |
| | 8 | | | Aged | | 33 | 38 | -5 | | 0.3 | 1 | 430 | 200 | 110 (30, 160) | 74 (61, 95) | 3.2 |
| 02 | 9 | 50°39'07" | 122°23'53" | Aged | | 38 | 43 | -4 | | 0.29 | 0.8 | 520 | 250 | 160 (70, 230) | 71 (58, 87) | 2.8 |

| Group | Sub | Latitude | Longitude | | | | | | | | | | | |
|---|---|---|---|---|---|---|---|---|---|---|---|---|---|---|
| 93 | 2 | 50°24'50" | 124°54'20" | 8 | 5 | 10 | 16 | 0.18 | 1.25 | 108 | 97 | 77 (49,97) | 29 (11,55) | 2.2 |
|  | 1 |  |  |  | 0 | 1 | – | – | 0.07 | 172 | 97 | 70 (39,90) | 53 (39,73) | 2.5 |
| 94 | 2 | 50°09'05" | 125°46'06" | 8 | 2 | 6 | 21 | 0.21 | 0.175 | 313 | 136 | 100 (46,128) | 68 (50,85) | 2.7 |
| 95 | 1 | 50°54'43" | 127°04'50" | 18 | 0 | 1 | 24 | – | 0.1 | 1173 | 404 | 260 (108,363) | 78 (69,91) | 2.8 |
|  | 1 |  |  |  | 0 | 3 | 22 | 0.23 | 0.175 | 278 | 110 | 75 (37,100) | 73 (64,86) | 2.6 |
|  | 2 |  |  |  | 3 | 8 | 23 | 0.18 | 0.9 | 230 | 83 | 54 (25,73) | 77 (68,89) | 2.7 |
|  | 3 |  |  |  | 8 | 13 | 25 | 0.27 | 1 | 94 | 50 | 27 (7,40) | 71 (55,92) | 3.2 |
|  | 4 |  |  |  | 14 | 18 | 25 | 0.14 | 1.3 | 524 | 308 | 131 (–,237) | 75 (54,148) | 4.8 |
| 96 | 1 | 49°47'41" | 126°43'13" | 13 | 0 | 5 | 22 | 0.37 | 0.08 | 1340 | 585 | 241 (21,332) | 83 (76,99) | 3.4 |
|  | 2 |  |  |  | 5 | 10 | 23 | 0.55 | 0.6 | 234 | 58 | 45 (6,74) | 81 (69,98) | 3.3 |
|  | 3 |  |  |  | 10 | 14 | 22 | 0.22 | 1 | 377 | 126 | 78 (26,111) | 80 (71,93) | 3.0 |
| 97 | 1 | 47°39'18" | 131°13'0" | 20 | 0 | 5 | 22 | 0.36 | 0.2 | 1005 | 362 | 236 (93,306) | 76 (69,94) | 2.8 |
|  | 2 |  |  |  | 5 | 10 | 18 | 0.27 | 1.1 | 160 | 79 | 45 (15,65) | 44 (15,77) | 3.0 |
|  | 3 |  |  |  | 10 | 15 | 14 | 0.32 | 0.6 | 32 | 23 | 9 (3,18) | 74 (44,90) | 3.1 |
| 98 | 1 | 45°25'38" | 130°58'55" | 35 | 0 | 5 | 20 | 0.22 | 0.09 | 2154 | 2080 | 1461 (754,1818) | 32 (16,65) | 2.5 |
|  | 2 |  |  |  | 5 | 10 | 23 | 0.24 | 0.5 | 641 | 349 | 241 (123,307) | 58 (46,79) | 3.6 |
|  | 3 |  |  |  | 10 | 14 | 14 | 0.24 | 0.75 | 97 | 55 | 25 (10,45) | 74 (52,90) | 3.4 |
|  | 4 |  |  |  | 15 | 20 | 14 | 0.26 | 0.75 | 64 | 36 | 46 (7,31) | 75 (48,94) | 3.4 |
|  | 5 |  |  |  | 20 | 25 | 14 | 0.28 | 1.1 | 432 | 264 | 166 (62,221) | 62 (49,86) | 2.8 |
|  | 6 |  |  |  | 25 | 30 | 11 | 0.69 | 1.3 | 848 | 165 | 112 (66,148) | 68 (58,84) | 2.5 |
|  | 7 |  |  |  | 30 | 35 | – | 0.33 | 1.3 | 288 | 118 | 72 (37,104) | 75 (65,87) | 2.6 |
| 99 | 1 | 43°36'01" | 125°42'04" | 14 | 0 | 5 | 2 | 0.18 | 0.085 | 1459 | 922 | 936 (235,823) | 54 (40,84) | 2.8 |
|  | 2 |  |  |  | 5 | 12 | 3 | 0.23 | 1.1 | 5847 | 3778 | 2882 (1549,3559) | 51 (39,74) | 2.5 |
|  | 3 |  |  |  | 12 | 18 | 4 | 0.28 | 1.1 | 2795 | 1360 | 1059 (630,1306) | 62 (53,77) | 2.3 |
|  | 4 |  |  |  | 18 | 24 | 4 | 0.28 | 0.7 | 2027 | 1518 | 1105 (537,1384) | 46 (32,73) | 2.6 |

| | | | | | | | | | | | | | | |
|---|---|---|---|---|---|---|---|---|---|---|---|---|---|---|
| 100 | 1 | 43°30'45" | 127°15'34" | 18 | 0 | 2.5 | — | 0.14 | 0.075 | 1739 | 703 | 508 (250, 651) | 71 (63, 98) | 2.6 |
| | 2 | | | | 2.5 | 4 | 2 | 0.18 | 0.25 | 4126 | 2891 | 1729 (—, 2371) | 58 (43, 103) | 3.6 |
| | 3 | | | | 4 | 6 | 2 | 0.23 | 0.9 | 1092 | 564 | 414 (208, 529) | 60 (49, 80) | 2.6 |
| | 4 | | | | 6 | 15 | 3 | 0.24 | 0.8 | 1601 | 728 | 541 (266, 691) | 66 (57, 83) | 2.6 |
| 101 | 1 | 43°47'25" | 125°46'08" | 22 | 0 | 5 | 19 | 0.24 | 0.07 | 5634 | 5070 | 3651 (1333, 4644) | 35 (18, 76) | 2.9 |
| | 2 | | | | 5 | 10 | 17 | 0.24 | 0.9 | 4318 | 3826 | 2575 (882, 3284) | 40 (24, 79) | 2.9 |
| | 3 | | | | 10 | 15 | 15 | 0.25 | 1.1 | 657 | 389 | 313 (206, 386) | 46 (33, 63) | 3.4 |
| | 4 | | | | 15 | 20 | 14 | 0.38 | 1.3 | 297 | 254 | 188 (108, 239) | 39 (21, 67) | 3.5 |
| 102 | 1 | 42°12'34" | 126°37'55" | 46 | 0 | 3 | 8 | 0.14 | 0.07 | 2109 | 1737 | 1357 (808, 1755) | 37 (19, 61) | 3.3 |
| | 2 | | | | 3 | 8 | 8 | 0.36 | 0.1 | 594 | 377 | 279 (149, 371) | 53 (37, 75) | 2.6 |
| | 3 | | | | 8 | 13 | 8 | 0.24 | 0.3 | 2513 | 2088 | 1623 (425, 2120) | 35 (15, 81) | 3.4 |
| | 4 | | | | 13 | 18 | 8 | 0.38 | 0.6 | 1622 | 1073 | 773 (229, 1013) | 60 (48, 88) | 3.9 |
| | 5 | | | | 18 | 23 | 7 | 0.32 | 1 | 1112 | 629 | 450 (197, 590) | 60 (47, 82) | 2.7 |
| | 6 | | | | 23 | 28 | 6 | 0.27 | 1.2 | 1466 | 1281 | 903 (300, 1167) | 38 (21, 80) | 2.9 |
| | 7 | | | | 28 | 33 | 5 | 0.26 | 1.2 | 858 | 493 | 344 (133, 457) | 66 (53, 87) | 2.9 |
| | 8 | | | | 33 | 38 | 5 | 0.3 | 1 | 426 | 197 | 109 (27, 163) | 74 (61, 95) | 3.2 |
| | 9 | | | | 38 | 43 | 4 | 0.29 | 0.8 | 524 | 245 | 157 (67, 220) | 71 (58, 87) | 2.8 |

**Table 2.** Estimates of integrated snowpack BC content in seasonal snow in the study sites for 2010 and 2014. (change as surface and average BC)

| Site | Date sampled (2014) | | Snowpack average Integrated BC (ng g$^{-1}$) |
|---|---|---|---|
| 90 | 10 Jan | | 334 |
| 91 | 13 Jan | | 161 |
| 92 | 14 Jan | | 65 |
| 93 | 15 Jan | | 94 |
| 94 | 15 Jan | | 260 |
| 95 | 16 Jan | | 62 |
| 96 | 17 Jan | | 140 |
| 97 | 18 Jan | | 105 |
| 98 | 19 Jan | | 264 |
| 99 | 23 Jan | | 1507 |
| 100 | 24 Jan | | 592 |
| 101 | 26 Jan | | 1635 |
| 102 | 27 Jan | | 583 |

|  |  |  |
|---|---|---|
|  |  | 53 |
|  |  |  |
|  |  |  |
|  |  |  |
|  |  |  |
|  |  |  |

**Table 23.** Chemical species (ng g$^{-1}$) in surface snow for sites across northeastern China in January 2014. The datasets of SO$_4^{2-}$, NO$_3^-$, and NH$_4^+$ were reprinted from Wang et al. (2015).

| Site | MD | BC | OC | K$^+_{biosmoke}$ | SO$_4^{2-}$ | NO$_3^-$ | NH$_4^+$ | Sea salt |
|------|------|------|-------|------|-------|------|------|------|
| 90 | 1900 | 380 | 6700 | 327 | 1685 | 213 | 22 | 868 |
| 91 | 1700 | 180 | 590 | 179 | 853 | 465 | 36 | 827 |
| 92 | 1300 | 60 | 280 | 150 | 511 | 105 | 19 | 456 |
| 93 | 1700 | 80 | 450 | 213 | 718 | 387 | 90 | 960 |
| 94 | 3300 | 300 | 2700 | 118 | 1335 | 550 | 28 | 554 |
| 95 | 2000 | 90 | 600 | 164 | 587 | 523 | 39 | 669 |
| 96 | 2300 | 280 | 3900 | 309 | 1285 | 493 | 91 | 1227 |
| 97 | 2400 | 280 | 2400 | 173 | 1163 | 407 | 38 | 753 |
| 98 | 3900 | 1600 | 13300 | 633 | 3096 | 747 | 195 | 2516 |
| 99 | 3000 | 770 | 4700 | 372 | 3379 | 1492 | 155 | 2310 |
| 100 | 3800 | 570 | 4000 | 260 | 4237 | 2258 | 487 | 2195 |
| 101 | 3500 | 4200 | 32000 | 1337 | 12382 | 2364 | – | 5131 |
| 102 | 5800 | 1700 | 2400 | 488 | 8034 | 3631 | 769 | 4420 |

|  |  |  |  |
|------|------|------|------|
|  |  |  |  |
|  |  |  |  |
|  |  |  |  |
|  |  |  |  |
|  |  |  |  |
|  |  |  |  |

[Figure]

**Fig. 1.** Spatial distribution of the averaged AOD retrieved from Aqua-MODIS over northern China from October 2013 to January 2014.; Tthe red  dots are MODIS active fire locations,; the black dots are the sampling locations. The site

numbers beginning at 90 in this study are numbered in chronological order followed by Wang et al. (2013a), and Ye et al. (2012). The "A" and "F" refer to aged snow and fresh snow, respectively.

[Figure]

**Fig. 2.** The variation in AOD at 500 nm at different sites measured using a Microtops Π Sun photometer over northeastern China in January 2014.

[Figure]

**Fig. 3.** Vertical temperature, snow density, and measured snow grain radius $(R_m)$ profiles at each site during the 2014 Chinese snow survey.

[Figure]

[Figure]

**Fig. 4.** The spatial distribution of $C_{BC}^{est}$ in the (a) surface and (b) average snow in 2014  across northeastern China.

[Figure]

**Fig. 5.** Comparisons between the calculated and optically measured $C_{BC}^{est}$  in surface snow during 2010 and 2014 snow surveys. The datasets of

measured $C_{BC}^{est}$ in 2010 from sites 3-40 were reprinted from Wang et al. (2013a).

**Fig. 6.** Ratios of OC and BC, $NH_4^+$ and $SO_4^{2-}$, $NH_4^+$ and $NO_3^-$, and $K^+$ and Al in surface snow in January 2014.

[Figure]

[Figure]

**Fig. 6̶7̶. MD.** The major components include A̶D̶MD, BC, OC, $K^+_{Biosmoke}$ b̶i̶o̶m̶a̶s̶s̶ ̶s̶m̶o̶k̶e̶ ̶p̶o̶t̶a̶s̶s̶i̶u̶m̶, secondary a̶e̶r̶o̶s̶o̶l̶ ions ($SO_4^{2-}$, $NO_3^-$, and $NH_4^+$ s̶u̶l̶f̶a̶t̶e̶,̶ ̶n̶i̶t̶r̶a̶t̶e̶,̶ ̶a̶n̶d̶ ̶a̶m̶m̶o̶n̶i̶u̶m̶), and sea salt in the surface snow samples collected in January 2014. The distribution of 17 different surface vegetation types retrieved from MODIS global land cover type product (MCD12C1) with 0.05 spatial resolution w̶e̶r̶e̶was used in this study. The datasets of $SO_4^{2-}$, $NO_3^-$, and $NH_4^+$ were reprinted fromo̶r̶i̶g̶i̶n̶a̶t̶e̶d̶ ̶f̶r̶o̶m̶ Wang et al. (2015).

[Figure]

Absorption (%)
■ BC
■ OC
■ AD

0 Water                5 Mixed Forest        10 Grasslands          15 Snow and Ice
1 Evergreen Needleleaf  6 Closed Shrublands   11 Permanent Wetlands   16 Bare orsparesly vegetate
2 Evergreen Broadleaf   7 Open Shrublands     12 Croplands            254 Unclassified
3 Deciduous Needleleaf  8 Woody Savannas      13 Urban and Built-up
4 Deciduous Broadleaf   9 Savannas            14 Cropland Mosaics

[Figure]

Absorption (%)
■ BC
■ OC
■ MD

0 Water                5 Mixed Forest        10 Grasslands          15 Snow and Ice
1 Evergreen Needleleaf  6 Closed Shrublands   11 Permanent Wetlands   16 Bare orsparesly vegetate
2 Evergreen Broadleaf   7 Open Shrublands     12 Croplands            254 Unclassified
3 Deciduous Needleleaf  8 Woody Savannas      13 Urban and Built-up
4 Deciduous Broadleaf   9 Savannas            14 Cropland Mosaics

[revised manuscript text omitted]

---

## Author Response (AR3)

**Response to Co-Editor**

We greatly appreciate this editor's critical comments and suggestions, which have helped us improve the paper quality substantially. We have addressed all of the comments carefully as detailed below in our point-by-point responses. Our responses start with "R:".

Comments to the Author:

Please further revise the paper considering the following comments:

Introduction section is too long. Many materials are not direly related to the topic of this study. Reorganize this section in the following way: First briefly discuss ILAPS roles in climate, especially ILAPS in snow. Then discuss what studies have been conducted on this specific topic, such as available field measurements and modeling treatments of these particles, using summarizing form, not listed by one study after another. Then point out knowledge gaps on this topic and why the present study is needed. And finally point out the goals of the present study.

R: We have simplified the introduction section, which became more related to the topic of this study based on the comments from the editor.

Avoid repetition or redundancy wherever possible. Some materials provided for reviewers' information (that have been posted as response to reviewers 'comments) do not necessarily be presented in the final version of the paper if these materials do not add much scientific value to the paper.

R: We have revised the final manuscript avoiding repetition and redundancy based on the comments from the editor.

Several paragraphs are too long to follow easily. Split into short ones for easy reading. Some very long sentences do not have the clean meaning and need to be fixed.

R: We have revised the long paragraphs into short ones.

Conclusion section needs to be polished. Avoid repeating statements that are already in the abstract.

R: We have revised the conclusion section more accurately to better reflect the topic of this study.

Polish the language and remove grammar issues.

R: We have made major revisions to polish the language and remove grammar issues of the manuscript.

**Observations and model simulations of sSnow albedo reduction in seasonal snow due to insoluble light-absorbing particles during 2014 Chinese survey anthropogenic dust and carbonaceous aerosols across northern China**

Xin Wang[1], Wei Pu[1], Yong Ren[1], Xuelei Zhang[2], Xueying Zhang[1], Jinsen Shi[1], Hongchun Jin[1],

Mingkai Dai[1], Quanliang Chen[3]

[1] Key Laboratory for Semi-Arid Climate Change of the Ministry of Education, College of Atmospheric Sciences, Lanzhou University, Lanzhou, 730000, China
[2] Key Laboratory of Wetland Ecology and Environment, Northeast Institute of Geography and Agroecology, Chinese Academy of Sciences, Changchun 130102, China
[3] College of Atmospheric Science, Chengdu University of Information Technology, and Plateau Atmospheric and Environment Laboratory of Sichuan Province, Chengdu 610225, China

Correspondence to: X. Wang (wxin@lzu.edu.cn)

**Abstract**.

A snow survey was carried out to collect 13 surface snow samples (10 for fresh snow, and 3 for aged snow) and 79 sub-surface snow samples in seasonal snow at 13 sites in January 2014 across northeastern China. A spectrophotometer combined with chemical analysis was used to quantify snow particulate absorption by insoluble light-absorbing particles (ILAPs, e.g. black carbon, BC; mineral dust, MD; and organic carbon, OC) in snow.Snow albedo was measured by using a field spectroradiometer. A new radiative transfer model (Spectral Albedo Model for Dirty Snow, or SAMDS) was then developed to simulate the spectral albedo in snow based on the asymptotic radiative transfer theory. A comparison between SAMDS and an existing model - the Snow, Ice, and Aerosol Radiation (SNICAR) indicates that good agreements in the model simulated spectral albedos of pure snow, however, the SNICAR model values tended to be slightly lower than those of SAMDS when BC and MD were considered. Given the measured BC, MD and OC mixing ratios of 100-5000, 2000-6000, and 1000-30000 ng g$^{-1}$, respectively, in surface snow across northeastern China, SAMDS model produced a snow albedo in the range of 0.95-0.75 for fresh snow at 550 nm with a snow grain optical effective radius ($R_{eff}$) of 100 μm. The snow albedo reduction due to spherical snow grains assumed asaged snow is gradually larger than fresh snow such as fractal snow grains, and hexagonal plates/columns snow grains associated with the increased BC in snow. For typical

BC mixing ratios of 100 ng g$^{-1}$ in remote areas and 3000 ng g$^{-1}$ in heavy industrial areas across northern China,  the snow albedo for –internal mixing of BC and snow is lower by 0.005 and 0.036 than that of external mixing for hexagonal plates/columns snow grains with R$_{eff}$ of 100 μm.  The result also shows that the simulated snow albedos by both SAMDS and SNICAR agree well with the observed values at low ILAPs mixing ratios, but tend to be higher than surface observations at high ILAPs mixing ratios.

~~This study offers not only an explanation for the discrepancy of the snow albedo reduction between modeled and observed snow albedo reduction due to ILAPs in snow, but also demonstrates the enhancement of the model simulations of the snow albedo reduction by using the optical effective radii (R$_{eff}$) of snow grains than that the measured snow grian radii due to SMDAS and SNICAR models, especially in the case of near-infrared wavelengths.A survey was performed to collect 92 seasonal snow samples at 13 sties across northern China in January 2014, and the mixing ratios of Insoluble Light Absorbing Particles (ILAPs, e.g. black carbon, organic carbon, and mineral dust (MD) in seasonal snow were measured by using an integrating sphere/integrating sandwich spectrophotometer (ISSW), and the chemical analysisBased on the surface measurements of ILAPs in snow, a new~~

radiative transfer model (Spectral Albedo Model for Dirty Snow, or SAMDS) is developed to simulate the spectral albedo reduction due to ILAPs in snow based on the asymptotic radiative transfer theory. We calculate that XX-XX %, and XX-XX % of snow albedo reduction in surface snow resides within the concentrations of BC and MD in the ranges of XX ng g, and XX ng g, respectively. A comparison between SAMDS and the SNICAR models indicated that the snow albedo can reduce 1-3%, 2-5%, and 2-4% due to ng g, XXX ng g, and xxx ng g of black carbon (BC), and mineral dust (MD) in snow. We note that tThe organic carbon (OC) is also another key parameter in affecting snow albedo of 1-5% due to XXX ppb in snow due to SDAMS model simulation. For a given shape (Koch snowflake, hexagonal plates/columns, and spheres), the snow albedo reduction due to spherical snow grains is graduately larger than Koch snowflake, and hexagonal plates/colums with the increased concentration of BC in snow. The internal mixing of BC in snow absorbs substantially higher than the external mixing at the wavelengths of 400 nm–1400 nm. In addition to the BC and AD parameters in the Snow, Ice, and Aerosol Radiation (SNICAR) model, the OC content in snow is considered an initial parameter for calculating snow albedo through a new radiative transfer model (Spectral Albedo Model for Dirty Snow, or SAMDS). The spectral albedo of snow reduction caused by OC (20 μg g$^{-1}$) is up to a factor of 3 for a snow grain size of 800 μm compared to 100 μm. We find a larger difference in snow albedo levels between the model simulations and surface measurements for higher insoluble light-absorbing impurities (ILAPs) using the measured snow grain radii.  Compared with the observed snow albedo, we also note that the optical effective radii (R$_{eff}$) of snow grains can significantly enhance the model simulations of snow albedo reduction than that the measured snow grian radii due to SMDAS and SNICAR models, 
[revised manuscript text omitted]
 aggregates and snow grain shapes (fractal particlesgrains, hexagonal plates/columns, and spheres) with the snow albedo reduction due to and internal and /external mixing structuresof BC and snow on snow albedo. Finally, conclusions areConcluding and discussing remarks are given in section 4. across northeastern China, which are highly correlated with industrial pollution resulting from human activity. Therefore, ILAPs in seasonal snow are examined during a snow campaign, and the snow albedo is measured using an HR-1024 field spectroradiometer and simulated using two radiative transfer models (i.e., SNICAR and SAMDS).

**2    Experimental procedures**

**2.1    Snow field campaign in January 2014 January**

In 2014, tThere was less snowfall in January 2014 than in previous years (e.g., 2010), and thus only 92 snow samples (13 surface snow including 10 fresh and 3 aged ones, and 79 sub-surface snow samples) at 13 sites were collected during this snow survey.

  The snow sampling sites in this study were numbered starting  at 90 (see Figure 1 and Table 1) following the chronological order from Wang et al. (2013a), and Ye et al. (2012). Samples  at sites 90-93 were collected from grassland and cropland areas in Inner Mongolia. Sites 94-98 and  99-102 were located in the Heilongjiang and Jilin provinces, respectively, which  were the most heavily polluted areas in northern China during winter. The snow sampling procedures  were similar to those used in the previous survey conducted in 2010 across northern China (Huang et al., 2011). To prevent contamination, the sampling sites were positioned 50 km from cities and at least 1 km upwind of approach roads or railways; the only exception was site 101, which was positioned downwind and close to villages. Two vertical profiles of snow samples ("left" and "right") were collected through the whole depth of the snowpack at all the sites to reduce the possible contamination by artificial effects during the sampling process, and  the dusty or polluted layers were separately collected during the sampling process. All of the datasets  $C_{BC}^{equiv}$, $C_{BC}^{equiv}$,  $C_{BC}^{est}$ in seasonal snow listed in  Table 1 are  average values from the two adjacent snow samples through the whole depth of the snowpack. Snow grain sizes ($R_m$) were measured by visual inspection on millimeter-gridded sheets viewed through a magnifying glass. The snow samples were kept frozen until the filtration process was initiated. In a temporary lab based in a hotel, we quickly melted the snow samples in a microwave, let them settle for 3-5

minutes, and then filtered the resulting water samples through a 0.4-µm nuclepore

filter to extract particulates.

**2.2    Chemical speciation**

Major water-soluble ions and trace elements in surface snow samples during

this snow survey have already been investigated by Wang et al. (2015). However, the

importance of ILAPs in seasonal snow during this survey has not been discussed

 yet. which will be addressed below.

analysed

 Briefly, major ions ($SO_4^{2-}$ , $NO_3^-$ , Cl⁻, Na⁺, K⁺, and

$NH_4^+$) were analyzed with an ion chromatograph (Dionex, Sunnyvale, CA), and

trace elements of Fe and Al were measured by inductively coupled plasma mass

spectrometry (ICP-MS). These analytical procedures have been described elsewhere

(Yesubabu et al., 2014). In this paper, the major ions are used to retrieve the sea salt

 and biosmoke potassium ($K_{Biosmoke}^+$) $K_{Biosmoke}^+$

 Previous studies have revealed considerable variations in iron (Fe) of

2-5% in dust (Lafon et al., 2006), although Al is more stable than Fe in the earth's crust. Hence, we retrieved the mass concentration of  MD via the Al concentration assuming a fraction of 7% in MD (Arhami et al., 2006; Lorenz et al., 2006; Zhang et al., 2003). Sea salt was estimated following the method presented in Pio et al. (2007):

$$\text{Sea salt} = \text{Na}_{Ss}^{+} + \text{Cl}^{-} + 0.12\text{Na}_{Ss}^{+} + 0.038\text{Na}_{Ss}^{+} + 0.038\text{Na}_{Ss}^{+} + 0.25\text{Na}_{Ss}^{+}, \tag{1}$$

where subscript Ss means sea salt sources,  $\text{Na}_{Ss}$ was calculated using the following formula (Hsu et al., 2009):

$$\text{Na}_{Ss} = \text{Na}_{Total} - \text{Al} \times (\text{Na/Al})_{Crust}. \tag{2}$$

Following Hsu et al. (2009), the contribution of  $\text{K}_{Biosmoke}^{+}$ was determined using the following equations:

$$\text{K}_{Biosmoke}^{+} = \text{K}_{Total} - \text{K}_{Dust} - \text{K}_{Ss}, \tag{3}$$

$$\text{K}_{Dust} = \text{Al} \times (\text{K/Al})_{Crust}, \tag{4}$$

$$\text{Na}_{Ss} = \text{Na}_{Total} - \text{Al} \times (\text{Na/Al})_{Crust}, \tag{5}$$

$$\text{K}_{Ss} = \text{Na}_{Ss} \times 0.038, \tag{6}$$

where  $\text{K}_{Biosmoke}^{+}$ $\text{K}_{Dust}$ and $\text{K}_{Ss}$ refer to biosmoke potassium, dust-derived potassium, and sea-salt-derived potassium, respectively. Equations (4), (5), and (6) were derived from Hsu et al. (2009) and Pio et al. (2007).

**2.3 Spectrophotometric analysis**

Recent studies indicated that the light absorption by MD should be more sensitive to the presence of strongly absorbing iron oxides such as hematite and goethite than to other minerals (Alfaro et al., 2004; Sokolik and Toon, 1999)MD

their content in iron oxides (hematite, goethite, etc.) based on Mie theory. Sokolik and Toon., (1999) also pointed out that computations performed with optical models show that the absorbing potential of mineral dustMD is 
[revised manuscript text omitted]

10   varied considerably from 0.07 to 1.3 mm. $R_m$The snow grain size increased with the snow depth from the surface to the bottom and was, larger than that previously recorded in previous studies because of snow melting by solar radiation and the ILAPs (Hadley and Kirchstetter, 2012; Motoyoshi et al., 2005; Painter et al., 2013; Pedersen et al., 2015). The snow density exhibited little geographical variation across

15   northern China at 0.13 to 0.38 g cm$^{-3}$. High snow densities were resulted from melting or snow aging. Similar snow densities have been found in the Xinjiang region in northern China (Ye et al., 2012). AtIn this studysite 90, we only collected one layer of snow samples from central Inner Mongolia, and $C_{BC}^{est}$the BC mixing ratio was 3304 ng g$^{-1}$ forin aged snow. Along the northern Chinese border at sites 91-95, $C_{BC}^{est}$BC

20   contamination in the cleanest snow ranged from 3027 to 260 ng g$^{-1}$; with only a few values exceeded 200 ng g$^{-1}$. The $f_{nonBC}^{est}$ f$_{non-BC}^{est}$ value varied remarkably from 29 to 78%, although BC was still a major absorber in this region. Heavily polluted sites were located in industrial regions across northeastern China (sites 99-102). The surface snow $C_{BC}^{est}$ BC in this region ranged from 51008 to 3651700 ng g$^{-1}$, and the

highest $C_{BC}^{est}$  in the sub-surface layer of the four sites  was 29882 ng g$^{-1}$ ( Table 1). In addition,  $f_{nonBC}^{est}$ $f_{non\text{-}BC}^{est}$ was typically 35-74%, indicating significant light-absorbing contributions by

5    OC and MD from human activity in the heavily polluted areas. Å$_{tot}$ ranged from 2.1 to 4.8. A higher Å$_{tot}$ is a good indicator of soil dust, which is primarily driven by the composition of mineral or soil dust. In contrast, a lower Å$_{tot}$ of 0.8-1.2 indicates that ILAPs in the snow are dominated by BC (Bergstrom et al., 2002; Bond et al., 1999).

10  To better understand the  distribution of $C_{BC}^{est}$ $C_{BC}^{est}$ in seasonal snow across northern China,  the spatial distribution of  $C_{BC}^{est}$ $C_{BC}^{est}$  in the surface and average snow measured during the snow  surveyare shown in Figure 4 . The spatial distributions of $C_{BC}^{est}$ $C_{BC}^{est}$  in the surface and average snow

[revised manuscript text omitted]

5 in snow. It shows that snow albedo by spherical snow grains is typically decreaselower by 0.0175-0.0763 than the fractal snow grains, and by 0.008-0.036 than the, and 0.0087-0.0361 as a function of BC mixing ratios (0-5000 ng g$^{-1}$), which is comparedthan that with by the fractal snow grains and hexagonal plates/columns snow grains as a function of BC mixing ratios (0-5000 ng g$^{-1}$). Dang et al. (2016)

10 assessed the effects of snow grain shape on snow albedo using the asymmetry factors g of nonspherical ice crystal developed by Fu (2007). They obtained similar result that the albedo reduction caused by 100 ng g$^{-1}$ of BC for spherical snow grains is larger by 0.007 than nonspherical snow grains with the same area-to-mass ratio for $R_{eff}$ of 100 μm. Figure 10b shows the spectral albedo of snow for the internal/external mixing of

15 BC and snow with $R_{eff}$ of 100 μm for a solar zenith angle $\theta$ of 60° as a function of BC mixing ratio. For a given shape (hexagonal plates/columns), we find found that snow albedo as a function of BC mixing ratios calculated from this study decreases as the fraction of the internal mixing increases (Figure 101b). In previous studies, the BC mixing ratioss in seasonal snow wereis up to 3000 ng g$^{-1}$ ppb 
[revised manuscript text omitted]
. In this study, we indicated that thepresent a new spectral snow albedo model (SAMDS) for simulating the surface albedo of snow with deposited ILAPsaerosol impurities (e.g. Black carbon, Organic carbon, Mineral dust, volcano ash, and snow algae) by using the asymptotic analytical radiative transfer theory. Given the measured BC, MD and OC mixing ratios of 100-5000 ng $g^{-1}$, 2000-6000 ng $g^{-1}$, and 1000-30000 ng $g^{-1}$ in surface snow across northeastern China, we ran the models at a solar zenith angle $\theta$ of 60°, and the results indicated that the albedo of fresh snow at 550 nm is generally in a range of 0.95-0.75 with $R_{eff}$ of 100 μm. This model can also be used to investigate the snow albedo influenced by the internal/external mixing of BC and snowwith impurities, irregular morphology of snow grains and impurities , aging processes of snow grains and soot aggregates, and the vertical distribution of snow grains and impurities for multilayer snow. Additionally, the properties of different snow grain shapes (Fractal particles, Hexagonal plate/column, and spheres) and the internal/external mixing with BC in snow by using SAMDS model might be useful to researchers who are conducting studies involving ILAPs and snow interaction and feed back in snow albedo reduciton. Compare to the SNICAR model, the snow albedo reduction is in agreement with the

SAMDS model, different types of impurity could be included in the parameterization in SAMDS model, such as organic carbon and biogenic particles. For instance, thea given shape (spheres, hexagonal plates/columns, and fractal particles), it shows that snow albedo for spherical snow grains is typically lower by 0.017-0.073, and 0.008-0.036 than that for the fractal snow grains and hexagonal plates/columns snow grains as a function of BC mixing ratios (0-5000 ng g$^{-1}$) with R$_{eff}$ of 100 μm.it shows that snow albedo by spherical snow grains typically decrease by 0.017-0.073, and 0.008-0.036 as a function of BC mixing ratios (0-5000 ng g$^{-1}$), which is compared with the fractal snow grains and hexagonal plates/columns snow grains. The internal mixing of BC and -snow absorbs substantially more light than external mixing subsequently. For fresh snow grains of hexagonal plates/columns with R$_{eff}$ of 100 μm, the difference of snow albedo between internal and external mixing of BC and snow is up to 0.036 for 3000 ng g$^{-1}$ BC in snow in the heavy industrial regions across northeastern China, whereas by low to 0.005 for 100 ng g$^{-1}$ BC in snow in the further north China near the border of Siberia. The spectral albedo of snow reduction caused by OC (20 μg g$^{-1}$) is larger by up to a factor of 3 for a snow grain size of 800 μm compared to 100 μm by using SAMDS model.

Then, OC emitted from biomass burning and SO$_4^{2-}$ and NO$_3^{-}$ generated from fossil fuels and biofuels also played key roles in the mixing ratios of chemical components in seasonal snow. Finally, aAa comparison between measured and simulated snow albedos was conducted. Generally, tThe sSnow albedos measured from a spectroradiometer and simulated using the SNICAR and SAMDS models using R$_m$ [LZ1]agreed well with the measured ones from the spectroradiometer with at the lower mixing ratios of BC, OCMD, and ADMDOC, but with. However, a large discrepanciesdiscrepancy 
[revised manuscript text omitted]

[Figure]

Absorption (%)
- ■ BC
- ■ OC
- ■ AD

| 0 | Water | 5 | Mixed Forest | 10 | Grasslands | 15 | Snow and Ice |
| 1 | Evergreen Needleleaf | 6 | Closed Shrublands | 11 | Permanent Wetlands | 16 | Bare orsparesly vegetate |
| 2 | Evergreen Broadleaf | 7 | Open Shrublands | 12 | Croplands | 254 | Unclassified |
| 3 | Deciduous Needleleaf | 8 | Woody Savannas | 13 | Urban and Built-up | | |
| 4 | Deciduous Broadleaf | 9 | Savannas | 14 | Cropland Mosaics | | |

[Figure]

Absorption (%)
- ■ BC
- ■ OC
- ■ MD

[revised manuscript text omitted]